# Necessary Storage As a Signature of Discharge Variability: Towards Global Maps

Kuniyoshi Takeuchi[1,*] and Muhammad Masood[2]

[1]International Centre for Water Hazard and Risk Management (ICHARM), Public Works Research Institute (PWRI), Tsukuba, 305-8516, Japan
[2]Bangladesh Water Development Board (BWDB), Design Circle-1, Dhaka, Bangladesh

*Correspondence to*: Kuniyoshi Takeuchi (takeuchi@yamanashi.ac.jp)

**Abstract.** This paper proposes the use of necessary storage to smooth out discharge variability to meet a discharge target for flood and drought management as a signature of discharge variability in time. Such a signature has a distinct advantage over other statistical indicators such as standard deviation (SD) or coefficient of variation (CV) as it expresses hydrological variability in human terms, which directly indicates the difficulty and ease of managing discharge variation for water resource management. The signature is presented in the form of geographical distribution, both in terms of necessary storage [km$^3$] and normalized necessary storage [months], and is related to the basin characteristics of hydrological heterogeneity including geophysical and geographical heterogeneities. The signature is analysed in different basins considering the Hurst equation of Range as a reference. which is expressed as a function of SD and CV. The slope of such a relation and the scatter of departures from the average relation are analysed with their relationship to basin characteristics of hydrological heterogeneity. As a method of calculating necessary storage, the flood duration curve (FDC) and drought duration curve (DDC) method is employed for its advantage over other methods to calculate the necessary storage easily over many grids in a large basin. The Ganges-Brahmaputra-Meghna (GBM) basin is selected for a case study and the hydrological model BTOPMC with the WATCH Forcing Data (WFD) is used for estimating FDC and DDC. It is concluded that the necessary storage serves as a signature of discharge variability and it is worthwhile to extend analysis of its geographical distribution for global mapping, and in the way seeking new insight into hydrological variability in storage domain at a range of scales.

*Currently at Iwakubo-cho 392-2, Kofu, Yamanashi 400-0013, Japan

## 1 Introduction

Storage is the only means to smooth out discharge variability to meet targets for flood and drought management. Necessary storage depends on discharge variation, flood control targets, and water use targets. By fixing flood control and water use targets, necessary storage depends only on discharge variability and serves as a signature of discharge variability expressed in human terms, which directly conveys the sense of the difficulty or ease of controlling discharge for water resources use. For this reason, such a signature has a distinct advantage over other statistical indicators of variability such as standard deviation (SD) or coefficient of variation (CV). Besides, by plotting the signature in a geographical map, hydrological variability in time and space can be compared with the geographical distribution of hydrological (including geophysical and geographical) heterogeneity over a basin such as hydro-meteorological, topographical, geological, pedological, vegetation and land-use conditions. This paper is the first attempt to look into how hydrological variability compares with the signature, i.e., necessary storage, through its absolute volume [km$^3$] or its normalized form [months] divided by a local long-term mean discharge, to get new insights into the factors controlling the relationship between hydrological variability and hydrological heterogeneity.

[ここに入力]

## 1.1 Objectives

This paper has three objectives. One is to propose necessary storages to smooth out discharge variation during floods and droughts as a new or an alternative indicator or a signature of discharge variability in time and compare it with standard variability indices such as SD and CV. Another is to present geographical distribution of this signature and explore how the signature is related to the various hydrological heterogeneity of a basin. The last objective is to introduce an efficient methodology to calculate necessary storage at all grid points of a basin to perform geographical analysis using the flood duration curve and drought duration curve (FDC-DDC) method. These objectives are elaborated below.

### 1.1.1 Necessary storage as a signature of discharge variability in time

Necessary storage to smooth out the variation of flow in time to control hazards and utilize resources depends on the variation of flow, flood channel capacity and the target level of water use. While flood channel capacity and the target water use are socio-economic parameters varying with societal needs for water and the environment, flow variation depends on natural hydrological phenomena of a basin over time and space. Therefore, if socio-economic targets are fixed at certain levels, necessary storage can be used as a signature of natural flow variability. While flow variability is very often described by the standard deviation (SD) or the coefficient of variation (CV) of discharge time series, its physical meaning is not necessarily clear in human terms. On the other hand, necessary storage has a concrete meaning that can be directly related with the physical size of a reservoir necessary for flood and drought management. Necessary storage is a necessary physical volume of storage space for flood control or volume of water stored for drought management in order to keep outflow downstream at a certain level. In this paper, we propose the use of necessary storage as a signature of hydrological variability in time and try to relate it with the hydrological heterogeneity of a basin over time and space.

Hydrological heterogeneity has been studied by many researchers such as Wood et al. (1988) in terms of scale effects, Creager et al. (1945) in terms of extremes, and Blöschl et al. (2013) in terms of predictability. This paper extends such analyses to water resources manageability in terms of necessary storage to smooth out hydrological variation as an integrated signature. Blöschl et al. (2013) proposed 6 runoff signatures that determine runoff variability: namely, annual runoff, seasonal runoff, flow duration curve, low flow, floods and hydrographs. They are indeed important indicators of river runoff phenomena especially for analytical diagnosis on flow prediction. On the other hand, for flood control and water supply, managers would be more interested in integrated information that indicates water resources manageability. For that purpose, necessary storage to smooth out discharge variation into given levels of constant flow can be as an informative indicator. Necessary storage is an integrated result of hydrological variation that reflects short- and long-term means; daily, seasonal and long-term patterns of variation; extremes, duration or persistence characteristics, and frequencies or periodicities; available flood channel capacity, necessary water withdrawal, etc. It is therefore truly an integrated signature of discharge variability and an integrated indicator of manageability of discharge variation.

### 1.1.2 Geographical mapping of necessary storage

In order to analyse necessary storage as a signature of discharge variability in relation to hydrological heterogeneity in a basin, it is essential to plot the signature in geographical distribution maps. For this purpose, necessary storage is calculated not only along the main streams in a river basin as it is often practiced for reservoir design but also at all grid cells over a widespread area where the stream networks are formed. It is areal information rather than point information. This calculation method can help water resources managers grasp regional discharge variability of floods and droughts in natural conditions with relative ease.

In this paper, however, calculated necessary storage cannot be interpreted as actual necessary storage for reservoir design and water management since the actual status of channel capacity and water withdrawal at a given location are treated as given parameters and not accounted for in the calculation. It is only a signature of discharge variability in the unit of necessary

[ここに入力]

storage to maintain certain constant flow levels for flood control and water use in downstream areas. As certain flow levels, the mean annual flow or some percentage of it is used to help users to easily imagine the assumptions.

Geographical maps of necessary storage for flood and drought will be compared with geographical maps of elevation, precipitation, land use and land cover, discharge and its standard deviation and coefficient of variation. In addition, using scatter diagrams, the hydrological heterogeneity will be analysed by looking into the slope of the average relation and the scatter of departures from the average relation of necessary storages with catchment area A, SD and CV of various sub-basins. Such departure will be related with the hydrological characteristics of different basins. It will be the first step of looking into the relation between necessary storage and hydrological heterogeneity including land cover and other geographical conditions.

### 1.1.3 Flood Duration Curve and Drought Duration Curve (FDC-DDC) to calculate necessary storages

In order to produce a spatial distribution map of necessary storage of a large basin, eventually of the globe, it is necessary to employ an efficient methodology for calculating necessary storage. There are various ways of calculating necessary storage depending on various conditions and objectives. The most widely known is the mass curve method developed by Rippl (1883), which has been used over a century as a standard methodology of calculating necessary storage with a given constant target water use (Klemes, 1979). For more complicated target water use patterns, simulation method was introduced and extensively studied by the Harvard Water Program in the 1950s (Maass et al., 1962). The former initiated epoch making research movements on storage analyses of Range (Hurst, 1951; Feller, 1951; Moran, 1959), and the latter on streamflow synthesis (Thomas and Fiering, 1962; Young, 1967; Mandelbrot and Wallis, 1969; Valencia and Schaake, 1972).

In this paper, instead of those methods, the flood duration curve and drought duration curve (FDC-DDC) method was used as it is relatively easy to calculate necessary storage under any given constant level of target release and better suited to calculate necessary storage at many grid points over a space for areal mapping of reservoir storage. Such a relative ease comes from two simplifications, a stationarity assumption and a short-term focus. The stationary assumption is inherent to the use of intensity-duration-frequency curves and its estimation accuracy is bounded by the available length of data. The short-term focus in an annual smoothing rather than inter-annual smoothing is in a sharp contrast to a long-term asymptotic focus of the Range analyses. It is unacceptable in general for water resources management but is used here for simplicity as an example to introduce the indicator for eventual global presentation and analyses. The main theory of the Range analysis is briefly introduced in 2.3.2. The drought duration curve (DDC) was first proposed by Kikkawa and Takeuchi in the 1970s (Kikkawa and Takeuchi, 1975; Takeuchi and Kikkawa, 1980). The DDC was shown useful for reservoir operation (Takeuchi, 1986) and by adding flood duration curve (FDC), the FDC-DDC was also proven useful for hydrological statistics and classification (Takeuchi, 1988).

## 1.2 The case study area

This paper takes a case study approach to demonstrate the proposed use of necessary storage as an indicator of discharge variability. For that purpose, the Ganges-Brahmaputra-Meghna (GBM) basin was selected as it has heterogeneous geographical conditions and is large enough to eventually extend the analyses into the globe. Figure 1 depicts the Ganges-Brahmaputra-Meghna (GBM) basin (Pfly, 2011). According to FAO AQUASTAT (2011), the total basin area is about 1.7 million km$^2$, shared by India (64%), China (18%), Nepal (9%), Bangladesh (7%) and Bhutan (3%). It is the world's third largest freshwater outlet to the oceans. Figure 2 shows the land cover distribution which well reflects the elevation and precipitation depicted in Fig. 3a and 3b, respectively. The land cover data were collected from the Global Land Cover by National Mapping Organizations (GLCNMO) which was prepared by using MODIS data with remote sensing technology (Tateishi *et al.*, 2014). The GLCNMO classifies the status of land cover of the whole globe into 20 categories based on the Land Cover Classification System (LCCS) developed by FAO. The elevation data of Fig. 3a is acquired from HydroSHEDS, which is derived from remote sensing data of the Shuttle Radar Topography Mission (SRTM) at 3 arc-second (90 m) resolution during an 11-day

[ここに入力]

mission in February 2000 (Lehner *et al*. 2006). The mean precipitation map of Fig. 3b is created from the daily rainfall and snowfall data from WATCH Forcing Data (WFD) set to be explained in 3.1.1.

These data would be useful to relate the geographical distribution of necessary storage indicators to the hydrological heterogeneity of a basin. All geographical maps except Fig. 1 indicate the borders of the Ganges, the Brahmaputra and the Meghna in thick black lines, the selected inner sub-basins in thin black lines and national boundaries in white lines.

As Fig. 3b indicates, the Ganges basin is characterized by a large spatial variation of precipitation that causes water scarcity in the western Ganges and water excess in the southeast. The Ganges is a snowmelt-fed river, which is regulated by 75 artificial dams (Lehner et al., 2011). The Brahmaputra basin is characterized by high precipitation in the eastern India and a large amount of snow in the upstream that provides a huge volume of discharge to the river. On the other hand, the Meghna is a comparatively smaller, rain-fed, and relatively flashier river. The basin contains the world's top two highest precipitation (about 12,000 mm year$^{-1}$) areas; Mawsynram and Cherrapunji (Masood and Takeuchi, 2015b).

The GBM river basin contains about a tenth of the world's population. As the population is still steadily increasing, the usage of water is increasing rapidly to meet municipal, agricultural and industrial water demands. In addition, the basin is also recognized as one of the areas most prone to waterborne natural hazards, floods and droughts in the world, which threat a large number of population each year. Therefore, the management of water resources plays a crucial part in ensuring the sustainability of the region.

## 1.3 Structure of the paper

In Chapter 2, the methodology is presented; namely, the flood and drought duration curves, FDC-DDC, in 2.1 and the method of calculating necessary storage using FDC-DDC in 2.2. The comparison of this method with the standard mass curve method and the analysis of spatial distribution of necessary storage in contrast to analysis of asymptotic temporal nature several decades ago are also discussed in 2.3. In Chapter 3, application of the methodology to the case study area, the GBM basin, is described. Also described are the necessary precipitation and temperature data set, Watch Forcing Data (WFD), and the BTOPMC hydrological model to translate them to discharge data. Chapter 4 presents application results and discusses their implications. Discussed in Chapter 4.1 are geographical distribution of necessary storage, in 4.2 the relation between necessary storage and catchment characteristics such as area A and SD and CV of discharge, and in 4.3 their zoom up relations in the selected 12 sub-basins. Finally, Chapter 5 presents conclusions.

## 2 Methodology

This chapter first introduces the methodology of calculating FDC-DDC in 2.1, which basically follows a similar notation used in Takeuchi (1988) with slight modifications to a simpler form neglecting seasonal parameter τ. Section 2.2 introduces the methodology of calculating necessary storage which follows the procedure originally introduced in Kikkawa and Takeuchi (1975) and in Takeuchi (1986) but in a more elaborated way. Finally, in 2.3, other discussions related to this paper will be introduced, including the relation between the FDC-DDC method and the traditional mass curve method (Takeuchi and Kikkawa, 1980) and the discussion of Hurst formula (Hurst, 1951).

## 2.1 Flood duration curve (FDC) and drought duration curve (DDC)

Both FDC and DDC are basically intensity-duration-frequency (IDF) curves of discharge time series, but not on actual discharge data but their moving averages. Let $x_t$ denote any hydrological variable at time $t$ and random variable $X(m)$ denote the annual maximum of moving averages of any $x_t$ over m days starting any day $t_1$ of a certain year. Its quantile value for exceedance probability α denoted by $f_a(m)$ is defined as a flood duration curve. Similarly random variable $X'(m)$ denotes the annual minimum of moving averages of m days starting from any day belonging to the year. Its quantile value for non-

[ここに入力]

exceedance probability β denoted by $f'_\beta(m)$ is defined as a drought duration curve. Namely,

$$X(m) = \max_{t_1 \epsilon\, a\; certain\; year} \frac{1}{m} \sum_{t=t_1}^{t_1+m-1} x_t \tag{1}$$

$$X'(m) = \min_{t_1 \epsilon\, a\; certain\; year} \frac{1}{m} \sum_{t=t_1}^{t_1+m-1} x_t \tag{2}$$

$$\text{Prob}\big(X(m) \geq f_\alpha(m)\big) \leq \alpha \tag{3}$$

$$\text{Prob}\big(X'(m) \leq f'_\beta(m)\big) \leq \beta \tag{4}$$

In this study, hydrological variable $x_t$ is discharge and the quantiles of flood duration curve (FDC) $f_a(m)$ and drought duration curve (DDC) $f'_\beta(m)$ in Eq. 3 and 4 were estimated by fitting Generalized Extreme Value (GEV) distribution, where parameters were estimated by the maximum likelihood method for α and β being 0.2, 0.1, 0.05 and 0.02 corresponding to 5, 10, 20 and 50-year return periods (T). The GEV distribution Type-1, Gumbel distribution, was used in this study as it was recommended for the major rivers in Bangladesh by Mirza (2002) and for relatively smaller data samples by Hirabayashi et al. (2013).

Figure 4a depicts the FDC and DDC of the Brahmaputra River at Bahadurabad station (Fig. 1). It shows for $m$ up to 1095 days or three years. The duration curves oscillate with duration time length reflecting the annual periodicity of hydrograph. For higher return periods, the duration curves lay outer from the long term mean discharge line, implying higher flood discharges and lower drought discharges. In this paper, however, as will be explained in 2.3.2, the portion for $m$ up to 365 days or one year, i.e., Fig. 4b, is used throughout the discussion to get a practical signature.

The duration curves are theoretically nothing but intensity-duration-frequency (IDF) curves, but practically they are considerably different. Ordinarily IDF curves are nearly exclusively used for design of storm drainage systems and accordingly concern precipitation on its high intensity side for a rather short term such as several hours to a few days. On the other hand, the FDC-DDC applies precipitation, discharge or any other time series not only on the high intensity side but also on the low intensity side. Also the concerned time length or duration extends to over months to multiple years.

In fact the original development of this duration curve was DDC rather than FDC for reservoir management during a drought and the idea was totally independent of IDF (Kikkawa and Takeuchi, 1975). Its central interest was the minimum discharge that can be expected for $m$ days in the future as the worst case scenario. Takeuchi (1986) applied the DDC to chance-constrained reservoir operation as DDC $f'_\beta(m)$ indicates the average inflow that can be expected for any $m$ days in the future with the failure rate less than $\beta$. Further later, the idea was extended to the flood side and FDC-DDC was developed as a means of classifying the persistence characteristics of regional hydrology, precipitation and discharge that serve as an indicator or a palm print of basin hydrology (Takeuchi, 1988).

## 2.2    Necessary storage

This section illustrates the detailed procedure of calculating necessary storage to smooth out discharge variations over time using FDC-DDC. In short, necessary storage is obtained as the largest inner rectangular area that just fits to the area surrounded by a duration curve, the target discharge level (such as the long term mean) and the vertical axis at the origin. Necessary storage indicates the empty space necessary for flood control at the beginning of the flood season and the stored volume of water necessary for water supply at the beginning of the drought season with the failure rate indicated by the duration curves.

### 2.2.1    Necessary storage to smooth out high flows (floods)

Figure 5 depicts a schematic FDC-DDC of discharge at the dam site concerned.

1. Suppose at a time before the flood season, a flood manager considers how much flood control space he/she needs to smooth out all the high flows expected in the season.
2. Suppose the flood channel capacity or the target river discharge level is the long term mean of river discharge EM in Fig. 5 (which may not be realistic but useful for simplicity to get a practical signature).

[ここに入力]

3. Suppose he/she chooses return period 20 years or failure rate 0.05. Then he/she chooses a flood duration curve $f_{0.05}(m)$.

4. Suppose he focuses on point A on the FDC $f_{0.05}(m)$ where $m$=50 days.

5. Horizontal line DA passing through point A, namely $f_{0.05}(50)$, is the annual maximum average discharge per day over any 50 days starting from any date in any year that will exceed only with probability 0.05. In other words, this level is the average flood discharge that the flood manager expects and likes to smooth out.

6. How much volume is the total flood discharge in those 50 days? It is simply $f_{0.05}(50) * 50$, which is the area of rectangular ADOC.

7. Now, how much flood discharge can the river safely flow down, or what is the flood channel capacity? That is, he assumes a long-term average discharge indicated by line EB (on EM). The flood volume that can safely flow down is the area indicated by rectangular BEOC.

8. Therefore, the necessary storage capacity the flood manger needs to prepare for flood space before the flood season is ADOC-BEOC=ADEB.

9. Now the flood manger has to consider the necessary storage space by moving point A all the way from the start to the end, in this case point $A_0$ to $A_{end}$, and identify the largest volume necessary for flood control. This can be expressed as:

$$V_{f,\alpha} = \max_m m * f_\alpha(m) \tag{5}$$

In this study, the interest duration is limited to a year, neglecting the need of multi-year smoothing. In reality, an over year storage is very important and in many practical cases critical, especially in arid/semi-arid zones, but as a signature to be examined eventually by a global map, we first concentrate on intra-annual smoothing for simplicity. Necessary storage for inter-annual smoothing increases with time, eventually to infinity, as Hurst (1951) showed, which is briefly discussed in 2.3.2. By this simplification, necessary storage can be expressed in a narrative way as the area of the largest rectangular that just fits to a right triangle surrounded by the flood duration curve, the channel capacity line, and the vertical axis of the origin.

### 2.2.2 Necessary storage to smooth out low flows (droughts)

A similar discussion for necessary storage for drought management will follow below. Again Fig. 5 will be used for explanation.

1. Suppose a drought manager considers how much water is necessary to be stored before the dry season starts.

2. Suppose he/she is obliged to keep supplying water equal to the long term mean of river discharge, which is indicated by line EM.

3. Suppose he/she chooses return period 20 years or failure rate 0.05. Then he/she chooses a drought duration curve $f'_{0.05}(m')$.

4. Suppose he/she focuses on point A' on the DDC $f'_{0.05}(m')$ where $m'$=150 days.

5. Horizontal line D'A' passing through point A', namely $f'_{0.05}(150)$, is the annual minimum average discharge per day over any 150 days in a year, which will fail only with probability 0.05. In other words, this level is the average low flow over the next 150 days, which the drought manager expects and likes to go around by augmentation from reservoir storage.

6. How much volume is the total discharge he can expect in those 150 days? It is simply $f'_{0.05}(150) * 150$, which is the area of rectangular A'D'OC'.

7. Now, how much water should be released to meet the water supply target during the next 150 days? The drought manger is obliged to supply the long-term average discharge indicated by line EB'. The necessary volume to be released to meet the target is indicated by the area of rectangular B'EOC'.

8. Therefore the necessary reservoir storage the drought manger needs to prepare for drought management before the dry season starts is A'D'OC'-B'EOC'=A'D'EB'.

9. Now the drought manger has to consider the necessary storage to be reserved for augmentation by moving point A' all the way from the start to the end, in this case point $A'_0$ to $A'_{end}$, and identify the largest volume necessary for drought management. This can be expressed as:

[ここに入力]

$$V_{d,\beta} = \max_{m'} m' * f'_\beta(m') \tag{6}$$

Again in this study, the range of averaging time length $m$ was limited to 365 days or a year. This assumption is more critical for drought management than for flood management as a multiyear drought is frequently experienced and a serious concern in many arid and semi-arid nations. But as a practical signature, the time length of a year is selected. Nevertheless, the methodologies themselves, i.e., Eq. (5) and (6), are valid for any $m$, and can work for calculating necessary storage for a multiyear drought regardless of its length.

### 2.2.3 With arbitrary target releases

Instead of assuming the target releases always equal to the long term mean, the targets can be set to the real flood channel capacity and the safe yield level of water supply as seen in Fig. 6. In such cases, the line of long-term mean EM in items 2 and 7 of the procedures above should be replaced by the channel capacity for flood control and the target water supply for drought management. Resultant necessary storage $V_f$ for flood control and $V_d$ for drought management can be calculated as in Fig. 6. As mentioned above, however, this study uses the long-term mean to get a signature for simplicity.

### 2.2.4 Expressions in km³ and months

Necessary storages $V_f$ and $V_d$ at each grid point may be expressed in [km³] and in general expressed as $V_{km3}$. However, the dominant areal distribution of necessary storage $V_{km3}$ would be similar to the distribution of catchment area because as catchment area increases, discharge increases in general and so does magnitude of flow variation, which would result in the increase of necessary storage to smooth out the variation. In order to better analyse necessary storage in relation to hydrological heterogeneity in a basin; therefore, a normalized signature $V_{month}$ expressed in [months] may be used by dividing the necessary storage volume $V_{km3}$ by local long term mean discharge $Q_{mean}$ [m³/s] expressed in [km³/months]. Namely,

$$V_{month} = V_{km3}/Q_{mean} \tag{7}$$

Value $V_{month}$ indicates an average residence or renewal time of water in the reservoir whose capacity is equal to the necessary storage and assumed full all the time. This is free from the mean annual flow and reflects other factors of hydrological variability.

## 2.3 Related discussions on necessary storage

### 2.3.1 Relation between the FDC-DDC method and the standard mass curve method

The methodology presented in 2.2 is considerably different from the standard mass curve method originally proposed by Rippl (1883) and widely used in engineering fields (Klemes, 1979). The major difference is its assignment of a return period. The original mass curve method does not translate the original hydrograph into the frequency domain or an intensity-duration-frequency (IDF) curve which has the same return period along a particular IDF curve for any duration. In reality there is no hydrograph that has always a same return period at any time for any duration but a mixture of many different high and low flow episodes with different return periods. As the mass curve method utilizes actual hydrological time series, the assignment of a return period or rate of failure is not necessarily in a strict manner. The total negative run sum or negative run length is often used to identify the return period which largely depends on the length of time on which an analysis focuses. The FDC-DDC method is free from such selection of length of time and the practical differences are in fact minor as shown by Takeuchi and Kikkawa (1980).

[ここに入力]

### 2.3.2 Necessary storage in temporal domain and in spatial domain

Research on Range will be briefly reviewed in this section as it is the analysis of necessary storage in a long-term asymptotic behaviour while this paper focuses, on the contrary, on a short-term spatial characteristics. Hurst (1951) considered necessary storage as "adjusted range" $R_n$ to keep long-term mean flow $Q_m = \bar{x}_n$ for $n$ consecutive time periods as follows:

$$S_t = S_{t-1} + (x_t - \bar{x}_n)$$
$$R_n = \max_{t=1,\ldots,n} S_t - \min_{t=1,\ldots,n} S_t$$

where $x_t$ is discharge at time $t$, $S_0 = 0$, $\bar{x}_n = \frac{1}{n}\sum_{t=1}^{n} x_t$, $s_n = \sqrt{\frac{1}{n-1}\sum_{t=1}^{n}(x_t - \bar{x}_n)^2}$

and found from his Nile study that:

$$R_n^* = R_n/s_n \propto n^H \tag{8}$$

where $R_n^*$ is named as "rescaled adjusted range" and $H$, the Hurst coefficient, was $\cong 0.72$.

This indicates that necessary storage $R_n$ increases limitless with time length $n$ to be considered and is proportional to $s_n$ as:

$$R_n \propto s_n n^H \tag{9}$$

or in a different form by dividing both sides by long-term mean $\bar{x}_n$ as:

$$R_n/\bar{x}_n \propto C_v n^H \tag{10}$$

where $C_v = s_n/\bar{x}_n$ coefficient of variation of discharge.

$R_n/\bar{x}_n$ is necessary storage normalized by the long-term mean flow which has the unit "time" and may be called normalized necessary storage or a mean residence time or a mean renewal time of water storage $R_n$ of the reservoir . Eq. 9 and 10 provide the theoretical background for $V_{km3}$ and $V_{month}$ of Eq. 7; namely, $V_{km3}$ is proportional to $s_n$ and $V_{month}$ is to $C_v$.

In this study, the length of available discharge data is 22 years (1980-2001) as explained in 3.1, and there is no way of
20 considering infinite length behaviour. But in theory, regardless of the choice of expression, formula Eq. 8 says that necessary storage $R_n$ or its normalized form $R_n/\bar{x}_n$ increases with time periods $n$ to be considered. It is a serious fact in terms of water resources management if long-term mean $\bar{x}_n$ is indeed the target of water supply or flood channel capacity, and it should be kept for a very long time such as over many years. But in reality, it is not usually the case because flood control and water supply targets do not usually require all fluctuations removed and keep the flow constant to the long-term mean. People living
along rivers are settled in the way to safely accommodate several-year return periods of natural floods and droughts. Namely, they can accommodate, without having storage reservoirs, floods far above the long-term mean discharge and can live with water use far below the long-term mean flow. It means that necessary smoothing of discharge variation over time is usually much shorter than multiple years. Especially in humid regions, particularly in small basins with limited reservoir capacity, the target water smoothing is often for a short-term in practice.

This paper therefore looks into spatial distribution of necessary storage with limited temporal scale, namely, a year or $m \leq$ 365 days. Admitting some contradiction in approach, the target water control level is still assumed as the long term mean discharge to make the discussion simple. Note again that the objective of this paper is not reservoir design for construction but introduction of a signature of discharge variability in relation to hydrological heterogeneity.

Based on such assumptions, necessary storage is calculated at each grid cell of a basin, and instead of focusing on their temporal
behaviour, their spatial behaviour will be examined. The geographical distribution of necessary storage will be discussed in 4.1, and the scale effects of normalized necessary storage will be discussed in 4.2.

### 3. Application

In order to demonstrate an example of the spatial distribution of FDC-DDC necessary storage, this study presents a case study in the Ganges-Brahmaputra-Meghna (GBM) basin. As the objective of the case study is methodological demonstration, no
other cases are analyzed.

[ここに入力]

### 3.1    Data used

The necessary discharge data at all internal grid points of the GBM basin for the case study were obtained by model simulation with the reanalysis precipitation and radiation data and the observed discharge data for calibration as described below.

### 3.1.1    Precipitation and temperature data used

The precipitation and temperature data used over the GBM basin for the input to the distributed hydrological model were the Water and Global Change (WATCH) Forcing Data set (WFD) (Weedon et al., 2011) for the period of January 1, 1980, to December 31, 2001. The WFD is the dataset based on the three-hourly ERA-40 reanalysis product of the European Centre for Medium Range Weather Forecasting (ECMWF) developed in the European Union WATCH project (www.eu-watch.org). ERA-40 was derived from successive short-term integrations of a general circulation model (GCM) that assimilated various
satellite data along with atmospheric soundings and land-sea surface observations. The one-degree resolution ERA-40 reanalysis data were interpolated into the half-degree resolution on the Climate Research Unit of the University of East Anglia (CRU) land mask. Average temperature and average diurnal temperature range from CRU TS2.1 gridded observations were used to remove monthly bias and lack of climatic trends which existed in 2-m temperatures product of ERA-40. After bilinear interpolation elevation correction on temperature data was done via environmental lapse rate. Monthly bias of precipitation
data was corrected by using CRU number of "wet days", version four of the Global Precipitation Climatology Centre (GPCCv4) precipitation totals and ERA-40 rainfall-snowfall proportion. FLUXNET data (https://fluxnet.ornl.gov/) were used to validate the data. For details on WFD data generation see Weedon et al., 2010, 2011.

### 3.1.2    Discharge data used

Using the forcing data above, the discharge data at all grid points of the GBM basin were created by model simulation using a
distributed hydrological model, BTOPMC. The model and the model set up and verification are described below in 3.2 and 3.3. For calibration of the model, the discharge data at Hardinge Bridge, Bahadurabad and Bhairab Bazar, three outlets of Ganges, Brahmaputra and Meghna, were constructed from the observed daily water level data, provided by the Processing and Flood Forecasting Circle, Bangladesh Water Development Board (BWDB) by using the rating equations developed by the Institute of Water Modelling (IWM, 2006) and Masood et al. (2015c).

### 25  3.2    Hydrological model to obtain discharge data

A physically-based distributed hydrological model, BTOPMC, was used for simulating runoff. The BTOPMC (Block-wise use of TOP model with Muskingum-Cunge method) was developed at the University of Yamanashi and ICHARM, PWRI, Japan (Takeuchi *et al.*, 1999, 2008; Ao *et al.*, 1999, 2006; Hapuarachchi *et al.*, 2008). It is an extension of TOPMODEL (Beven and Kirkby, 1979) to apply to large basins. The extension is made by introducing the effective contributing area concept; that
is, the discharge generation from a grid cell in a large basin is not necessarily contributed by its whole upstream catchment but only a portion of it. Based on this concept, the original topographical index is modified by replacing upstream catchment area $a$ by effective catchment area $af(a)$ and transmissibility coefficient $T_0$ by dischargeability $D_0$ (Takeuchi, 2008). For flow routing, basically the Muskingum-Cunge (MC) method (Cunge, 1969) is adopted to take diffusive factors into account. But a modification was made to conserve the continuity of water volume at each segment of river reach (Masutani and Magome,
2009). The BTOPMC has been applied in many river basins throughout the world including poorly gauged basins utilizing globally available data, and found that the BTOPMC can simulate river discharges quite well especially in warm humid regions (Takeuchi et al., 2013; Magome et al., 2015; Gusyev et al., 2016).

[ここに入力]

### 3.3 Model set up and verification

The BTOPMC model was set up for simulations of the WFD dataset at the 10-arc-min grid (approximately 20-km grid resolution) using DEM data derived from HydroSHEDS. The model set up procedure followed the work by Masood and Takeuchi (2015a) although their study used a hydrological model, instead of BTOPMC, H08 (Hanasaki, 2008).

The calibration period was from 1980 to 1990 (11 years) and verification was from 1991 to 2001 (11 years). Most of BTOPMC parameters are related to and identifiable by physical features of land cover and soil as specified by Takeuchi *et al.* (2008). For three particular parameters, decay factor ($m$), drying function ($\alpha$) and Manning's roughness co-efficient ($n_0$) were determined by calibration examining all the combinations of three parameters in 8 (eight) different values selected from their feasible physical ranges described in Takeuchi et al. (2008). A total of $8^3$ (=512) simulations were conducted. The identified parameters are listed in Table 1.

Figure 7 plots the daily hydrograph comparisons at the outlets of three river basins with the corresponding daily observations for both calibration and validation periods. Model performance was evaluated by comparing observed and simulated daily streamflows by the Nash-Sutcliffe efficiency (NSE) (Nash and Sutcliffe, 1970), the optimal objective function for assessing the overall fit of a hydrograph (Sevat and Dezetter, 1991). The obtained NSEs range from 0.80 to 0.91 for three basins (Table 1). Statistical indices suggest that the overall model performance is satisfactory.

The simulated daily discharge data set at all grid cells of the GBM basin were obtained. Figure 8a shows the distribution of annual mean discharge $Q_{mean}$ [m$^3$/s]; Fig. 8b, the distribution of standard deviation $s$ (SD) of daily discharge [m$^3$/s]; and Fig. 8c, the distribution of coefficient of variation $C_v$ (CV) of the daily discharge. The necessary storage was also calculated by the DDC-FDC method described in 2.2, the results of which will be discussed and analysed in the next section.

### 4. Results and discussion

In this chapter, the results obtained by the case study described in Chapter 3 will be presented and discussed. Namely, the geographical distribution of necessary storage over the basin was presented in 4.1, the effects of catchment area A and relation to statistical parameters of discharge SD and CV in 4.2, and their zoom-up relations in the selected 12 basins in the GBM basin in 4.3. Potential future study agenda are briefly discussed in 4.4.

### 4.1. The geographical distribution of necessary storage over the basin

Figures 9, 10 and 11 show the spatial distribution of necessary storage to smooth out the discharge variation over the Ganges-Brahmaputra-Meghna basin. Figures 9 and 10 assume the target release equal to long-term mean $Q_{mean}$ both for flood and drought management while Fig. 11 assumes $3Q_{mean}$ for flood management and $0.5Q_{mean}$ for drought management. Figure 9, 11a and 11b are in physical volume [km$^3$] and Fig. 10, 11c and 11d in residence time expression [months], the necessary storage divided by long-term mean $Q_{mean}$. All those figures directly imply the difficulty or ease of water resources management in a discharge variability aspect. Although the hydrological aspect is only a part of the actual difficulty of water resources management, its manageability in smoothing discharge variation is definitely an important factor. As Eq. 9 indicates, the distribution of necessary storage must be proportional in physical nature similar to the distribution of standard deviation (SD) of discharges in Fig. 8b, but its physical meaning in [km$^3$] is much more directly indicative to the size of dams, retardation ponds, tanks and the like than SD. Similarly, Eq. 10 indicates that the normalized necessary storage is proportional to the coefficient of variation (CV) of Fig. 8c, but the mean residence time or renewal time in [months] is much more understandable and instructive than CV for water resources managers.

Comparing Fig. 9 and 10, or necessary storage in [km$^3$] and in [months] under target release $Q_T=Q_{mean}$, the most obvious characteristics identifiable are:

[ここに入力]

1) The distributions of necessary storage for flood control and that for drought control; namely, the maps of Fig. 9a and 9b in km$^3$ and those of Fig. 10a and 10b in months are similar. But Fig. 9 and 10 are very different. It means that necessary storages for flood and drought are rather similar but the expressions in km$^3$ and in months make them totally different. It is quite reasonable, as the target release is set equal to long term mean $Q_{mean}$, that the necessary storages for flood control and drought control are the same in a long run. The difference is only due to the short term focus (one year in this case), and such focus makes especially the necessary storage for drought smaller, which is clearly visible in Fig. 10. Note that Fig. 9a and 9b seem to resemble each other more than Fig. 10a and 10b, which is simply because Fig. 9 is expressed in logarithm to make small storage visible while Fig. 10 in real number.

2) The main difference between Fig. 9 and 10 is in the main stream. In terms of km$^3$ in Fig. 9, along the main stream, the necessary storage increases as the catchment area increases towards downstream whereas in the upstream basins, the necessary storage is small and varies by location. But in terms of months in Fig. 10, along the main stream, months or the mean residence time decreases with catchment area. This is because the discharge increases in general with catchment area which increases the magnitude of variation and accordingly necessary storage to smooth it out. But in terms of residence time or the number of months of mean discharge, it decreases as the relative magnitude of variation to the mean discharge decreases with catchment area, similar to the specific flood peak discharge.

3) Both for flood and drought management, the necessary storage is small in the Himalayan high mountain areas where the discharge is stable with snow and glacier. On the contrary in north-eastern India approaching to the Himalayan areas in Fig. 10a, for flood, the very high months areas are concentrated, which reflects the semiarid climate with occasional floods, which makes flood control difficult. This is not visible for drought in Fig. 10b as multi-year fluctuation is not considered in this analysis. As stated in 1) above, the necessary storages for flood and drought control should become identical in a long run. The necessary storage for drought in this analysis therefore indicates the degree of difficulty of annual drought control. The difference between flood and drought is not very visible in km$^3$ in Fig. 9 again partly because of its logarithmic expression.

Such characteristics well correspond with SD and CV of Fig. 8b and 8c, which are the proof of Eq. 9 and 10.

4) The geographical distribution of necessary storage in km$^3$ in Fig. 9 and that of SD in Fig. 8b show a remarkable agreement and this is also the case in months in Fig. 10 and CV in Fig. 8c.

5) Nearly identical to Fig. 10a of the necessary storage in months, large CV areas are scattering in north-eastern India in Fig. 8c. But for drought, it is not visible as inter-annual smoothing is omitted.

Figure 11 is the same as Fig. 9 and 10 but with different target discharges, $3Q_{mean}$ for flood and $0.5Q_{mean}$ for drought management. Figures 11a and 11b are in km$^3$, and 11c and 11d in months.

6) The most remarkable feature of Fig. 11 is that there are many blank areas in the maps, which means that no storage is necessary as the natural discharge variation itself is within the $(0.5Q_{mean}, 3Q_{mean})$ range as long as 5-year return period discharges are concerned.

7) The blank areas of Fig. 11 well coincide with the blue areas (low necessary storage areas) of Fig. 9 and 10, both for flood and drought. They are mostly headwater areas especially in the high Himalayan mountain areas with snow. In case for drought the blank areas extend to the areas of low precipitation in the Western Gages as shown in Fig. 3b. The reason is simply low discharge variation in those areas which corresponds to relatively low CV in Fig. 8c. On the other hand, the areas with high CV are not blank, and especially for flood, the high red areas of Fig. 8c correspond to the red areas of Fig. 11c.

[ここに入力]

## 4.2 Comparison of necessary storage with catchment area and SD and CV of discharge in scatter diagram

Following the findings of spatial resemblance of geographical distributions of necessary storage and SD and CV of discharge, this section further looks into their relation by basins and catchment area using scatter diagrams. This is an initial trial to see if any extra information may be derivable from necessary storage signature that is different from SD and CV on hydrological characteristics of basins. Figure 12 shows the relation between necessary storage $V_{km3}$ [km$^3$] and catchment area A [km$^2$], and Fig. 13, the same on normalized necessary storage $V_{month}$ [months]. Figure 14 shows the relation between $V_{month}$ and SD [m$^3$/s] of daily discharge and Fig. 15, the same with CV. Each dot corresponds to each 10-arc-min grid cell in the basin and there are all together 5263 dots (Ganges 3220, Brahmaputra 1812 and Meghna 231) in Fig. 12-15.

There are two common observations to be explained.

1) In all Fig. 12-15, the dots are discontinuous. In Fig. 12 and 13, the reason is because the catchment area and discharge increase discontinuously where tributaries converge to the main river merging sub-basins of some size and flow. In Fig. 14, it is because the SD increases following the discontinuous increase of catchment area. In the case of Fig. 15, it is simply because each sub-basin has a different CV.

2) The other common observation is that all figures a) for flood and b) for drought are quite similar, which is due to the same reason mentioned in 4.1 1). At the same time, their levels are considerably different; that is, the necessary storage for drought control is smaller than that for flood control. This is because in this paper the necessary storage is calculated for short term intra-annual control (m≤365 days). In general, flood is huge within a year and no need to consider inter-annual flood control space to keep the long term mean discharge but in the case of drought, intra-annual control of drought may not require much storage to replenish but inter-annual drought (sometimes it extends over a decade) becomes predominantly large. By limiting the consideration into annual smoothing in this paper, the necessary storage for drought is considerably less than that for flood control in GBM basin where there is a distinct wet season and a dry season. The difference between the necessary storage for flood and drought would indicate the inter-annual dependency of drought which would be an interesting subject for future study.

Figure 12 $V_{km3}$-A relation indicates how necessary storage increases with catchment area. (1) necessary storage $V_{km3}$ increases generally in proportional to catchment area A as discharge variability increases with catchment area through increase in discharge volume. But (2) the slope is different in each basin which would reflect the overall hydrological variability of sub-basins merging into the main stream. The relative differences of slopes of three sub-basins of the GBM would be explained as follows:

3) The Meghna has a larger slope of necessary storage against catchment area than the other basins both for flood and drought, which must reflect large specific discharge with large variability of the Meghna.

4) The Brahmaputra has a distinct slope change from low to high at around 250,000 km$^2$ point which corresponds to the outlet from the Tibet. As Fig. 3b indicates, in upstream, precipitation is low with snow and glacier being dominant and in downstream, precipitation is high, making specific discharge and thus variability similar to the Meghna, which makes the slope of necessary storage markedly changes from low to high.

5) The Ganges has a rather low stable slope in the entire river length. This reflects rather stable river flow as a collection of many dry sub-basins in the East, mildly humid southern basins and ample but stable snow fed discharge in high mountains. The slope is as low as the Tibetan part of the Brahmaputra.

Figure 13 $V_{month}$-A relation has a distinct feature showing: (1) In the areas with smaller catchment A less than a few 10,000 km$^2$, the normalized necessary storage $V_{month}$ varies widely, but (2) as the catchment area becomes large, it converges to a stable line unique to each basin. (3) Those lines are not necessarily horizontal but either a decreasing or an ascending trend with catchment area. Such characteristics do reflect the characteristics of the basins and would be explained as follows:

6) Large scatter at the origin of catchment area must reflect many different grid cells having many different hydrological conditions including land use, land cover and other geographical and geological variability. Such detailed relation should

[ここに入力]

be examined based on detailed land cover information, which requires much finer resolution maps than Fig. 2 and 3, which is beyond the scope of this first step analysis and left for future study.

7) How large a basin should be to have a converging line or if there is no convergence in general would be a very interesting question on scale effect relating to hydrological heterogeneity of a basin. In Fig. 13 it seems a few 10,000 km$^2$ and no details can be said without more elaborate examination in many different basins in the world. In the next section 4.3, a further look of the GBM basin in sub-basin levels are presented.

8) The levels of converging lines at the river mouth seem roughly as follows:

For flood, $V_{month}$=6.2 months for the Ganges and 5.4 months for the Brahmaputra

For drought, $V_{month}$=5.5 months for the Ganges and 4.5 months for the Brahmaputra.

The Meghna case seems somewhere between the two but too short to observe the convergence.

Figures 13a for flood and 13b for drought are roughly in parallel although the levels are different. The difference in levels (namely, 6.2-5.5=0.7 months for the Ganges and 5.4-4.5 = 0.9 months for the Brahmaputra at the river mouth) reflects their difference in persistence characteristics of discharge variability. The Brahmaputra has more inter-annual variability for drought than the Ganges at the mouth, and limiting smoothing length by a year results in higher difference in necessary storage for flood and drought. Note the discussions in 4.1 1) and 4.2 2) on difference between $V_{month}$ for flood and drought.

9) On the slopes of the converging lines, the Ganges has a descending slope and the Brahmaputra an ascending slope. Namely, in the Ganges, it is $\frac{dV_{months}}{dA} < 0$ and in the Brahmaputra, it is $\frac{dV_{months}}{dA} > 0$. It should be related to their basin transition from upstream to downstream. In the Ganges, the river flows from the more storage (in months) needed western Ganges under semi-arid climate to the less storage needed eastern Ganges under more humid and stable climate (see Fig. 10).  In contrast in the Brahmaputra, it flows from the northern Brahmaputra with less storage-needed snow-affected region to the southern Brahmaputra with high precipitation without snow stabilizer.

Figure 14  $V_{km3}$-SD and Fig. 15 $V_{month}$-CV are the examination of indicated formula of Eq. 9 and 10 based on Eq. 8, the finding of Hurst (1951). From them the following are observable:

10) Figure 14 shows a remarkable proportionality between $V_{km3}$ and SD as indicated by Eq. 9 of Hurst (1951). The relation seems, by visual judgment, roughly the following:

$V_f$ (km$^3$) $\approx$ (1/70) SD (m$^3$/s)

$V_d$ (km$^3$) $\approx$ (1/80) SD (m$^3$/s)

Such relations, however, would be valid only along the large main streams of the GBM basin because the most points visible in Fig. 14 are along the large main streams having large discharge with a large SD. Most of the points in the figure concentrate near the origin with a small SD and a small V to which those formulas do not apply.

11) The slope of the V-SD relation seems a bit larger in the Meghna, then the Brahmaputra and the smallest, in the Ganges. This relation is similar in Fig. 12 A-V relation although shapes are different. This is because the relation between SD and catchment area A is roughly proportional although not linear with a constant coefficient. The coefficient of A-V changes distinctly where the Brahmaputra leaves the Tibet as indicated by Fig. 12 but not clearly visible in Fig. 14. This is because the SD-V relation does not change by discharge amount as Eq. 9 indicates.

12) Figure 15 $V_{month}$-CV relation shows widely scattered points distinctly different from Fig. 14$V_{km3}$-SD relation. This is because the variation of CV is independent of the magnitude of discharge and the size of the basin, while SD is roughly proportional to the magnitude of discharge and accordingly the size of the basin. As a result, the points representing smaller head basins are more visible in Fig. 15 and their hydrological heterogeneity is more prominently reflected in the location within the scatter. Note that in small head basins, hydrological heterogeneity more directly influences discharge variability than in large basins at their outlets. This heterogeneity will be more focused in the next section 4.3.

13) Figure 15 indicates the relation of Eq. 10 and the coefficients of average linear relations for flood and drought seem by "crude" visual judgement as follows:

[ここに入力]

$V_{month-f} \approx 5$ CV for the Meghna, 3.3 CV for the Ganges and 3.3 CV for the Brahmaputra

$V_{month-d} \approx 3.5$ CV for the Meghna, 2 CV for the Ganges and 2.5 CV for the Brahmaputra

The relative difference in coefficients would indicate the difficulty factor or the disadvantageous factor of water resources management with respect to necessary storage. A higher coefficient implies that with a given level of discharge variability in CV, one basin needs more storage to smooth out the variation than another basin. It should be related to the persistence structure of a hydrological process which is influenced by basin heterogeneity such as geophysical and geographical conditions including climatology, land cover and land use. Besides, the relative departure of a point from the average relation would also indicate the relative ease or difficulty of discharge control within the basin. Although we have no answer yet on what hydrological conditions control such a factor, it is a very interesting area of hydrological sciences to be studied in the future.

### 4.3 Effect of heterogeneity of basin hydrology on necessary storage

As found in the analyses of 4.1 and 4.2, the slope of the linear relationship between necessary storage V and A, SD and CV, and the scatter of deviations of V from its average relationship vary by basin depending on its characteristics of hydrological conditions such as elevation, topography, geology, precipitation and land cover including vegetation, soil and land use of the basin, some of which are shown in Fig. 2 and 3. Such variation is the source of extra information on hydrological heterogeneity of a basin that necessary storage has but traditional SD and CV do not. This section therefore looks into details of slopes and scatter of $V_{month}$-A relation in Fig. 13 and $V_{month}$-CV in Fig. 15 by their zoom-ups, Fig. 16 and 17, respectively in selected 12 sub-basins indicated in the map in the center. They are selected to cover different areas of hydrological condition.

Note that the zoom-ups of V-A relation in Fig. 12 and V-SD in Fig. 14 were not examined as they are basically linear as Hurst (1951) showed in Eq. 9 and strong linearity of catchment area A with SD.

Figure 16 is the zoom up of Fig. 13 on the A-$V_{month}$ relation of each basin. The following are observable:

1) Similar to Fig. 13, $V_{month}$ for flood (blue) is larger than that for drought (red) in general although they are mixed near the origin (head basins). The reasons were explained in 4.1 1) and 4.2 2).

2) There is a wide scatter of $V_{month}$ near the origin which eventually converges to near horizontal lines around 6 months as catchment becomes larger. The converging lines are not necessarily just one from the beginning but a few representing branch rivers before they merge into the main stream of sub-basins. The total shape may be roughly described as a horizontally laid bell shape which is similar to Fig. 13 and explained in 4.2 6)-9).

3) The upper six basins of Ganges, Yamuna, Ganga, Ghanhara, Gandak and Kosi and a western Brahmaputra basin, Teesta have upward converging lines. These basins originate from the Himalayan mountains receiving ample snowmelt water and flowing down to the south. Their form is considerably distorted from Fig. 13 with wide scattering $V_{month}$ near the origin, followed by very low $V_{month}$ especially for drought and a gradual increase with the catchment area. A large scatter is due to wide heterogeneity of the basin due to large catchment and varieties of climatic conditions in the basin. Small $V_{month}$ near the origin may be attributed to a dominant snowmelt influence with little variability as the dark blue areas in Fig. 8b SD and 8c CV show. But $V_{month}$ gradually increases as the main river flows down to low land in the South, receiving ample rainfall with high variability as shown in Fig. 8.

4) Two basins Subansiri and Lohit in the Brahmaputra are small and less heterogeneous (rather uniform conditions) with snowmelt influence and quickly converge to the average.

5) The basins along the southern edges of the Ganges and the Meghna to the southern slopes of the Brahmaputra, Chambal, Betwa, Son, Barak have a horizontally laid bell shape similar to Fig. 13. This may be because they receive no snow fall and the river flow increases rather proportionally with catchment area, and so does necessary storage that results in a quick convergence to a basin average although the headwater basins vary considerably in $V_{month}$. The Chambal is a bit unique in that the $V_{month}$ for flood and drought are the same in the main stream over $5 \times 10^5$ km$^2$ and the converging line is slightly

[ここに入力]

downward. This may reflect large variability upstream by the influence of semi-arid climate in the West and rather stable humid climate in the East as Fig. 3b indicates. Similarity for flood and drought is due to similarity of their inter-annual variability which needs further study in the future.

6) The average catchment area (scale) where the A-$V_{month}$ relation converges to a near horizontal line seems a few 10,000 km$^2$ in most sub-basins. The hydrological mechanism to determine such critical scale would be an interesting subject of further study as discussed in 4.4.2.

Figure 17a is the zoom up of Fig. 15 on the $V_{month}$-CV relation.  Here diagrams of the numerical summary of Fig. 2 and 3 are also attached for easy reference, i.e., elevation (H), precipitation (P) and land cover (L) distributions of the basin. Figure 17b shows the legend of the summary diagrams. Figure 17a indicates:

7) Again, the normalized necessary storage is larger for flood than for drought in general with considerable mixture in wide range of CV. The wide range is due to small catchments (head water basins) scattered in CV

8) The H-P-L distribution diagrams indicate, on elevation (H), over 2000 m area is more than 30 % in sub-basins in the northern Ganges to the Brahmaputra basins, including Gandak, Kosi, Teesta, Subansiri and Lohit. The flat or low land prevails in the sub-basins in the western to the southern Ganges to southern Meghna, Ganga, Yamuna, Chambal, Betwa, Son and Barak. On precipitation (P), dry basins with less than 500-1000 mm are dominant in Ganga, Yamuna, Chambal and Betwa; on the other hand, basins with over 2000 mm/y prevails in in the Meghna to the Brahmaputra, Barak, Rohit, and Subansiri. On land cover (L), the salient feature is that crop land prevails in Ganga, Yamuna, Chambal, Betwa and Son, and that forests prevail in Barak, Lohit and Subansiri and Teesta.

9) Six sub-basins in the western to the northern Ganges, from the Chambal to the Kosi, have wide ranges of CV, 0-6 or 0-8. They are relatively large sub-basins covering heterogeneous regions. On the other hand, the other six located in the Brahmaputra and the southern edges of the Ganges and the Meghna show a rather concentrated cluster of points or a narrow range of CV. This would be because sub-basins are smaller, compared with the upper six and although the H-P-L variability seems large as the summary diagram indicates, discharge variability is kept small in CV. The reasons may be in the dominant effects of hydrology, partly explained in the following 10) and 11)..

10) The scatter is especially large in the Yamuna and the Ganga followed by the Chambal and the Ghaghara which, as Fig. 3a indicates, except the Chambal, originate from the high Himalayan Mountains and flow down to the highland plateau and eventually to the lowland plain. In the plateau and the plain of the Chambal, Yamuna and Ganga, it is rather dry or semi-arid climate, and, as the summary diagrams of Fig. 2 and 17a show, it is used for extended cropland, which makes discharge variability high through evapotranspiration. Such heterogeneity in land use contributes for the large scatter of $V_{month}$.

11) On the other hand, the scatter is especially small in three Brahmaputra sub-basins the Teesta, the Subansiri and the Lohit, which start from strong snowmelt effect in high Himalayan mountains and flow down to one of the thickest forest regions in the world where discharge variability is small.

12) In relation to Eq. 10, the average line of the points (corresponding to each grid-cell of the sub-basin) is expected to pass the origin. It is the case in Fig. 15 but not in Fig. 17a. The height of each point ($V_{month}$ signature) still indicates the relative difficulty of discharge control among the points having the same CV but the slope of average lines of a basin discussed in 4.2 13) is not clearly visible in Fig. 17a. The reason or the implication of Eq. 10 would be another interesting subject for further study.

## 4.4 Some other thoughts on future investigation

Although a number of unknown areas are mentioned for further analyses in discussions above, some more important areas are briefly described below.

[ここに入力]

### 4.4.1 Potential use of spatially distributed necessary storage information for water resources management

In this paper, necessary storage is proposed as a signature of hydrological variability in time. Its advantage over other statistical indicators is obvious as it is in human terms and has a direct implication of the ease and difficulty of water resources management. But what about its spatially distributed information? Does this have any use to water resource managers? It is a difficult question and the next step of this analysis. But some potential areas of investigation may be indicated. One would be an implication of spatial differences of necessary storage in months in independent sub-basins that may have a potential benefit of water transfer. It may be beneficial to transfer water stored in an area with a shorter mean refilling time (months) to another area with a longer refilling time (months). Another would be the relation to land use for agriculture, that is, the area with smaller necessary storage may indicate relative advantage for water demanding vegetation. Gao et al. (2014) looked into the terrestrial ecosystem's root zone moisture capacity at the catchment scale and found it equivalent to the necessary storage for a 10-40 year drought. If the necessary storage is small, it would need smaller root zone capacity, and larger if not, which may indicate some suited agricultural or forestry land use. The other potential area would be the investigation of impact on downstream when a reservoir was built or to be operated. That would necessitate study on longitudinal changes in necessary storage along river lines, which may indicate an advantageous site of dam construction or operation in a hydrological sense. Reservoir construction sites or reservoir operation which has less impact on downstream areas would be desirable from an environmental point of view.

### 4.4.2 Scale effect of normalized necessary storage in months

Another potential area of further study is the scale effect of normalized necessary storage in months, i.e., the $V_{month}$-A relation. Figure 13 is its large basin behaviour and Fig. 16 is its zoom up in the selected 12 sub-basins. All figures indicate that in a few 10,000 km$^2$ catchment area, the normalized necessary storage becomes stable unless different large branch rivers join with different characteristic months. This may have a conceptual analogy similar to the discussion of representative elementary area (REA) concept (Wood et al., 1988) based on the finding that the variability of hydrological processes becomes low once the area becomes larger than around 1 km$^2$. Hydrological variability in storage domain and its governing mechanism would deserve further attention.

### 5.    Conclusions

This paper introduced necessary storage to smooth out discharge variation to meet a given target as a signature of discharge variability in time and presented it in a geographical distribution to analyse its relation with hydrological including geophysical and geo-graphical heterogeneity of a basin. The signature of necessary storage has a distinct advantage over other indicators of variability such as SD and CV as it is expressed in concrete human terms and integrated with persistence characteristics of variability. This paper showed that the scatter pattern of departure of signature from its average relationship with parameters such as A, SD and CV indicated characteristics of hydrological heterogeneity of a basin. Analysis of departure by extending it to global maps would lead to new scientific understanding of hydrological heterogeneity.

The case presented here was with the target to maintain long-term mean discharge $Q_{mean}$ in 5-year return period. But such target level can be chosen arbitrarily such as $3Q_{mean}$ for flood and $0.5Q_{mean}$ for drought (Fig. 11) and in different return periods or rates of failure. The signature was calculated from intensity-duration-frequency curves of daily discharge called flood duration curves (FDC) and drought duration curves (DDC) to ease the calculation of necessary storage at all grid points of a basin and plotted to geographical maps for the Ganges-Brahmaputra-Meghna (GBM) basin. In addition to analyses of the basin as a whole, 12 selected sub-basins were focused and tried to relate the necessary storage signature with their hydrological characteristics. Although this paper showed only the first trial of the analysis of signature, we believe that the potential use of it and the validity of the methodology were demonstrated and the following would be concluded:

[ここに入力]

1) The necessary storage serves as a signature of discharge variability in time and its geographical distribution provides a means of analysing hydrological heterogeneity.

2) The necessary storage signature has a distinct advantage to measure hydrological variability over other conventional statistical indicators such as SD and CV, as it has human terms "reservoir storage in $km^3$ or mean refilling time in months" which directly indicates the ease and difficulty of flood and drought management. In addition, it is an integrated indicator including the effects of persistence characteristics of variability which SD and CV do not have.

3) Necessary storage is a signature that has potential to show something extra to statistical variability indicators such as SD and CV. The analyses of the scatter pattern of departures from the normal relations of V-A, V-SD and V-CV would lead hydrological sciences to new insights on geographical distribution of hydrological heterogeneity.

4) To calculate necessary storage by intensity-duration-frequency curves, FDC and DDC, is a practical and convenient way that is beneficial to extend the study to the globe.

5) The signature in $km^3$ generally becomes larger with increase in catchment area so that the major river routes emerge out as blood vessels with larger values than the surrounding. The signature in months, normalized by the local long-term mean, however, converges to smaller values than the surrounding points representing smaller catchment as river discharge is stabilized as a catchment becomes larger.

6) In the headwater areas where river routes do not distinctly emerge as catchments are small and concentration effect is not dominant, the signature both in $km^3$ and months reflects much on hydro-meteorological and other geophysical and geographical heterogeneity of a catchment. The initial observations of departures from the average relationship with A and CV seem to indicate that necessary storage both for flood and drought management is small in snow-and-glacier-affected areas and large in lower plains under both dry and humid climate reflecting their degree of variability. In addition, necessary storage is small in thick forests and large in heavy cropland.

7) In main streams, on the other hand, the variation converges to the average basin characteristics towards the river mouth but whether the converging trend is descending, ascending or flat depends on basin characteristics. Such unique scale effects in different basins would be an important area of investigation to be related to hydrological heterogeneity of a basin. It seems to have some similarity to the concept of representative elementary area (REA) introduced by Eric Wood and his colleagues (Wood et al., 1988) on scale effects on spatial heterogeneity of hydrological processes.

8) A creation of global maps of necessary storage would be effective for analyses of such hydrological heterogeneity in storage domain and a useful challenge for assessment of the current state of water resources and climate change impact on water resources.

**Acknowledgements**

We express our deepest gratitude to Professor Eric Wood, Princeton University, for giving us an inspiring opportunity to present the results of this research in his honour symposium, and to Professor Murugesu Sivapalan, University of Illinois, for his kind encouragement and invaluable suggestions throughout the presentation and publication processes. Our gratitude extends also to the anonymous reviewers who provided many valuable comments.

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

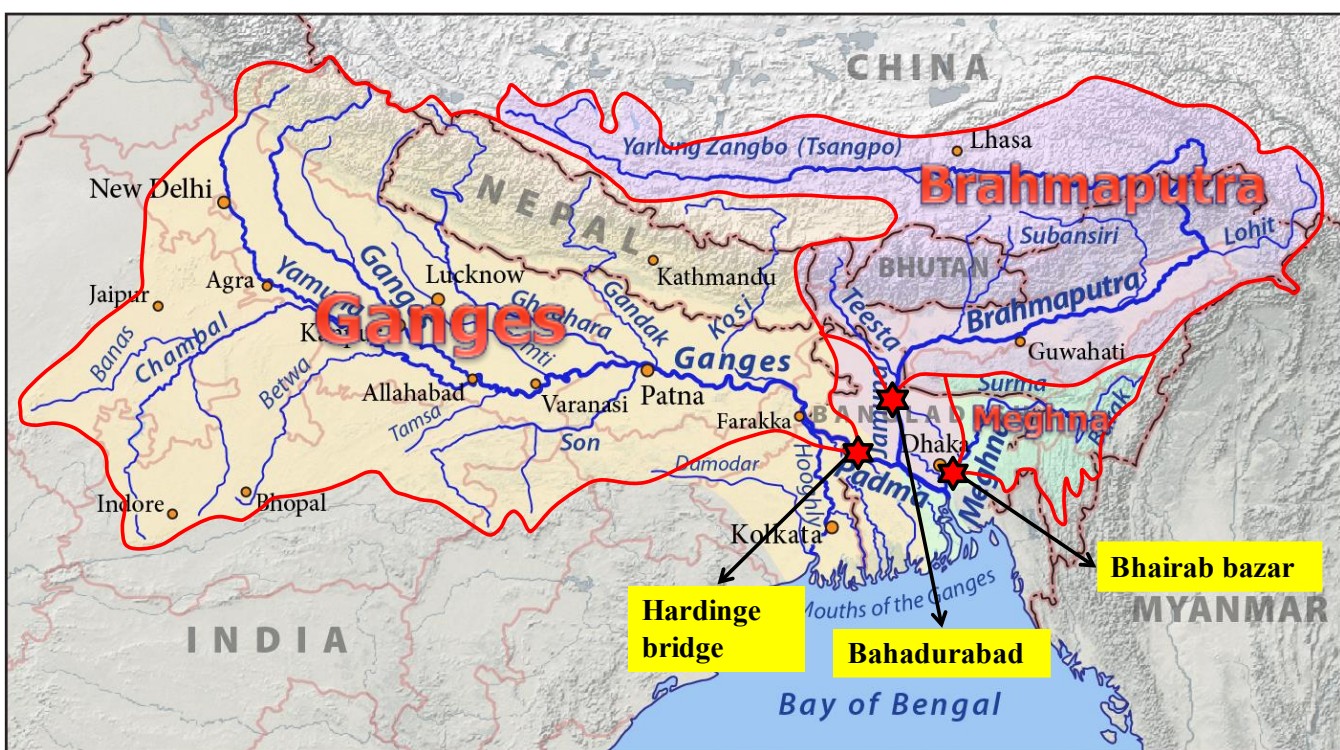

**Figure 1: The Ganges-Brahmaputra-Meghna (GBM) basin (modified from Pfly, 2011).**

[ここに入力]

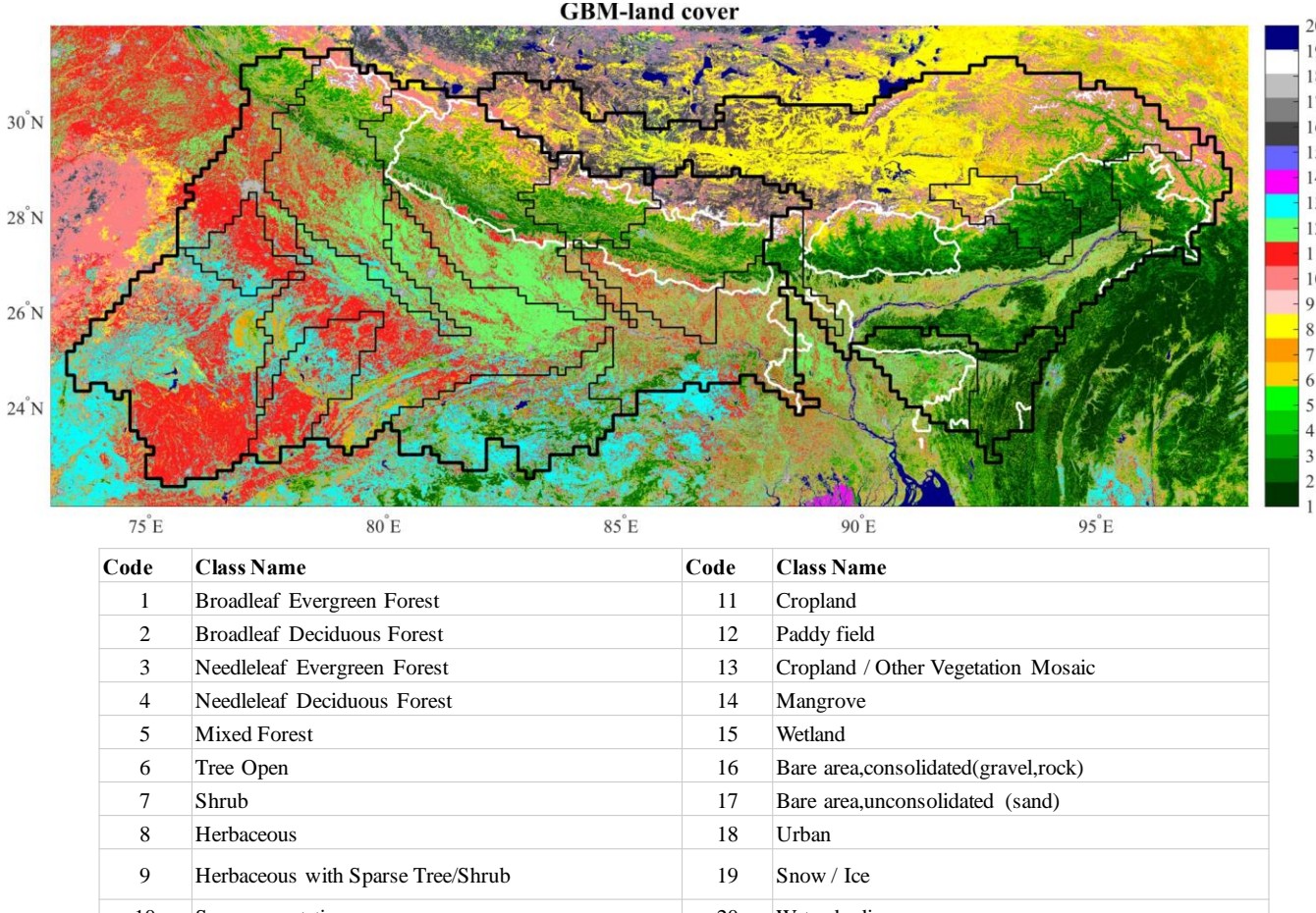

| Code | Class Name | Code | Class Name |
|------|-----------|------|-----------|
| 1 | Broadleaf Evergreen Forest | 11 | Cropland |
| 2 | Broadleaf Deciduous Forest | 12 | Paddy field |
| 3 | Needleleaf Evergreen Forest | 13 | Cropland / Other Vegetation Mosaic |
| 4 | Needleleaf Deciduous Forest | 14 | Mangrove |
| 5 | Mixed Forest | 15 | Wetland |
| 6 | Tree Open | 16 | Bare area,consolidated(gravel,rock) |
| 7 | Shrub | 17 | Bare area,unconsolidated (sand) |
| 8 | Herbaceous | 18 | Urban |
| 9 | Herbaceous with Sparse Tree/Shrub | 19 | Snow / Ice |
| 10 | Sparse vegetation | 20 | Water bodies |

**Figure 2: Land cover of the GBM basin from GLCNMO (Tateishi et al, 2014).**

[ここに入力]

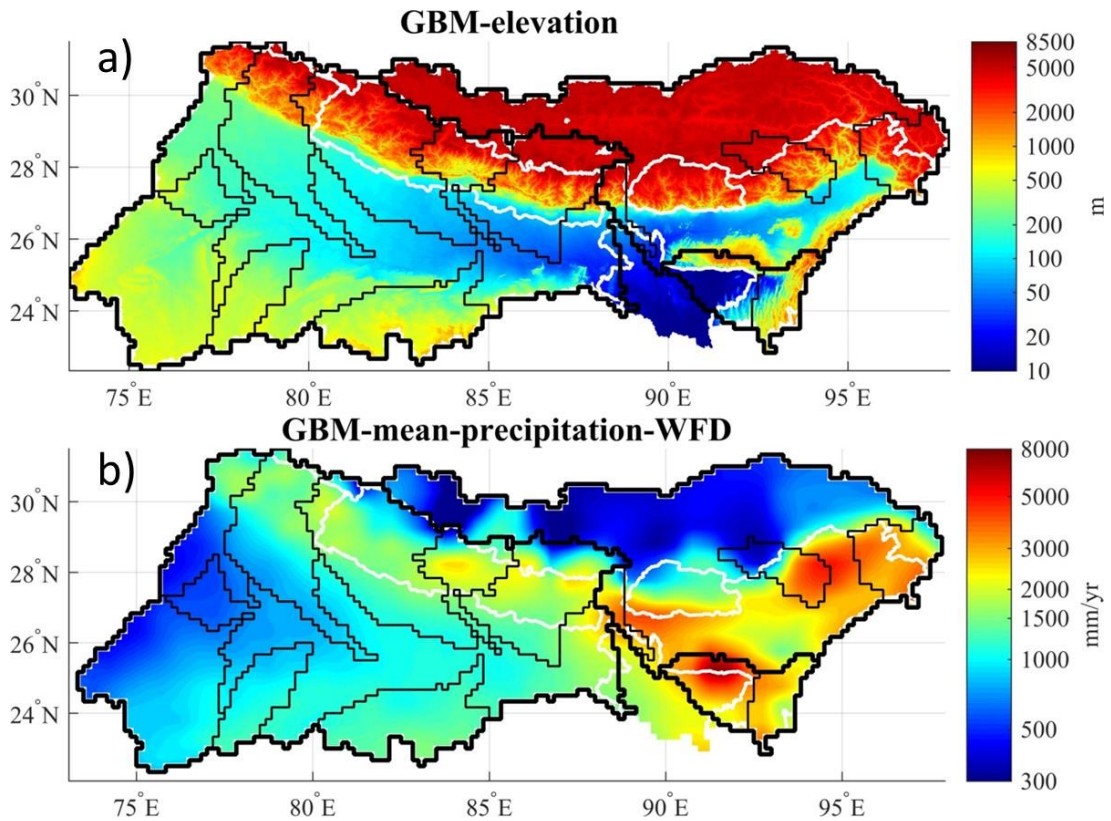

Figure 3: The a) elevation and b) precipitation distributions of the GBM basin.

[ここに入力]

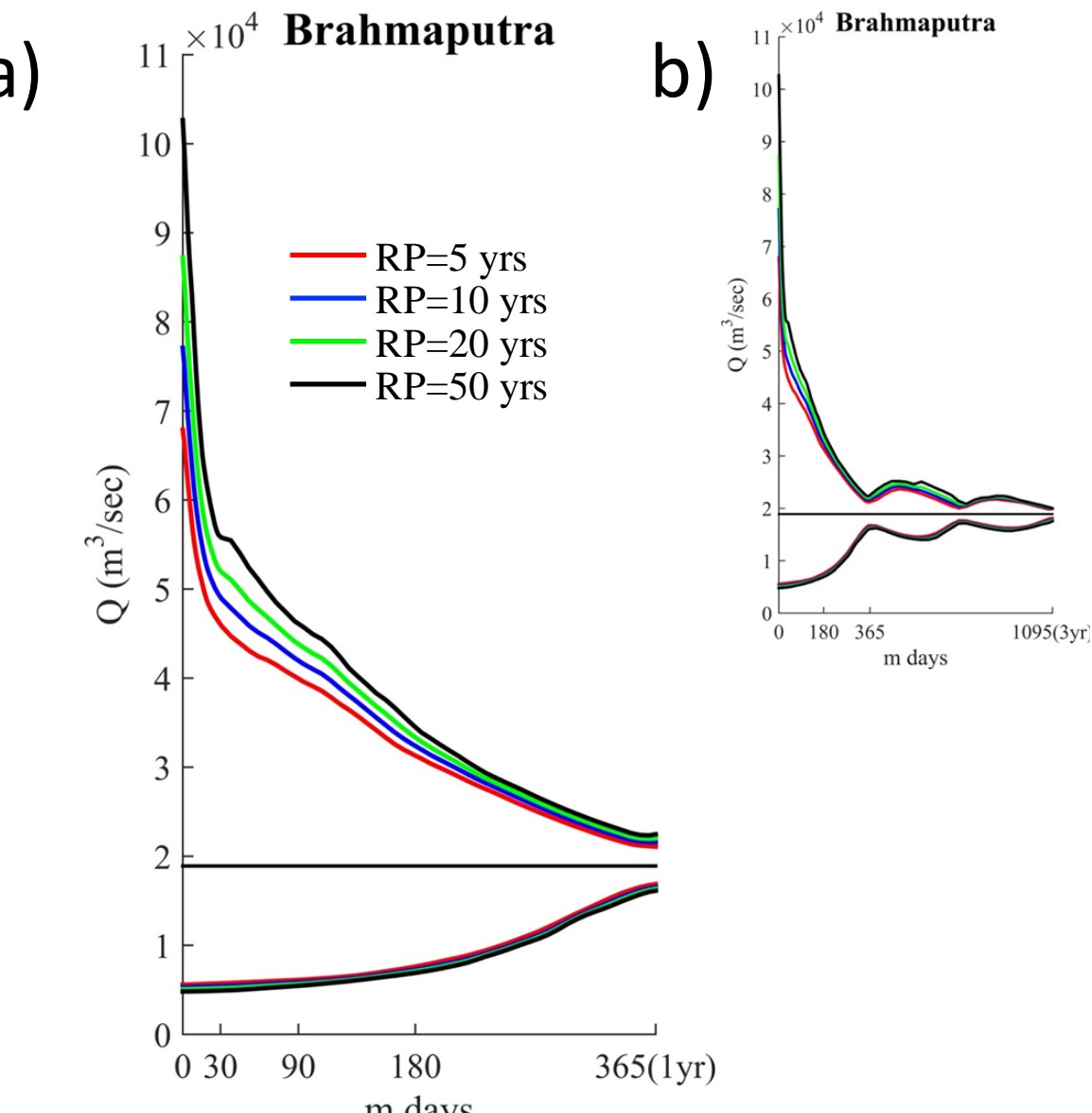

**Figure 4: FDC-DDCs of the Brahmaputra River at Bahadurabad station a) for one year and b) for three years. The discharge data were simulated by BTOPMC with WFD data.**

[ここに入力]

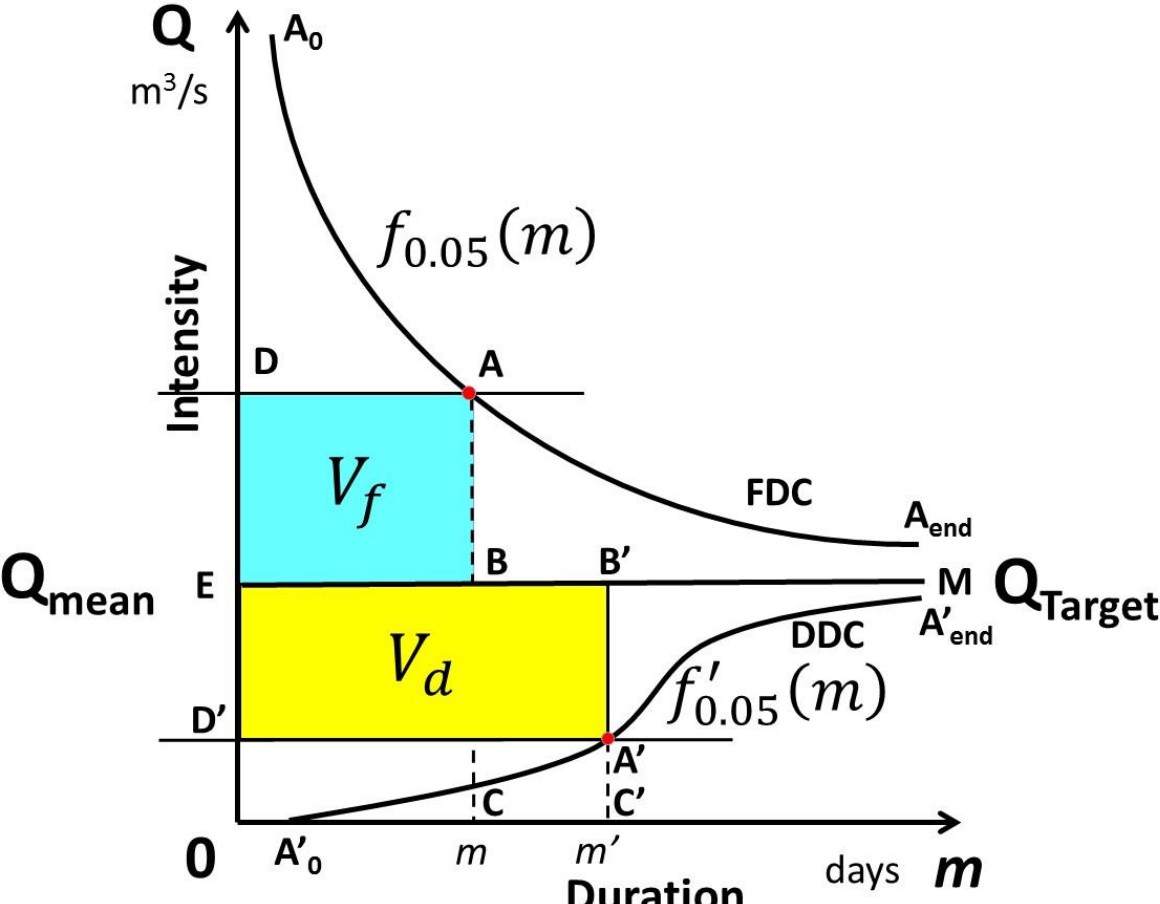

**Figure 5: Schematic illustration of how to calculate necessary storages V$_f$ and V$_d$ to smooth out the hydrological variation to the long term mean Q$_{mean}$ for flood control and drought management.**

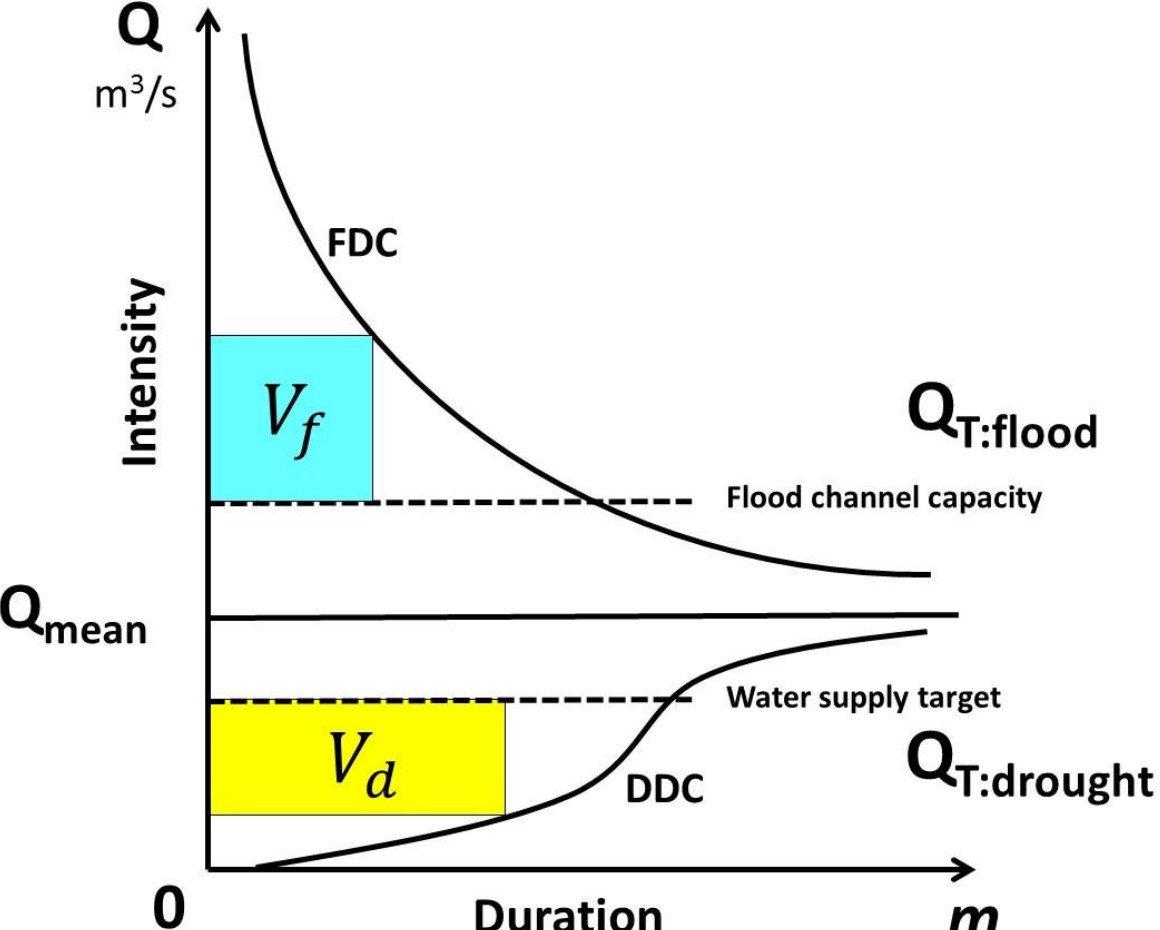

**Figure 6: Schematic illustration of how to calculate necessary storages V$_f$ and V$_d$ to smooth out the hydrological variation to 3Q$_{mean}$ for flood control and 0.5Q$_{mean}$ for drought management.**

[ここに入力]

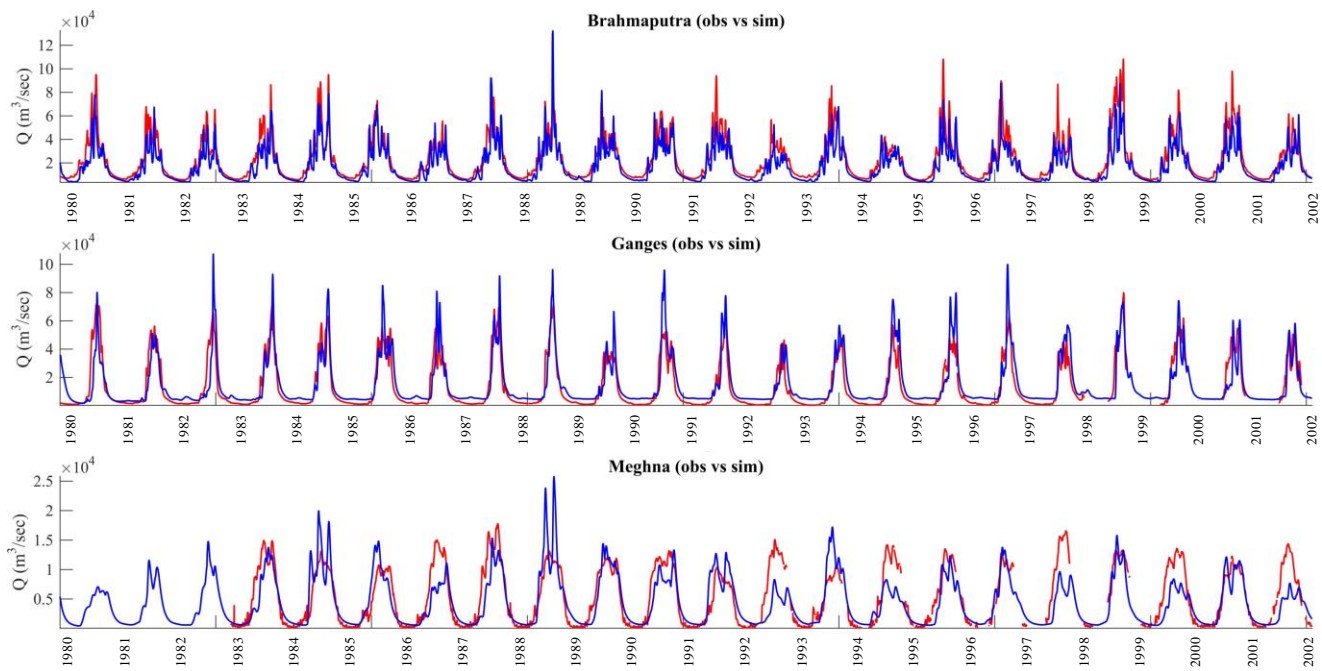

**Figure 7: Comparisons of the observed (red) and the BTOPMC simulated (blue) discharges at (top) Bahadurabad, the Brahmaputra, (middle) Hardinge Bridge, the Ganges and (bottom) Bhairab Bazar, the Meghna. Calibration period is 1980-1990. Precipitation data used are WATCH Forcing Data (WFD). See Table 1 for Nash-Sutcliffe efficiency.**

[ここに入力]

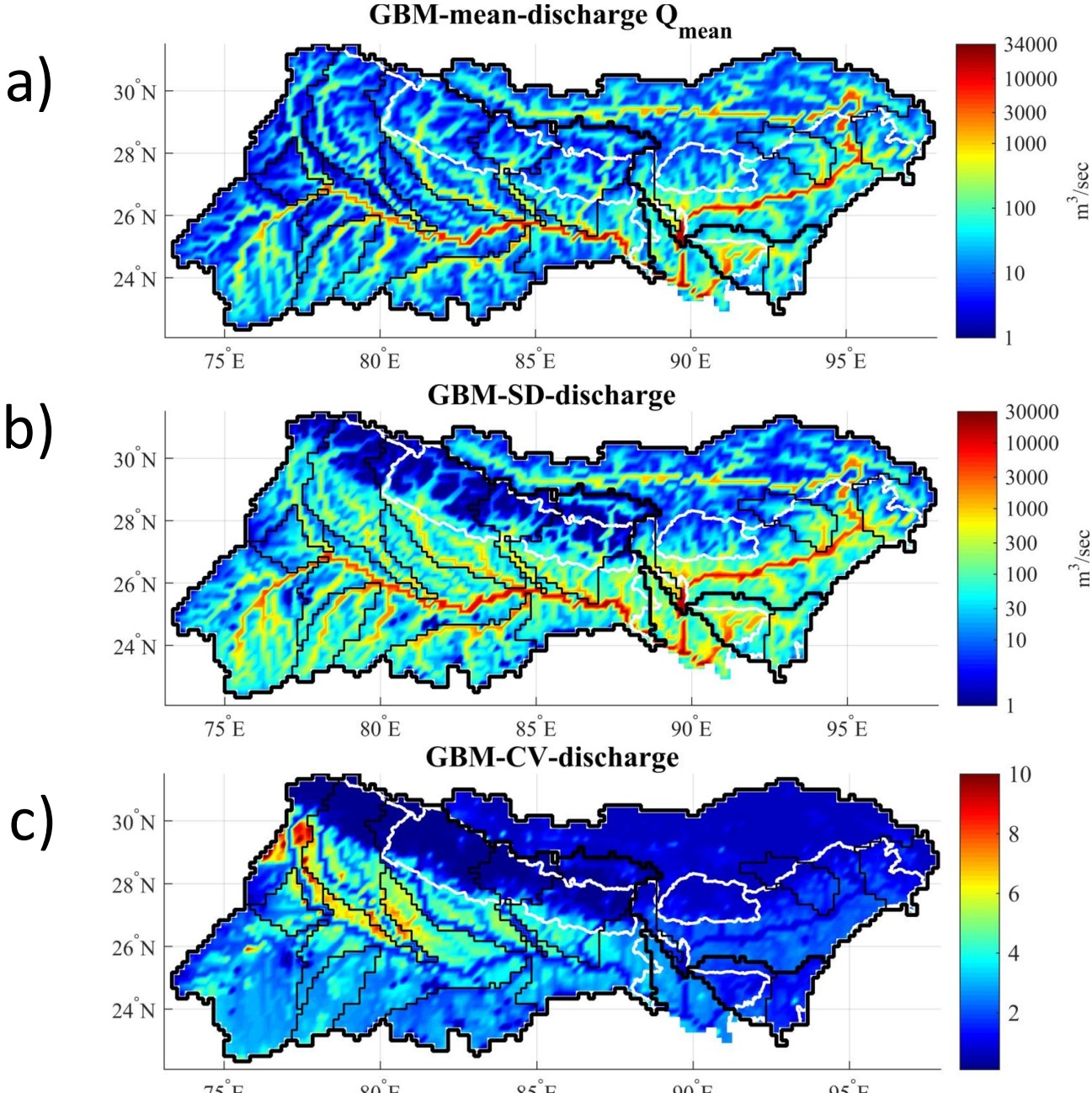

**Figure 8: Maps of a) the mean, b) SD and c) CV of the simulated discharge for 1980-2001.**

[ここに入力]

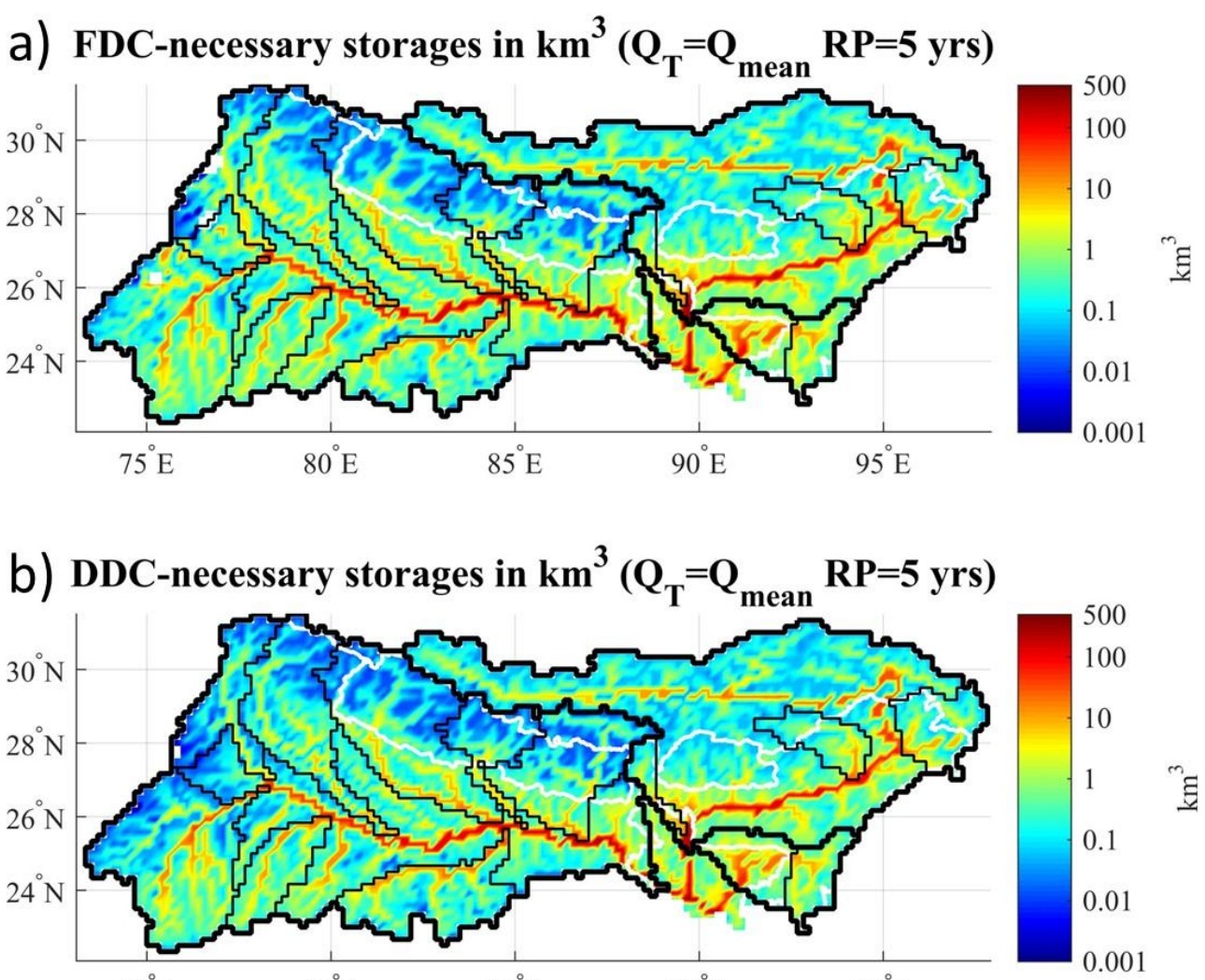

**Figure 9: Necessary storages (km³) for a) flood and b) drought management with target discharge $Q_T=Q_{mean}$ of return period 5 years. The simulated discharge was for 1980-2001.**

[ここに入力]

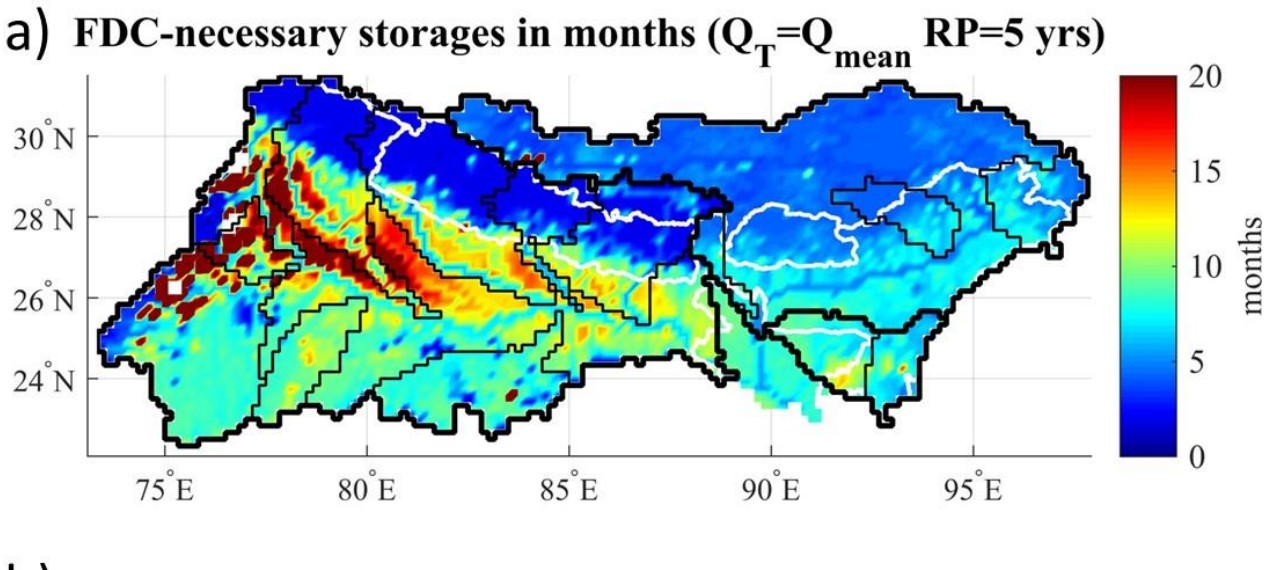

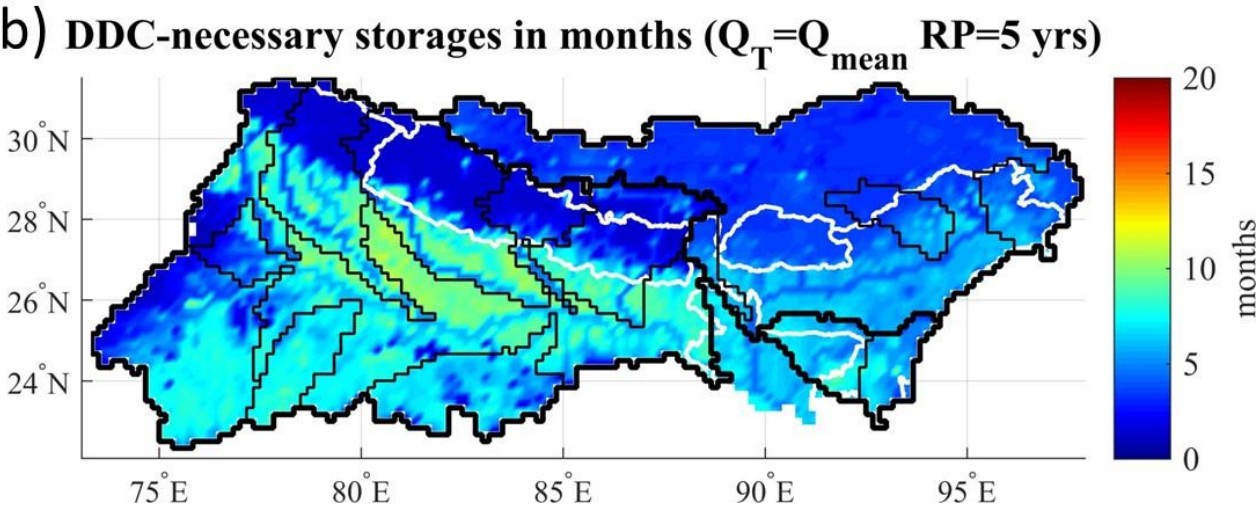

**Figure 10: Necessary storage in months for a) flood and b) drought management with maintaining discharge $Q_T=Q_{mean}$ of return period 5 years. The simulated discharge was for 1980-2001.**

[ここに入力]

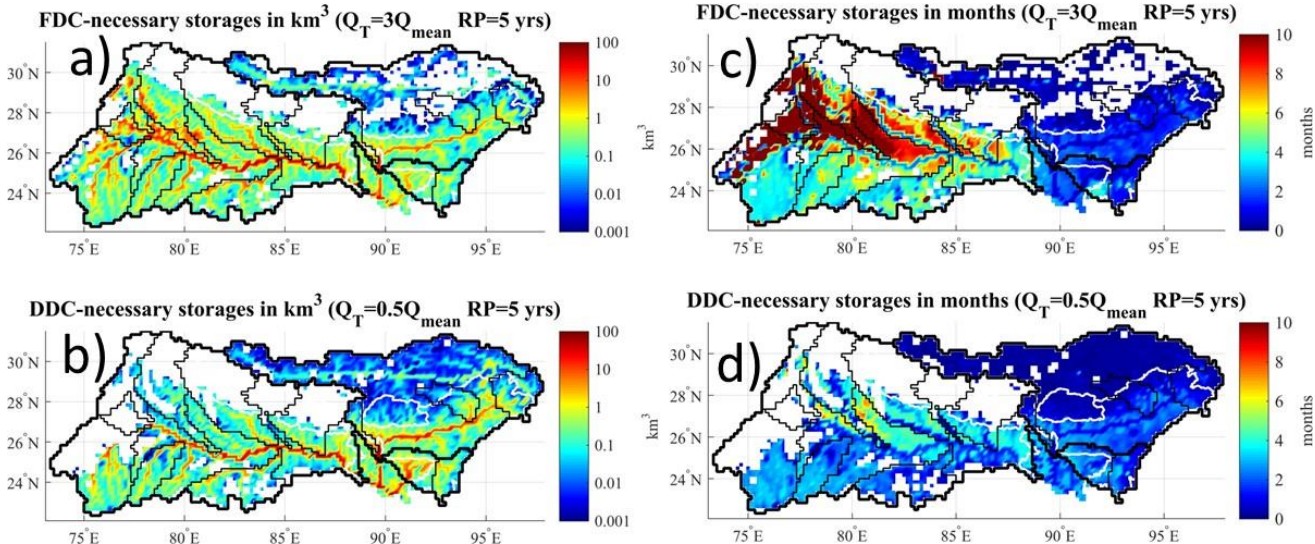

**Figure 11: Necessary storages (a, b) in km³ and (c, d) in months (a, c) for flood and (b, d) for drought management to maintain $Q_T=3Q_{mean}$ during flood and $Q_T=0.5Q_{mean}$ during drought of return period 5 years. The simulated discharge was for 1980-2001.**

[ここに入力]

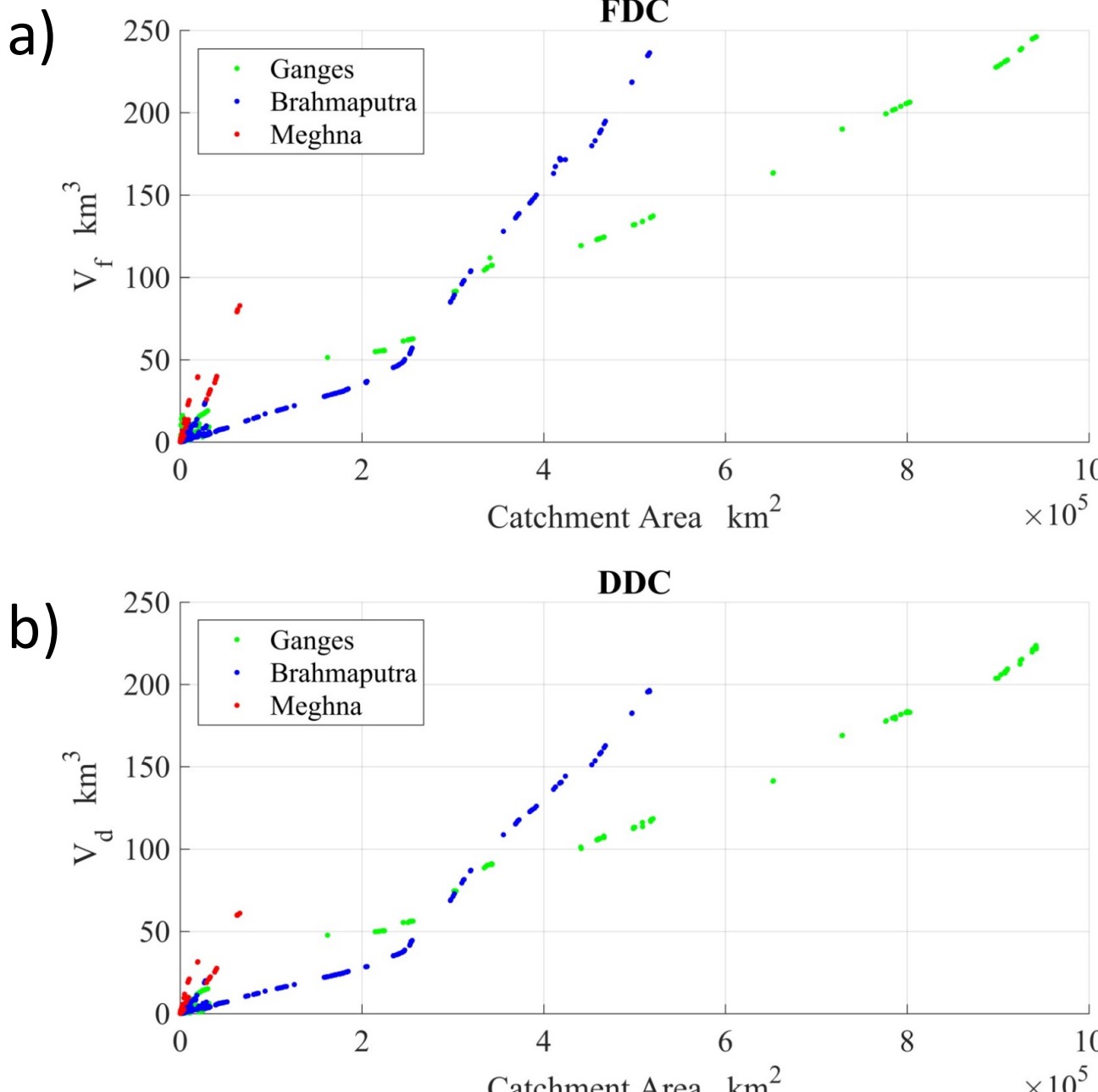

**Figure 12: Relation between necessary storage (km³) and catchment area (km²) for a) flood and b) drought management in three basins of the GBM basin.**

[ここに入力]

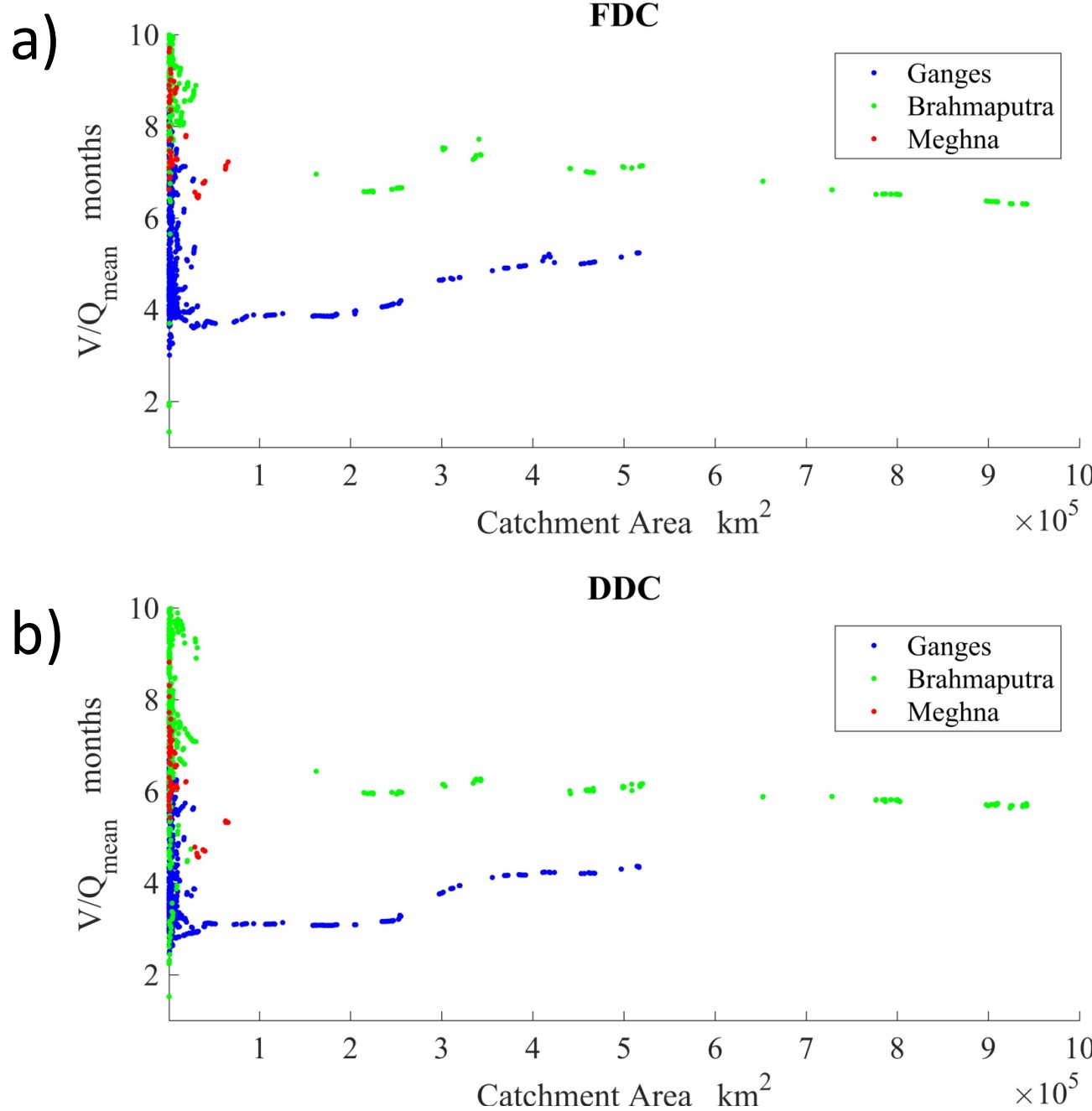

**Figure 13: Relation between necessary storage in months and catchment area (km$^2$) at all grid points of the basin with maintaining discharge Q$_T$=Q$_{mean}$ for a) flood and b) drought management with 5 years return period**

[ここに入力]

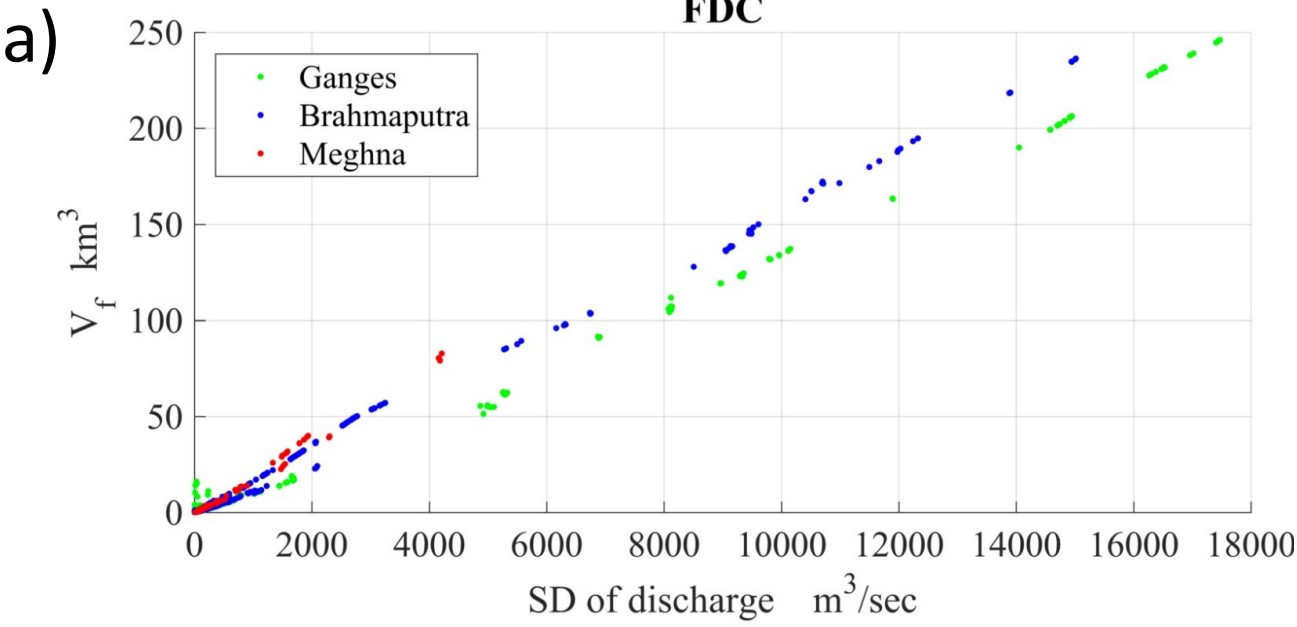

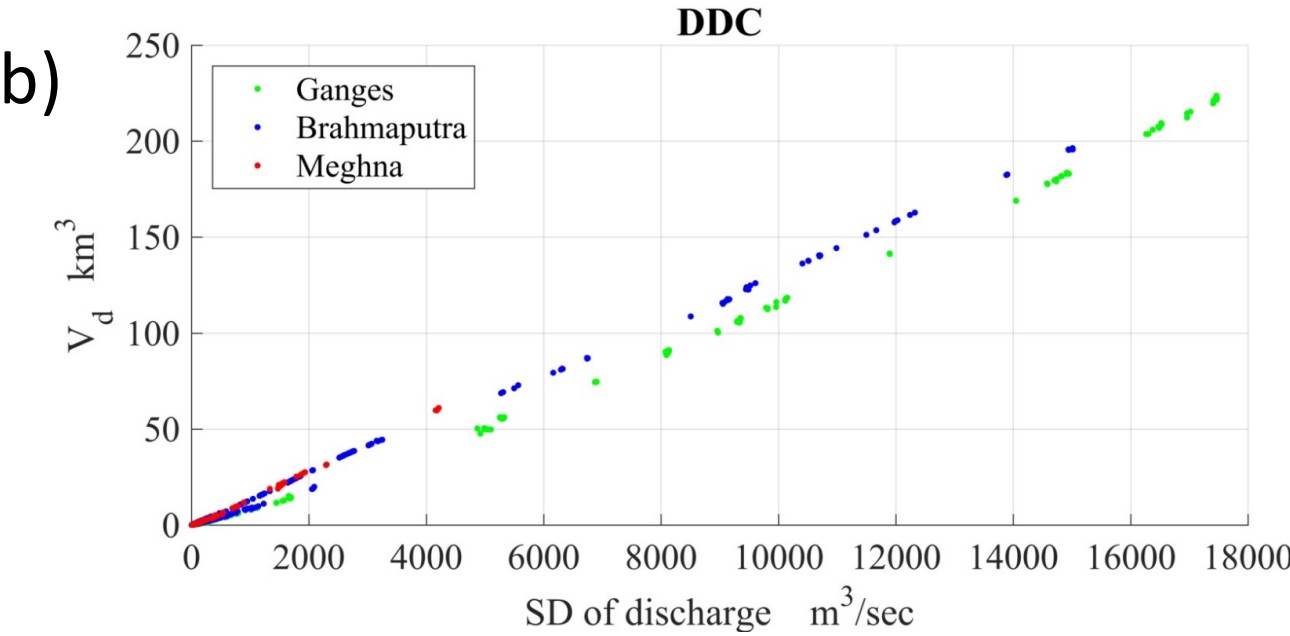

**Figure 14: Relation between necessary storage (km³) and SD (m³/s) of discharge at all grid points in the GBM basin for a) flood and b) drought management.**

[ここに入力]

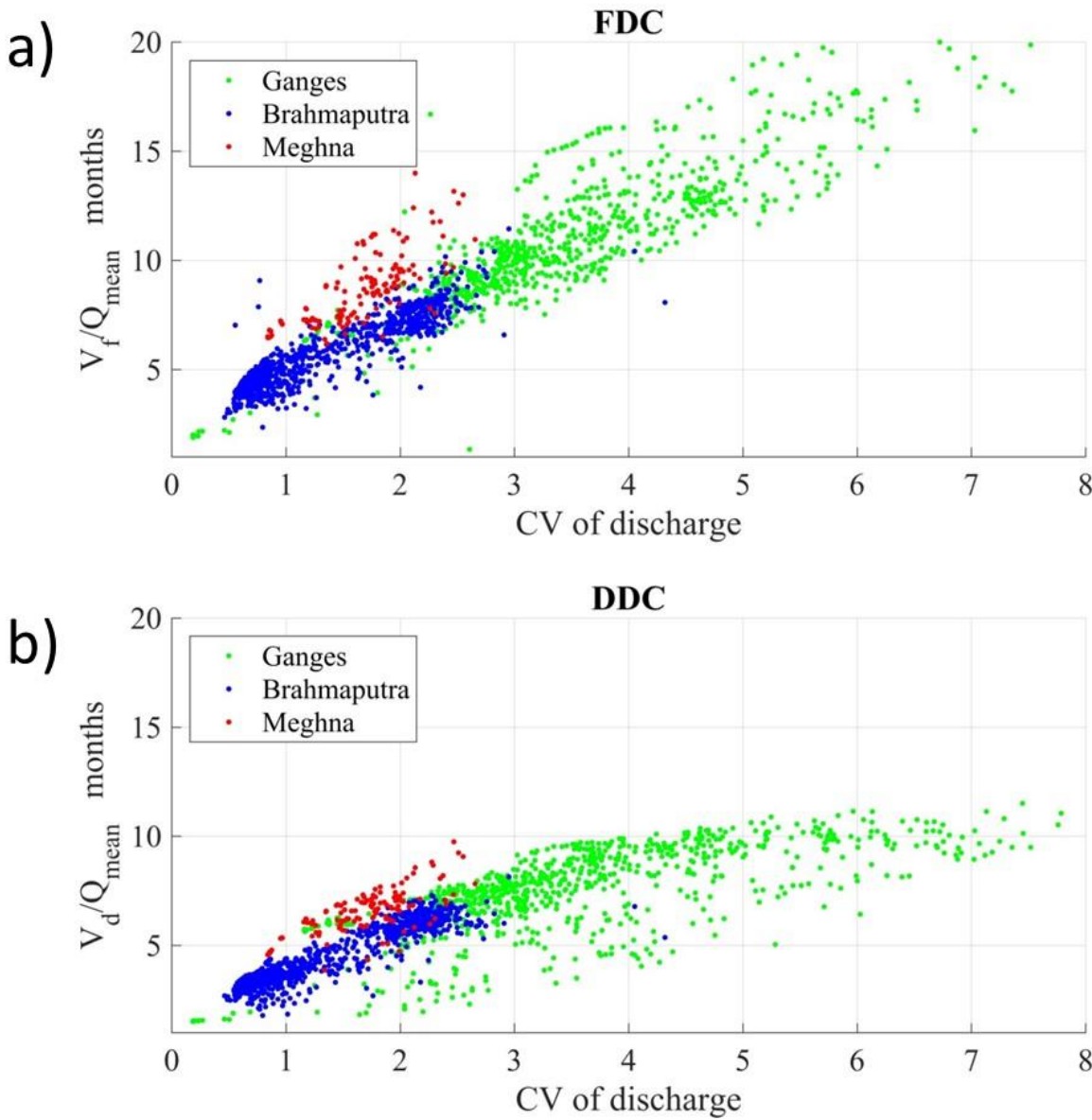

**Figure 15: Relation between necessary storage in months and CV of daily discharge of 5-year return period for a) flood and b) drought management at all grid points of the GBM basin.**

[ここに入力]

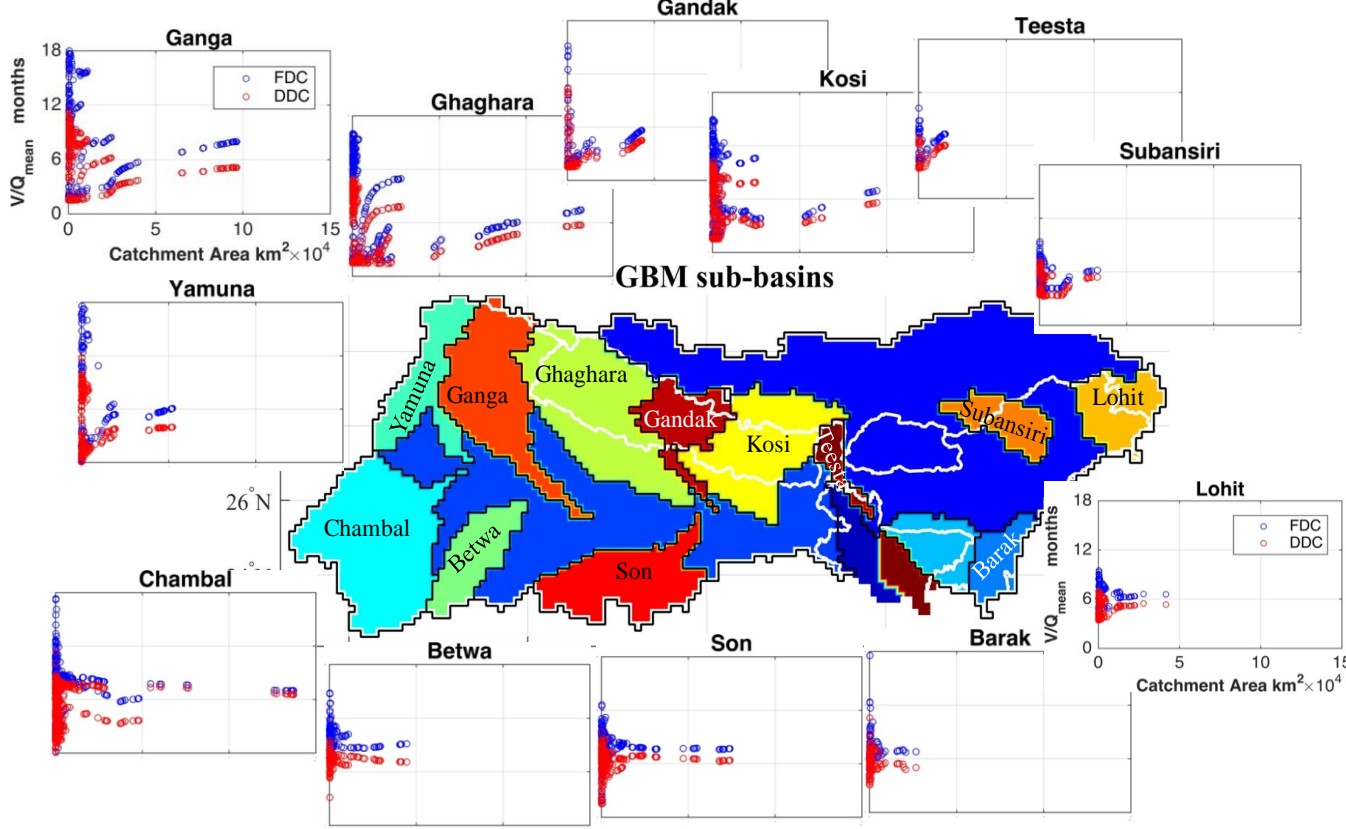

**Figure 16: Zoom-ups of Fig. 13 at the selected 12 sub-basins of the GBM, i.e., relations between necessary storage in months for flood (blue) and drought (red) management and catchment area (km²) at all grid points of GBM sub-basins as indicated in the center with maintaining discharge $Q_T=Q_{mean}$ during discharge with 5 years return period. The simulated discharge was for 1980-2001.**

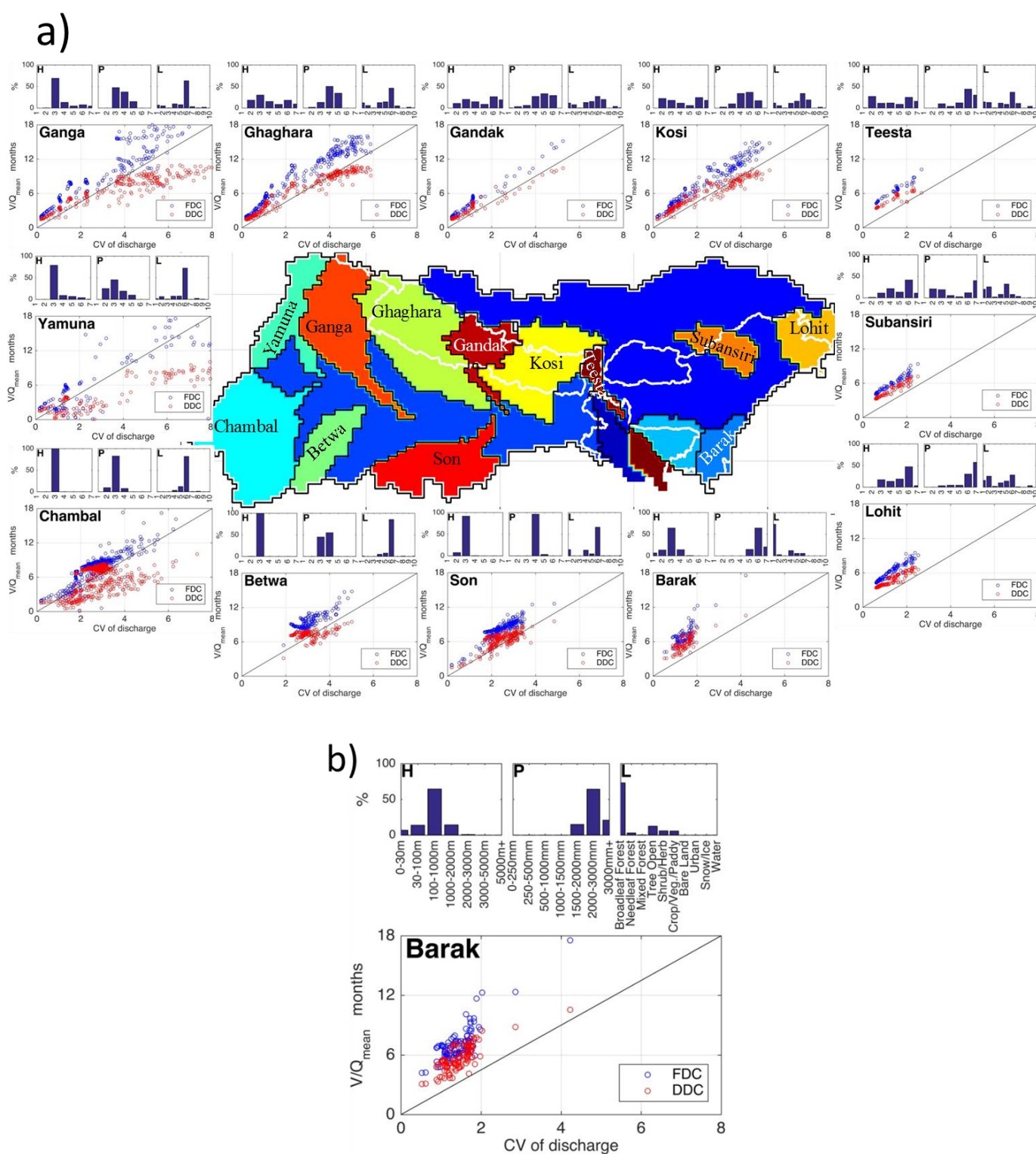

**Figure 17: a) Zoom-ups of Fig. 15 at selected 12 sub-basins of the GBM with summary of elevation (H), precipitation (P) and land cover (L) distribution, i.e., relation between necessary storage in months and CV of daily discharge for flood (blue) and drought (red) management at all grid points of sub-basins as indicated in the center with maintaining discharge $Q_T=Q_{mean}$ during with 5 years return period. The simulated discharge was for 1980-2001. b) Legend of**
 **diagrams in a).**

[ここに入力]

**Table 1: BTOPMC sensitive parameters and their optimal parameter values with simulation performance.**

| Name of parameter | Drying function parameter ($\alpha$) | Decay factor ($m$) | Block average Manning's roughness coefficient ($n_0$) | |
|---|---|---|---|---|
| Unit | - | meter | s/m$^{-1/3}$ | |
| Value range | -10 ~ 10 | 0.01 ~ 0.1 | 0.01 ~ 0.8 | (Takeuchi *et al.*, 2008) |
| **Basin** | **Best Parameter values obtained from parameter-sampling simulation** | | | **NSE (Nash-Sutcliffe efficiency)** |
| Brahmaputra | -10 | 0.06 | 0.009 | 0.80 |
| Ganges | 10 | 0.3 | 0.005 | 0.81 |
| Meghna | 2 | 0.3 | 0.1 | 0.91 |

[ここに入力]