# Peer review of "Necessary Storage As a Signature of Discharge Variability: Towards Global Maps"

_Hydrology and Earth System Sciences, 2016_

## Referee Comment (RC1) · Anonymous Referee #1 · 1 Nov 2016

General comments 1. Impact of climate change on necessary storages is assessed based on assumed target $QT = Qmean, 3Qmean, 0.5Qmean$. How can be applied for realistic conditions of the basin? If storage from 75 artificial dams in the Ganges are included as flood detention capacity or flood channel capacity, what will be necessary storage during flood?

Specific comments 1. P5 L2:". . . = AOEB" should be ". . .= ADEB". 2. P5L9: not found ". . .discussed in 5.1". 3. P5L19: if m' = 150 days, it is not consistent with the location of it on horizontal axis in Figure 3, where m' should be less than 50 days. 4. P5L30: ". . .= A'OEB' " should be ". . .= A'D'EB' ". 5. P9L29: what is the duration (m) for the results in Figure 7. 6. P10L5: ". . .in Fig. 5." should be ". . .in Fig. 6." 7. P11L9-10: ". . .in

Fig.9a" should be "...in Fig. 10a", "...in Fig.9b" should be "...in Fig. 10b". Define (a) and (b) in Figure 10. 8. P11L20: what is the Hurst coefficient for GBM? 9. Figure 1: include legend for 5 10 20 50 years and long-term mean discharge. 10. Figure 3: if m' is on the left of m, the value of m' < 50 days. 11. Figure 6-8, 10-11, what is the unit of both axis? Basin boundary presented by green and/or red line make confusion with color legend of necessary storages. 12. Table 2: there is no comment and discussion for the result in this Table.

[Figure]

---

## Referee Comment (RC2) · Anonymous Referee #2 · 8 Nov 2016

This is a very stimulating study on the geographical variations of "necessary storages". The sample river basin is the Ganges-Brahmaputra-Meghna basin in South Asia. Distributed river discharge data was calculated by a numerical hydrological model, BTOPMC.

What is "necessary storage" is not easy to describe very simply here in a limited space. But, I try to explain shortly what "necessary storage" is in this study as far as I can understand. Let's take a point or a place or a location in a river basin (and in the river channel network of the basin). In terms of "necessary storage" for flood, at the specified location, let's try to imagine how much volume of storage is necessary to keep flood water in the storage for releasing only reasonable water-flow to the downstream.

[Figure]

In terms of "necessary storage" for drought, let's imagine how much volume of storage is necessary at the specified location to maintain required water flow during a drought period in the downstream. Here, they do not mean an actual dam-reservoir would be constructed. "Necessary storage" is a virtual or hypothetical variable for representing a hydrological characteristic at the specified point in a river basin or in a river channel network. Please refer to the manuscript for more detailed and exact definitions. The authors expanded relatively old methodologies published in 1970's and 1980's or sometimes in 1950's and 1960's, and tried to apply those methodologies to a continental-scale or outputs of a continental-scale numerical hydrological simulation.

In the above, I mentioned "stimulating". It is true. I basically enjoy reading this manuscript. Their trial is somehow a novel one which I have never seen. In addition, this is a paper for the special issue in Honor of Eric F. Wood. Prof. Wood carried out studies on probabilistic aspects of hydrology in his 70's and 80's, and he carried out continental or global-scale modeling and remote-sensing studies in 90's and in 21st century. Thus, this manuscript by Takeuchi and Masood is very good for the special issue because they try to connect studies in 50-80's to contemporary continental-scale research. We sometimes neglect studies done in 50's, 60's and 70's. In this occasion, this can be a valuable paper.

However, I should mention that the structure and presentation of this manuscript is far from satisfactory. I do not know when is the deadline of this special issue, but I should demand thorough revision to the manuscript in terms of structure and presentation. I do not refer to problematic sentences and figures point-by-point, but I believe the authors can completely re-write this paper because they have experiences of writing papers in major journals.

Apart from the structure and presentation of the manuscript, for the contents of this paper, I also have several comments and I would recommend additional analyses in the following.

[Figure]

At first, the impact of climate change is not much important (and not much interesting, at this moment) in the context of this research. Thus, I recommend the authors to remove the aspects of climate change impact from this paper. Then, the paper will be a bit much more organized.

Next, although the authors can show the geographical distributions of outputs and variables as basic information like as Figure 6, 7, 8, I do not think geographical distribution is enough for what the authors want to discuss. As clearly seen in those figures in this paper, values of "necessary storages" are very much different between the main stream of a large river and a tributary. Whether the pixel is in the main stream (where the number of upstream pixels is several hundreds) or the pixel is in a tributary (where the number of upstream pixels is only several) affects "necessary storage" a lot. In accordance with what I wrote just above, the fact that two lines for Ganges and Brahmaputra in Figure 12 are separate is not much surprising. Those two lines just correspond to the main streams of two large rivers. Actually, dots in the same line (for the same major river) should not be treated independently.

What the authors want to show should not be the difference in "necessary storage" due to the difference in the number of upstream pixels (= upstream area). The difference in "necessary storage" due to the difference in the number of upstream pixels is not hydrological heterogeneity. In Figures 7 and 8, what can be seen most clearly is main streams of major rivers. That is probably not what the authors firstly want to show.

Rather, what the authors may be able to show is the impact of various parameters (like soil types, geology, arid or wet, snow-affected or not, hilly or not) on "necessary storages". For such an analysis, areas of catchments for analysis should be the same or similar. Such an analysis roughly corresponds to dots in the left edge in Figure 12. Why dots in the left edge in Figure 12 are very much scattering?

Even when the authors want to show the relation of "necessary storage" to catchment area, the authors are required to take different independent catchments; at least, even

in the same large river system, two independent tributaries should be taken for analysis. Or, in Figure 12, two different major rivers are converging to two different lines/points. This is somewhat interesting. Thus, another result which may be highlighted is various different converging lines/points of various major rivers of the world. For it, the simulation of the Ganges-Brahmaputra-Meghna only is not enough. Also, as written in the abstract, if the authors want to highlight representative elementary area of necessary storages which may be seen in Figure 12, more detailed analysis and discussion are appreciated. Even when the samples are only two (Ganges and Grahmaputra), detailed analysis and discussion will of help. By the way, what about Meghna?

I really appreciate the contribution of the authors to this special issue by trying to connect relatively old studies for catchment or local scale in 50's-80's to modern continental-scale research in the 21st century. This would be a very good contribution to the special issue for the Honor of Eric F. Wood. However, as a summary of what I mentioned above, and for making this manuscript to be published in HESS, I would recommend as follows:

- thorough revision to the presentation and structure of the manuscript. Here, I at least point out that: a) "case study area" and "data used" should come earlier because even Figure 1 uses the data, b) 4.2.1. may be combined with 2 as a theoretical framework, and how 4.2.1. was used for analysis, discussion and conclusion is unclear, c) captions of figures and tables should be written with enough information to understand figure/table adequately.

- I would recommend to stop showing the impact of climate change for making this manuscript simple, concise, clear, and appealing. I do not mean, climate change impact assessment is not of interest. But, because of time limitation and for highlighting what the authors want to say, I would recommend the authors to focus on hydrological characteristics under the current climate condition.

- Try to carry out additional analyses focusing on independent tributary catchments

(which have similar catchment areas, and belonging to different tributaries). For "necessary storages" of those catchments, the authors may be able to show influential parameters (soil types, geology, arid or wet, snow-affected or not, hilly or not) in a quantitative or qualitative way. I know, dots in the left edge in Figure 12 are outputs after standardization by average monthly flow. That is good. But, just showing a huge vertical scatter is not enough. We want to know what makes this vertical scatter. Only speculation is written in this manuscript. Additional analyses are necessary, even though this paper is a first trial and a first step.

- Of course, two different converging lines for Ganges and Brahmaputra is interesting, but at least a few more examples would be appreciated. Also, discussion on why those two converging lines are different should be made.

Finally, again, a stimulating paper is very important. Nowadays, well-organized but non-stimulating papers are dominating. That's not fun. But, well-organized or adequately-organized is necessary for publication. Depending on the focus of this study, the authors may not take all the points I raised.

---

## Referee Comment (RC3) · Anonymous Referee #3 · 15 Nov 2016

Review of "Indicators of Necessary Steps for Flood and Drought Management: Towards Global Maps"

I struggled with several aspects of this paper, all of which contributed to my decision to recommend rejection of the manuscript.

1 - The novelty of the proposed method is not clear. The method appears to be a minor adjustment to previously published versions of this approach. The authors note this in page 3, lines 3-5. Although in my reading of Takeuchi (1986), it appears that the methodology presented using both the FDC and DDC may not have been previously published. Regardless, there are fairly well-established methods to determine the necessary storage in hydrologic design; some of these papers are mentioned briefly in the

introduction but there is no attempt to demonstrate the utility of this approach in the context of these other well-established methods - not necessarily show this method is better but - at a minimum - that it performs as well as other methods.

2 - This approach is based on the flow-duration curve, which does not consider the timing or variability of the discharge and, therefore, the accumulation or depletion of storage over time. It is then not clear how this approach can be useful, as it does not consider the storage in the previous time step, particularly for rivers where variability is large and storage is most needed to control this variability.

3 - Following from comment 2, it is repeated throughout the manuscript, "storage is the means to control discharge variation." The relation between variation in discharge and necessary storage is well established in the literature, with more storage needed as the coefficient of variation in the discharge increases; and yet, there is no consideration of this point in the demonstration of these methods. The coefficient of variation (CV) is not reported for the 3 demonstration sites so the reader has no idea of the variation of the discharge that is being "smoothed" by storage under the calibration/validation dataset.

4 - There are large gaps in the approach and justification missing from the manuscript:

a) The decisions made in Sections 2.2.1 and 2.2.2 seem quite arbitrary with no justification or support to suggest that these would be choices that a water manager or operator would likely choose.

b) Another example is in the assumption that the FDC and DDC curves follow a generalized extreme value distribution. There are no references provided to support this choice of distribution from previously published work and no evidence is presented to demonstrate that this is a reasonable choice.

c) Page 3, line 28 states that the FDC and DDC curve applies precipitation, yet there is no explanation of this further in the manuscript. How is precipitation used in the

method?

d) The methodology described on p. 5, line 6 states that the "interest duration is limited to a year" but I am left to wonder how the analysis is applied to rivers where over-year storage is an important component of controlling variability?

4 - Editorial issues

a) There are incomplete sentences: p. 8, l. 13; p. 2, l. 12-14

b) Acronyms that are not explained before being used. For example, p. 7, l. 10; Abstract, l. 18-19; p. 5, l. 1-2.

c) Although I do not recall that HESS has guidance on the use of "he" and "his," I think is a lack of sensitivity to use a gendered pronoun and there are alternative ways to phrase these sentences. Page 4, line 17 is the first appearance but gendered pronouns appear in quite a few other places, such as throughout Section 2.2.2.
* * *

---

## Referee Comment (RC4) · Anonymous Referee #4 · 22 Nov 2016

This is an interesting paper that could eventually be published but significant revisions are required. I outline the main issues below along with some comments and questions that should be addressed:

1.     The writing/grammar and arrangement of the paper is poor.     Several spelling/grammar issues and "awkward" sentences that have to be read a few times to try and understand what the authors are trying to say. Most of the issues are minor but there is too many to list and I suspect this is detracting from the main points the paper is trying to make.

2. The concept being introduced is interesting. The FDC-DDC method proposed has some advantages over the mass curve method.  It is a simple approach but is also

flawed in that stationary hydroclimatic conditions are assumed. Also, a major strength of the FDC-DDC method is its simplicity and transferability (e.g. the authors say it could be used to create global maps of necessary storage and the state of water resources) but then I wonder why if the method is so simple and transferable is it only demonstrated for one basin?

3. The other major problem I have with this paper is the way the impact of climate change is simulated. Only 3 lines worth of explanation (sect 3.2.1) are given to explain this and it is not clear at all how the GCM outputs were used as inputs to the hydrological modelling? Which variables were used? At what time step? I assume daily (or maybe monthly) and if so there are known to be significant issues associated with daily GCM data and bias correction is usually required? Precipitation data from GCMs is particularly problematic, especially in the Asian monsoon region where this study is focussed. How were the biases associated with GCM outputs addressed?

4. Sect 3.2.2.1, lines 15-21 is also a bit confusing. . ...here you say CRU data was used for PET. . ..but then in the next sentence you also say that Zhou et al (2006) method was used to compute PET? Why do you need to compute PET if you already have it from CRU. Similarly, you say APHRODITE precip data is used but the previous section and the next section indicate that MRI-AGCM model data is used for the hydro modelling? Maybe you used APHRODITE for the bias correction or maybe APHRODITE was used as the baseline data and the perturbed based on climate change factors from the MRI-AGCM??? Either way some more detailed explanation is required as to what you actually used to run the hydro model (under both the current and future climate simulations).

5. Other problem is you have just used one GCM and just one emissions scenario (and it is an out of date emission scenario also, IPCC has moved from SRES to RCP several years ago now). I realise at start of Sect 3 you explain you just use one GCM projection as proof of concept. This I guess is ok in a paper like this where you are just demonstrating a method but given one of your main claims is that this FDC-DDC

method is easy to apply it should be the case that running multiple GCM/emission scenarios through the method and comparing the differences should be ok. This would make your argument for the acceptance of this method more convincing (as would inclusion of a few other case study locations—as per comment #2). Assuming one GCM is enough to demonstrate your concept I guess is possibly ok. . . . . .but what is definitely not ok is to then make concluding statements that suggest that what the findings/results from your one GCM example are somehow indicative of what will happen (they might be but there is a lot of uncertainty associated with future projections and you need to convey that). For example, concluding point #7 you say "CC impacts on floods increases". . . . . . . . .based on your single model study maybe this might be true but that is just one plausible scenario. . . . . .there are many other equally plausible scenarios and, as per latest IPCC findings and many other papers focussing on this region and elsewhere around the world, there is no consensus either way on whether floods will increase or decrease. . . . . . . . ...same issue when you say "impact decreases the necessary storage for drought management,. . ..". . ...this is just based on the single GCM you assessed. . . . . .based on just a single GCM run using just one emission scenario you should not be making such a definite conclusion such as this (which could have quite serious and expensive practical implications if decision-makers accepted and acted on this conclusion).

6. Your concluding point #9. . ...this is a good recommendation to use this FDC-DDC method in as many places as possible to create global maps of the necessary storage and water resources situation. . . . . . . ..but as per previous comment, to cover the climate change impact bit you need to put something in about repeating this using multiple different future scenarios (i.e. different GCMs, different emission scenarios etc). . . . . . . ..then you might get towards some sort of consensus. Your method could be applied at major basins around the world using, for example, the GCM info available at the CMIP websites and some appropriate downscaling and bias correction methods (also GCM selection methods if required). . ..this would be a useful exercise but it is pointless doing it based on the outputs from just a single GCM as that doesn't really
tell us anything much about what is possible in the future. Refer to some of the work done by CSIRO for the Murray-Darling Basin in Australia and also some of the work done by Mekong River Commission for examples of how to comprehensively assess potential impacts of climate change on water resources (i.e. using climate change projections from multiple GCMs and multiple emission scenarios)

---

## Referee Comment (RC5) · Anonymous Referee #5 · 4 Dec 2016

This study presents the use of Flood Duration Curve (FDC) and Drought Duration Curve (DDC) as indicators of necessary storages for water management. The presented materials are generally very stimulating by revisiting the creative method developed in 1980s. The application of the traditional method to the latest spatially distributed model results with climate change projections can provide new insights into the interpretations of simulation results. I believe this paper is relevant also to the special issue in honor of Prof. Eric F. Wood.

My major review comment on the current manuscript, however, is that the central theme of this manuscript is ambiguous. In the manuscript, I see at least the following five topics are presented in a mixed manner.

1) Authors promote the application of various traditional analysis approaches, in particular FDC and DDC here, with a large dataset in modern days to obtain practical implications.

2) FDC and DDC curves have been used previously for a dam operation at a single site, while the authors in this manuscript extend the method to spatially distributed data.

3) Authors claim the use of FDC and DDC enabled to characterize necessary storages in the Ganges-Brahmaputra-Meghna (GBM) basin.

4) Authors claim most of recent climate change impact assessment studies simply evaluate the increase or decrease of hydrologic variables. On the other hand, the presented approach with FDC-DDC can provides different perspective to interpret climate change projections suitable for practical water resource management.

5) Finally the authors attempt to present the projected climate change impact in the GBM basin.

I believe all the above issues are equally important. Meanwhile with such a many topics, I found difficulty in understanding the main message by the authors. For example, the introduction mainly reviews the original concept of FDC-DDC with some other similar approaches but not necessary arguing the point of 1). The method section solely reviews the FDC and DDC methods with some extensions to the spatially application i.e. point 2). The result sections including the conclusions focus mostly on 3) - 5), whose issues are not well explained in the introduction.

Personally I believe this paper can improve the readability if the authors express their own points on the 1) and 2) in result, discussion and conclusion sections. Just revisiting traditional approach cannot be accepted in a scientific paper, but this is not the case with demonstrating further extensions.

In addition to the above major comments, I have the following minor review comments.

Please add some more explanations on the practical use of the quantified necessary

storages for river basin managers. Especially for such a large river basins, the meaning of smoothing discharge at a particular river section should be carefully discussed. For example, smoothing river discharge at an upstream point with smaller storage and at a downstream point with large storage have different impacts for both flood (at the downstream of the reservoirs) and drought. Hence I wonder for the effective use of the information, it requires some additional information such as the impact of smoothing to the downstream areas etc. for practical applications. This comment does not request for additional analysis but requesting for how the spatially distributed necessary storage inforamtion can be used in practices.

P2 L24-26 The part of "its scale is different from that of elementary hydrological processes in a small catchment" is unclear. The similar sentences appear also in 4.2.2 describing Representative Elementary Area (REA), but the current manuscript is still unclear how the scale issue dealt in this study is related to REA.

P5 L2 AOEB -> ADEB

P9 L1 What is the relationship between "WATCH Forcing Data set (WFD)" and previously described datasets including CRU and APHRODITE in 3.2.2.1 in the presented simulation.

P12 4.2.1 Please explain the motivation of this discussion at the beginning of this subsection or in the introduction, otherwise this part sounds a bit too sudden and not well connected to the other part.

---

## Editor Comment (EC1) · M. Sivapalan (Editor) · 1 Jan 2017

The authors have been well-served by the comments from 5 reviewers, who have offered very constructive comments and criticisms on the paper. I am also glad that the authors have responded positively to these comments.

The following is a summary of the outcomes of the public discussion on the paper so far:

1. There has been a misunderstanding in the minds of most (all) reviewers about the main aim of the paper, and the authors have partially clarified these. Really, the main of the paper is to characterize the global patterns of variation of long-term hydrologic

variability (in time), of droughts and floods. For this purpose the authors have decided to use a composite measure, which is the reservoir storage needed to meet an average demand (or something similar to this). This paper is a paper on hydrologic variability and not on reservoir design, as some of the reviewers misunderstood. It is important that the authors take particular care to present these clear aims at the beginning so this misunderstanding does not arise.

2. In the same spirit, the paper will become clearer and will have high impact if the authors can simplify the paper and reorganize it so that the main message comes out more clearly in the rest of the paper and does not veer off beyond the main message.

3. Also in this respect, all reviewers agree that the analysis of climate change impacts does not add much to the paper, and only distracts from the main message. I am glad the authors have already agreed to remove it from the paper, which I encourage. Depending on subsequent reaction to this paper, the authors may decide to look at a subsequent paper looking at how climate change affects the global-regional patterns long-term, temporal variability (as measured in terms of reservoir storage). I am not sure if it will make any contribution to climate change research.

4. There was some discussion about the appropriateness of the FDC approach to reservoir size estimation. The authors defended it, and I support their argument. The FDC approach was indeed used for reservoir sizing more than 50 years ago, and has now been forgotten, and superseded by more modern methods like range analysis etc. However, as a simple rule of thumb it is quite useful since one can easily see the connection between hydrologic variability and reservoir size. I don't mind the authors reviving the approach here - but they should explain its meaning for the average reader, and how good a measure it is about hydrologic variability.

In conclusion, this is an unconventional paper, and I am so glad that the reviewers did not dismiss it out of hand. If one takes a higher level and long-term historical perspective, and look at understanding the world as it is and finding ways to characterize

hydrological variability globally, this may be a very interesting approach and I applaud the authors for introducing it. It is entirely appropriate for a special issue in honor of Professor Eric Wood. It draws a connection between modern hydroclimatology as seen in the many other papers submitted to the special issue to approaches used more than 50 years ago when Professors Wood and Takeuchi started their careers, and learn how much has changed and how much has not. The rest of the hydrologic community can benefit from this broader perspective.

For this reason I encourage the authors to resubmit a substantially revised paper (along the lines suggested by the reviewers and the authors themselves), which I will consider for further (non-public) review by some of the critical reviewers before final consideration for publication in HESS.

---

## Author Comment (AC1) · 1 Jan 2017

RESPONSE TO THE REVIEWER #1'S COMMENTS

We are grateful to Reviewer #1 for the helpful and insightful comments. The provided comments have contributed substantially to improving the manuscript. Accordingly, we have made significant efforts to revise the manuscript with the details being explained as follows.

General comments Point #1

COMMENT: Impact of climate change on necessary storages is assessed based on assumed target $Q_T$ = Qmean, 3Qmean, 0.5Qmean. How can be applied for realistic

conditions of the basin? If storage from 75 artificial dams in the Ganges are included as flood detention capacity or flood channel capacity, what will be necessary storage during flood?

RESPONSE: Thanks for the comments. Although the proposed indicator claims that its use extends to climate change impact assessment, it is only on a basic nature in this paper and there is no intention for concrete practical assessment. Adaptation with existing 75 reservoirs is a major question but it is out of the scope of this paper. The focus is limited to variability of discharge due to hydrological heterogeneity and not socio-economic activities. In order to avoid confusion, we revised the paper omitting climate change impact assessment (4.1.2 and related paragraphs).

Specific comments Point #1

COMMENT: P5 L2:": : : = AOEB" should be ": : := ADEB".

RESPONSE: Thanks for the comment. Indeed! Accordingly, we have revised it.

Point #2

COMMENT: P5L9: not found ": : :discussed in 5.1"

RESPONSE: Thanks for the comments. Corrected it to 4.2.1.

Point #3

COMMENT: P5L19: if m' = 150 days, it is not consistent with the location of it on horizontal axis in Figure 3, where m' should be less than 50 days.

RESPONSE: Yes, it is. The schematic DDC curve has been changed to a more realistic one to avoid such confusion.

Point #4

COMMENT: P5L30: ": : := A'OEB' " should be ": : := A'D'EB' "..

RESPONSE: Thanks for the comments. It has been corrected accordingly.

Point #5

COMMENT: P9L29: what is the duration (m) for the results in Figure 7.

RESPONSE: Thanks for your comments. "m" is different at each point and geographical distribution of m opens another interesting discussion of hydrological heterogeneity. It is not treated here.

Point #6

COMMENT: P10L5: ": : :in Fig. 5." should be ": : :in Fig. 6."

RESPONSE: Thanks for your comments. Yes, it has been corrected accordingly.

Point #7

COMMENT: P11L9-10: ": : :in Fig.9a" should be ": : :in Fig. 10a", ": : :in Fig.9b" should be ": : :in Fig. 10b". Define (a) and (b) in Figure 10.

RESPONSE: Thanks for your comments. The differences between necessary storage in km3 and months are indicated not only in Fig. 10a and b but also 7a and b, 8a and b, 10a and b and 11a and b. They all are corrected and (a) and (b) are indicated in the Figures. According to the delete of climate change analyses, Fig. 10 and 11 were deleted.

Point #8

COMMENT: P11L20: what is the Hurst coefficient for GBM?

RESPONSE: Thanks for your comments. It is not calculated and beyond the scope of this paper.

Point #9

COMMENT: Figure 1: include legend for 5 10 20 50 years and long-term mean discharge.

[Figure]

RESPONSE: Thanks for your suggestion. Accordingly, the legend is included in Figure 1.

Point #10

COMMENT: Figure 3: if m' is on the left of m, the value of m' < 50 days.

RESPONSE: Thanks for your comment. Yes, but for a drought discussion, it is more practical to be 150 days and accordingly the schematic DDC curve was changed to a more realistic one.

Point #11

COMMENT: Figure 6-8, 10-11, what is the unit of both axis? Basin boundary presented by green and/or red line make confusion with color legend of necessary storages.

RESPONSE: Thanks for your comment. E (degree East) and N (degree North) were added. The color of all boundaries was changed to black lines.

Point #12

COMMENT: Table 2: there is no comment and discussion for the result in this Table.

RESPONSE: Thanks for your comment. It is referred and discussed in the last para of 4.1.2. in page 11. But they all are deleted to omit climate change analysis.

---

## Author Comment (AC2) · 1 Jan 2017

RESPONSE TO THE REVIEWER #2'S COMMENTS

We are grateful to Reviewer #2 for the helpful and insightful comments. The provided comments have contributed substantially to improving the manuscript. Accordingly, we have made significant efforts to revise the manuscript with the details being explained as follows.

Point #1

COMMENT: This is a very stimulating study on the geographical variations of "necessary storages". The sample river basin is the Ganges-Brahmaputra-Meghna basin in

South Asia. Distributed river discharge data was calculated by a numerical hydrological model, BTOPMC. What is "necessary storage" is not easy to describe very simply here in a limited space. But, I try to explain shortly what "necessary storage" is in this study as far as I can understand. Let's take a point or a place or a location in a river basin (and in the river channel network of the basin). In terms of "necessary storage" for flood, at the specified location, let's try to imagine how much volume of storage is necessary to keep flood water in the storage for releasing only reasonable water-flow to the downstream. In terms of "necessary storage" for drought, let's imagine how much volume of storage is necessary at the specified location to maintain required water flow during a drought period in the downstream. Here, they do not mean an actual dam-reservoir would be constructed. "Necessary storage" is a virtual or hypothetical variable for representing a hydrological characteristic at the specified point in a river basin or in a river channel network. Please refer to the manuscript for more detailed and exact definitions. The authors expanded relatively old methodologies published in 1970's and 1980's or sometimes in 1950's and 1960's, and tried to apply those methodologies to a continental-scale or outputs of a continental-scale numerical hydrological simulation.

RESPONSE: Thank you very much for your positive comment. Thank you also for your clear interpretation of "necessary storage". It is indeed "not an actual dam-reservoir would be constructed . . . but a virtual or hypothetical variable for representing a hydrological characteristic at the specified point in a river basin or in a river channel network." The foci of this paper are the importance of the study of hydrological heterogeneity in storage domain and for that purpose the use of FDC-DDC based calculation is useful.

Point #2

COMMENT: In the above, I mentioned "stimulating". It is true. I basically enjoy reading this manuscript. Their trial is somehow a novel one which I have never seen. In addition, this is a paper for the special issue in Honor of Eric F. Wood. Prof. Wood carried out studies on probabilistic aspects of hydrology in his 70's and 80's, and he carried

out continental or global-scale modeling and remote-sensing studies in 90's and in 21st century. Thus, this manuscript by Takeuchi and Masood is very good for the special issue because they try to connect studies in 50-80's to contemporary continental-scale research. We sometimes neglect studies done in 50's, 60's and 70's. In this occasion, this can be a valuable paper. However, I should mention that the structure and presentation of this manuscript is far from satisfactory. I do not know when is the deadline of this special issue, but I should demand thorough revision to the manuscript in terms of structure and presentation. I do not refer to problematic sentences and figures point-by-point, but I believe the authors can completely re-write this paper because they have experiences of writing papers in major journals.

RESPONSE: Thanks for the comments. We are clearly aware of the incompleteness of the paper and will completely rewrite it especially clarifying the focus and intention. The concrete plan of rewriting is in the next Point #3.

Point #3

COMMENT: Apart from the structure and presentation of the manuscript, for the contents of this paper, I also have several comments and I would recommend additional analyses in the following. At first, the impact of climate change is not much important (and not much interesting, at this moment) in the context of this research. Thus, I recommend the authors to remove the aspects of climate change impact from this paper. Then, the paper will be a bit much more organized.

RESPONSE: Thank you for your comment. We agree that a sharper focus of the paper is necessary to make the objective of the paper clearer. Accordingly, we remove the climate component of the presentation. This accords with quite many comments and suggestions of other reviewers, too.

The main revision and restructuring of the manuscripts are as follows: 1. Rewriting the whole manuscript with clearer structuring and explanations. 2. Introduction 1 states two main foci clear: geographical distribution of necessary storages and the use of FDC-

[Figure]

DDC to calculate necessary storages. 3. Removal of climate change analyses of 4.1.2 and related paragraphs. 4. Major improvement of 4.2.2 and Fig 12 on heterogeneity analyses of necessary storages. This will be done by adding extra analyses of sub-basins and new figures of topography, precipitation, vegetation cover, simulated root zone moisture contents etc. Also by zooming up of the wide variation of necessary storages in months at the left edges of Fig. 12 or in the smaller catchment areas. 5. Case study area 3.1 will be moved to earlier section 1 as the region is introduced in early stage. 6. Introduction of range theory 4.2.1 merges to 2.3 with other theories. 7. Captions of tables and figures will be improved such as adding a legend of Fig. 1 and Fig. 6-8 etc.

Point #4

COMMENT: Next, although the authors can show the geographical distributions of outputs and variables as basic information like as Figure 6, 7, 8, I do not think geographical distribution is enough for what the authors want to discuss. As clearly seen in those figures in this paper, values of "necessary storages" are very much different between the main stream of a large river and a tributary. Whether the pixel is in the main stream (where the number of upstream pixels is several hundreds) or the pixel is in a tributary (where the number of upstream pixels is only several) affects "necessary storage" a lot. In accordance with what I wrote just above, the fact that two lines for Ganges and Brahmaputra in Figure 12 are separate is not much surprising. Those two lines just correspond to the main streams of two large rivers. Actually, dots in the same line (for the same major river) should not be treated independently.

RESPONSE: Thanks for the comments. Fig. 6 shows mean annual precipitation, mean annual discharge in m3/sec, its standard deviation in m3/sec and coefficient of variation. Geographical information is indeed too little. Accordingly, topographical and landcover data of elevation, soil, vegetation and land use classification were added. Besides, Simulation results of root zone moisture were added that was analysed by Gao et al. (2014). Gao, H., M. Hrachowitz, S.J. Schymanski, F. Fenicia, N. Sriwongsi-

tanon, H.H.G. Savenije (2014), Climate controls how ecosystems size the root zone storage capacity at catchment scale, Geophysical Research Letters, 41, 7916-7923, doi: 10.1002/2014GL061668.

Point #5

COMMENT: What the authors want to show should not be the difference in "necessary storage" due to the difference in the number of upstream pixels (= upstream area). The difference in "necessary storage" due to the difference in the number of upstream pixels is not hydrological heterogeneity. In Figures 7 and 8, what can be seen most clearly is main streams of major rivers. That is probably not what the authors firstly want to show. Rather, what the authors may be able to show is the impact of various parameters (like soil types, geology, arid or wet, snow-affected or not, hilly or not) on "necessary storages". For such an analysis, areas of catchments for analysis should be the same or similar. Such an analysis roughly corresponds to dots in the left edge in Figure 12. Why dots in the left edge in Figure 12 are very much scattering?

RESPONSE: Thanks for your comments. As stated above, in addition to the difference with the upstream catchment area, relation with forcing input (unfortunately except snowfall so far), soil, vegetation, land use etc. will also be added. They would control the impact of catchment area on how soon the necessary storages converge to its areal average.

Point #6

COMMENT: Even when the authors want to show the relation of "necessary storage" to catchment area, the authors are required to take different independent catchments; at least, even in the same large river system, two independent tributaries should be taken for analysis. Or, in Figure 12, two different major rivers are converging to two different lines/points. This is somewhat interesting. Thus, another result which may be high-lighted is various different converging lines/points of various major rivers of the world. For it, the simulation of the Ganges-Brahmaputra-Meghna only is not enough. Also,

as written in the abstract, if the authors want to highlight representative elementary area of necessary storages which may be seen in Figure 12, more detailed analysis and discussion are appreciated. Even when the samples are only two (Ganges and Grahmaputra), detailed analysis and discussion will of help. By the way, what about Meghna?

RESPONSE: Thanks for your comments. In fact the Meghna case is already included in red color in Fig 12. But as you suggest we will add some sub-basin details. We select two sub-basins in the Ganges and one each from the Brahmaputra and the Meghna. Their distribution at the left edge and the converging lines in Fig 12 will be zoomed up and related with their basin characteristics in climate, topography, geology, land cover etc. Some upward conversing lines seen in the Ganges (in green dots) in Fig. 12 will also be examined why those increasing lines of sub-basins are formed both in FDC and DDC. They all are approaching to the high converging line of the Ganges. They seem to imply that the overall converging line is determined by trends of the sub-basins. In other words, as the Ganges sub-basins have the increasing trend, the final converging line is high.

Point #7

COMMENT: I really appreciate the contribution of the authors to this special issue by trying to connect relatively old studies for catchment or local scale in 50's-80's to modern continental-scale research in the 21st century. This would be a very good contribution to the special issue for the Honor of Eric F. Wood. However, as a summary of what I mentioned above, and for making this manuscript to be published in HESS, I would recommend as follows: - thorough revision to the presentation and structure of the manuscript. Here, I at least point out that: a) "case study area" and "data used" should come earlier because even Figure 1 uses the data, b) 4.2.1. may be combined with 2 as a theoretical framework, and how 4.2.1. was used for analysis, discussion and conclusion is unclear, c) captions of figures and tables should be written with enough information to understand figure/table adequately..

RESPONSE: Thanks for your very positive comments. We certainly will do revision according to your suggestions as described in #3. We will do our best for improving clarity and readability.

Point #8

COMMENT: - I would recommend to stop showing the impact of climate change for making this manuscript simple, concise, clear, and appealing. I do not mean, climate change impact assessment is not of interest. But, because of time limitation and for highlighting what the authors want to say, I would recommend the authors to focus on hydrological characteristics under the current climate condition.

RESPONSE: Thanks for your comments. We will remove all climate change discussions except mentioning its applicability showing before and after of FDC and DDC in Fig 9 which we believe enough to indicate potential climate change analysis in the future.

Point #9

COMMENT: - Try to carry out additional analyses focusing on independent tributary catchments (which have similar catchment areas, and belonging to different tributaries). For "necessary storages" of those catchments, the authors may be able to show influential parameters (soil types, geology, arid or wet, snow-affected or not, hilly or not) in a quantitative or qualitative way. I know, dots in the left edge in Figure 12 are outputs after standardization by average monthly flow. That is good. But, just showing a huge vertical scatter is not enough. We want to know what makes this vertical scatter. Only speculation is written in this manuscript. Additional analyses are necessary, even though this paper is a first trial and a first step.

RESPONSE: Thanks for your suggestion. Yes, we consider this is the major point we have to revise. We will select some independent sub-basins as indicated in #6, draw an equivalent of Fig. 12 and analyse their distribution from left edge to the converging

points in relation to topography, soil types, rainfall, vegetation, landuse etc. We hope more insights will be obtained about the hydrological heterogeneity in storage domain.

Point #10

COMMENT: - Of course, two different converging lines for Ganges and Brahmaputra is interesting, but at least a few more examples would be appreciated. Also, discussion on why those two converging lines are different should be made.

RESPONSE: Thanks for your comment. As mentioned in #6 and #9 above, independent sub-basin analyses will be added. The difference in converging level indicates the relative magnitude of variation of discharge which reflects variability of precipitation the basin receives and the surface and sub-surface retardation function of the basin. The quantitative relation needs much more study and this paper will only indicates the qualitative relations.

Point #11

COMMENT: Finally, again, a stimulating paper is very important. Nowadays, well-organized but nonstimulating papers are dominating. That's not fun. But, well-organized or adequatelyorganized is necessary for publication. Depending on the focus of this study, the authors may not take all the points I raised.

RESPONSE: Thank you so much for your encouraging comments. We will do our best for the improvements at all points you pointed out and we responded above.

---

## Author Comment (AC3) · 1 Jan 2017

RESPONSE TO THE REVIEWER #3'S COMMENTS

We are grateful to Reviewer #3 for the helpful and insightful comments. The provided comments have contributed substantially to improving the manuscript. Accordingly, we have made significant efforts to revise the manuscript with the details being explained as follows.

General comments Point #1

COMMENT: The novelty of the proposed method is not clear. The method appears to be a minor adjustment to previously published versions of this approach. The authors

note this in page 3, lines 3-5. Although in my reading of Takeuchi (1986), it appears that the methodology presented using both the FDC and DDC may not have been previously published. Regardless, there are fairly well-established methods to determine the necessary storage in hydrologic design; some of these papers are mentioned briefly in the introduction but there is no attempt to demonstrate the utility of this approach in the context of these other well-established methods - not necessarily show this method is better but - at a minimum - that it performs as well as other methods..

RESPONSE: Thank you for your critical comments. The paper indeed uses the methodology of FDC-DDC for calculating necessary storages that was developed during 1975 to 1988. But the objective of this paper is not to reintroduce this methodology and compare with other methods but to use it for identifying spatial heterogeneity of hydrology in storage domain for which this method is best suited. The FDC-DDC method is very different from Ripple's mass curve method or the well-established simulation method. It utilizes the intensity-duration-frequency curve which has never been used for calculating necessary storage before 1975 and still not well known. This is why this paper introduces the method fairly in detail.

Point #2

COMMENT: This approach is based on the flow-duration curve, which does not consider the timing or variability of the discharge and, therefore, the accumulation or depletion of storage over time. It is then not clear how this approach can be useful, as it does not consider the storage in the previous time step, particularly for rivers where variability is large and storage is most needed to control this variability.

RESPONSE: Thanks for the comments. Yes, this approach utilizes flow-duration curves which do consider timing and variability of discharge in the way that, for instance, whatever the timing or variability is fïĄć(m) will be available as an average over m days from now with the rate of failure no more than ïĄć. The observation that it does not consider the accumulation or depletion of storage over time is incorrect. The IDF is

a probabilistic and stationary approach and FDC-DDC can be used as a prediction of inflow before or during floods or drought. Such explanation will be added in the revised manuscript.

Point #3

COMMENT: Following from comment 2, it is repeated throughout the manuscript, "storage is the means to control discharge variation." The relation between variation in discharge and necessary storage is well established in the literature, with more storage needed as the coefficient of variation in the discharge increases; and yet, there is no consideration of this point in the demonstration of these methods. The coefficient of variation (CV) is not reported for the 3 demonstration sites so the reader has no idea of the variation of the discharge that is being "smoothed" by storage under the calibration/validation dataset.

RESPONSE: Thanks for the comments. Yes, although the CV is presented in Fig. 6 but not specific values at the demonstration sites in table 2. It will be shown and discussed in relation to necessary storages identified at three points (1.10, 0.69 and 0.90 at the outlets of the Ganges, the Brahmaputra and the Meghna). Obviously CV is the most directly connected parameter to the necessary storages to smooth out variations but we consider that the necessary storage for floods and droughts deliver much more concrete idea on the variability to hydrologists and basin managers.

Point #4a

COMMENT: The decisions made in Sections 2.2.1 and 2.2.2 seem quite arbitrary with no justification or support to suggest that these would be choices that a water manager or operator would likely choose.

RESPONSE: Thanks for the comments. Yes, it is arbitrary and does not represent real operator's choice. But we consider this represents the basic nature of reservoir operation that is to consider expected inflow, target output and available storage. In that

sense, the assumed decision making process serves to derive a meaningful indicator.

Point #4b

COMMENT: Another example is in the assumption that the FDC and DDC curves follow a generalized extreme value distribution. There are no references provided to support this choice of distribution from previously published work and no evidence is presented to demonstrate that this is a reasonable choice.

RESPONSE: Thanks for the comments. Generalized Extreme Value (GEV) distribution Type-1, Gumbel distribution, has been used in this study to estimate extreme values. Because, in previous studies, for frequency analyses, the Gumbel distribution has been recommended for the major rivers in Bangladesh by Mirza (2002) as well as for relatively smaller data samples by Hirabayashi et al. (2013). We have revised 2.1 with these references.

Point #4c

COMMENT: Page 3, line 28 states that the FDC and DDC curve applies precipitation, yet there is no explanation of this further in the manuscript. How is precipitation used in the method?

RESPONSE: Thanks for the comments. In this manuscript, FDC and DDC method is applied on discharge time series only. The application on precipitation time series was presented in other papers such as Takeuchi, 1988.

Point #4d

COMMENT: The methodology described on p. 5, line 6 states that the "interest duration is limited to a year" but I am left to wonder how the analysis is applied to rivers where over-year storage is an important component of controlling variability?

RESPONSE: Thanks for the comments. Yes, in many large rivers with a large reservoir over-year storage is vital especially for drought management with a large target output.

But in this paper the over-year storage was omitted for simplicity since the main focus is the relation between necessary storages and catchment heterogeneity where other than topography, geology, soil and vegetation, the seasonal variation of meteorology is the major controlling factor. We consider this applies most of basins in the world except deserts or very arid regions. Besides, the procedure for calculation necessary storages is same for any m however large it is. In the item 9 of both 2.2.1 and 2.2.2, the value of m does not necessarily limited to one year. If multi-year operation is concerned, the value of m in Eq. 5 and 6 should vary all the way of available FDC and DDC extending from m=1 to multi years.

Editorial issues Point #5a

COMMENT: There are incomplete sentences: p. 8, l. 13; p. 2, l. 12-14.

RESPONSE: Thanks for the comment. They are complete sentences, but not elaborate enough and accordingly elaborated. Besides, climate change aspects were removed.

Point #5b

COMMENT: Acronyms that are not explained before being used. For example, p. 7, l. 10; Abstract, l. 18-19; p. 5, l. 1-2.

RESPONSE: Thanks for the comments. All will be spelled out when they appear at first such as Meteorological Research Institute – Atmospheric Global Circulation Model 3.2S (MRI-AGCM3.2S).

Point #5c

COMMENT: Although I do not recall that HESS has guidance on the use of "he" and "his," I think is a lack of sensitivity to use a gendered pronoun and there are alternative ways to phrase these sentences. Page 4, line 17 is the first appearance but gendered pronouns appear in quite a few other places, such as throughout Section 2.2.2.

RESPONSE: Thanks for the comment. Indeed, only 6 places of he/she should not be

simplified by he. It was corrected.

---

## Author Comment (AC4) · 1 Jan 2017

RESPONSE TO THE REVIEWER #4'S COMMENTS

We are grateful to Reviewer #4 for the helpful and insightful comments. The provided comments have contributed substantially to improving the manuscript. Accordingly, we have made significant efforts to revise the manuscript with the details being explained as follows.

Point #1

COMMENT: The writing/grammar and arrangement of the paper is poor. Several spelling/grammar issues and "awkward" sentences that have to be read a few times

to try and understand what the authors are trying to say. Most of the issues are minor but there is too many to list and I suspect this is detracting from the main points the paper is trying to make.

RESPONSE: Thanks for the comments. Accordingly, for better understanding, we thoroughly revise our manuscript with correcting all spelling or grammatical mistakes and reformulating the awkward sentences as much as possible.

Point #2

COMMENT: The concept being introduced is interesting. The FDC-DDC method proposed has some advantages over the mass curve method. It is a simple approach but is also flawed in that stationary hydroclimatic conditions are assumed. Also, a major strength of the FDC-DDC method is its simplicity and transferability (e.g. the authors say it could be used to create global maps of necessary storage and the state of water resources) but then I wonder why if the method is so simple and transferable is it only demonstrated for one basin?

RESPONSE: Thanks for the comments. One of the aim of this manuscript is methodological demonstration of the extended application of FDC-DDC to show spatial distribution of necessary storages. And for that purpose, we took three distinct basins; the Ganges, the Brahmaputra and the Meghna as a case study. The global application is another step. But within the GBM basin, we have subdivided the three basins into several sub-basins with distinct characteristics to analyze hydrological heterogeneity. With such analyses, we consider it enough for demonstrating the theoretical framework.

Point #3

COMMENT: The other major problem I have with this paper is the way the impact of climate change is simulated. Only 3 lines worth of explanation (sect 3.2.1) are given to explain this and it is not clear at all how the GCM outputs were used as inputs to the hydrological modelling? Which variables were used? At what time step? I assume

daily (or maybe monthly) and if so there are known to be significant issues associated with daily GCM data and bias correction is usually required? Precipitation data from GCMs is particularly problematic, especially in the Asian monsoon region where this study is focussed. How were the biases associated with GCM outputs addressed?

RESPONSE: Thanks for the comments. We fully agree with the Reviewer's comment. In order to avoid the complexity and keep the focus of this paper clear, the climate change components including 3.2.1 will be removed from the entire manuscript. But as an example to show applicability to climate change analyses, Fig.9 is left remain and this explanation was put into its caption with citing references for detail. Regarding your question, daily time series of two variables, precipitation and temperature have been used as inputs to the hydrological modeling to obtain discharge time series for present and future time period. The bias of precipitation dataset has been corrected by multiplying using monthly correction coefficient (ratio between basin averaged long term monthly mean precipitation from WFD and that from the GCM) for each basin.

Point #4

COMMENT: Sect 3.2.2.1, lines 15-21 is also a bit confusing: : :..here you say CRU data was used for PET: : :.but then in the next sentence you also say that Zhou et al (2006) method was used to compute PET? Why do you need to compute PET if you already have it from CRU. Similarly, you say APHRODITE precip data is used but the previous section and the next section indicate that MRI-AGCM model data is used for the hydro modelling? Maybe you used APHRODITE for the bias correction or maybe APHRODITE was used as the baseline data and the perturbed based on climate change factors from the MRI-AGCM??? Either way some more detailed explanation is required as to what you actually used to run the hydro model (under both the current and future climate simulations).

RESPONSE: Thanks for the comments. We apologies for the confusion with this paragraph. We have removed it. It was mistakenly left from the original draft (which mentioned about the other application of the BTOPMC model).

Point #5

COMMENT: Other problem is you have just used one GCM and just one emissions scenario (and it is an out of date emission scenario also, IPCC has moved from SRES to RCP several years ago now). I realise at start of Sect 3 you explain you just use one GCM projection as proof of concept. This I guess is ok in a paper like this where you are just demonstrating a method but given one of your main claims is that this FDC-DDC method is easy to apply it should be the case that running multiple GCM/emission scenarios through the method and comparing the differences should be ok. This would make your argument for the acceptance of this method more convincing (as would inclusion of a few other case study locations—as per comment #2). Assuming one GCM is enough to demonstrate your concept I guess is possibly ok: : :: : :but what is definitely not ok is to then make concluding statements that suggest that what the findings/results from your one GCM example are somehow indicative of what will happen (they might be but there is a lot of uncertainty associated with future projections and you need to convey that). For example, concluding point #7 you say "CC impacts on floods increases": : :: : :: : :.based on your single model study maybe this might be true but that is just one plausible scenario: : :: : :there are many other equally plausible scenarios and, as per latest IPCC findings and many other papers focussing on this region and elsewhere around the world, there is no consensus either way on whether floods will increase or decrease: : :: : :: : :..same issue when you say "impact decreases the necessary storage for drought management,: : :..": : :..this is just based on the single GCM you assessed: : :: : :based on just a single GCM run using just one emission scenario you should not be making such a definite conclusion such as this (which could have quite serious and expensive practical implications if decision-makers accepted and acted on this conclusion).

RESPONSE: Thanks for your comments. Yes, it is true that without analyzing multiple projections, nothing can be concluded. As mentioned above, we decided to remove

the part of climate change analyses from this paper but just mention that the necessary storage indicator would be a valuable tool to analyse the impact of climate change, too.

Point #6

COMMENT: Your concluding point #9: : :..this is a good recommendation to use this FDC-DDC method in as many places as possible to create global maps of the necessary storage and water resources situation: : :: : :: : :.but as per previous comment, to cover the climate change impact bit you need to put something in about repeating this using multiple different future scenarios (i.e. different GCMs, different emission scenarios etc): : :: : :: : :.then you might get towards some sort of consensus. Your method could be applied at major basins around the world using, for example, the GCM info available at the CMIP websites and some appropriate downscaling and bias correction methods (also GCM selection methods if required): : :.this would be a useful exercise but it is pointless doing it based on the outputs from just a single GCM as that doesn't really tell us anything much about what is possible in the future. Refer to some of the work done by CSIRO for the Murray-Darling Basin in Australia and also some of the work done by Mekong River Commission for examples of how to comprehensively assess potential impacts of climate change on water resources (i.e. using climate change projections from multiple GCMs and multiple emission scenarios).

RESPONSE: Thanks for your comments. Once again we agree with you and remove the climate change impact assessment from this manuscript.

---

## Author Comment (AC5) · 1 Jan 2017

RESPONSE TO THE REVIEWER #5'S COMMENTS

We are grateful to Reviewer #5 for the helpful and insightful comments. The provided comments have contributed substantially to improving the manuscript. Accordingly, we have made significant efforts to revise the manuscript with the details being explained as follows.

Point #1

COMMENT: My major review comment on the current manuscript, however, is that the central theme of this manuscript is ambiguous. In the manuscript, I see at least the

following five topics are presented in a mixed manner.

1) Authors promote the application of various traditional analysis approaches, in particular FDC and DDC here, with a large dataset in modern days to obtain practical implications. 2) FDC and DDC curves have been used previously for a dam operation at a single site, while the authors in this manuscript extend the method to spatially distributed data. 3) Authors claim the use of FDC and DDC enabled to characterize necessary storages in the Ganges-Brahmaputra-Meghna (GBM) basin. 4) Authors claim most of recent climate change impact assessment studies simply evaluate the increase or decrease of hydrologic variables. On the other hand, the presented approach with FDC-DDC can provides different perspective to interpret climate change projections suitable for practical water resource management. 5) Finally the authors attempt to present the projected climate change impact in the GBM basin.

I believe all the above issues are equally important. Meanwhile with such a many topics, I found difficulty in understanding the main message by the authors. For example, the introduction mainly reviews the original concept of FDC-DDC with some other similar approaches but not necessary arguing the point of 1). The method section solely reviews the FDC and DDC methods with some extensions to the spatially application i.e. point 2). The result sections including the conclusions focus mostly on 3) - 5), whose issues are not well explained in the introduction.

RESPONSE: Thanks for the comments. We consider 1) and 2) are the core of the paper and 3) is an application case study in GBM. We follow your suggestion omitting climate change part 4) and 5). The use of FDC-DDC 1) for necessary storage calculation is indispensable in this paper as without its practical easiness of calculation of necessary storages at many grid points and production of an areal map would be very difficult. Its introduction is vital as it is not well known while its comparison with other methods was partially done by Takeuchi (1980) and is out of the scope of this paper.

Point #2

COMMENT: Personally I believe this paper can improve the readability if the authors express their own points on the 1) and 2) in result, discussion and conclusion sections. Just revisiting traditional approach cannot be accepted in a scientific paper, but this is not the case with demonstrating further extensions.

RESPONSE: Thanks for the comments. As the 2) spatial distribution of necessary storages is the main theme of the paper it is discussed in all sections, while the 1) FDC-DDC is only the way of calculating necessary storages discussed in section 2 only. But as you suggest, its methodological needs for this application will be discussed in result, discussion and conclusion sections.

Point #3

COMMENT: Please add some more explanations on the practical use of the quantified necessary storages for river basin managers. Especially for such a large river basins, the meaning of smoothing discharge at a particular river section should be carefully discussed. For example, smoothing river discharge at an upstream point with smaller storage and at a downstream point with large storage have different impacts for both flood (at the downstream of the reservoirs) and drought. Hence I wonder for the effective use of the information, it requires some additional information such as the impact of smoothing to the downstream areas etc. for practical applications. This comment does not request for additional analysis but requesting for how the spatially distributed necessary storage information can be used in practices.

RESPONSE: Thanks for the comments. You raise the most important question. Frankly we do not have the satisfactory answers yet. But we will discuss as much as possible on the potential use of necessary storage information to water managers. One may be an implication of spatial differences of necessary storages in months in relation to potential benefit of water transfer. Another would be that area with smaller storages may indicate relative advantage for agricultural use. Also, your point of impact to downstream is a question of changes of necessary storages along river lines that may indicate the

advantageous site of dam construction in hydrological sense.

Point #4

COMMENT: P2 L24-26 The part of "its scale is different from that of elementary hydrological processes in a small catchment" is unclear. The similar sentences appear also in 4.2.2 describing Representative Elementary Area (REA), but the current manuscript is still unclear how the scale issue dealt in this study is related to REA.

RESPONSE: Thanks for the comments. Our understanding of REA is the smallest area over which a measurement can be made that will yield a value representative of the whole. In various hydrological phenomena, they are about 1km2 but in storage domain, it seems in much higher order. This will be mentioned in discussion.

Point #5

COMMENT: P5 L2 AOEB -> ADEB

RESPONSE: Thanks for your comments. Accordingly, we have revised it.

Point #6

COMMENT: P9 L1 What is the relationship between "WATCH Forcing Data set (WFD)" and previously described datasets including CRU and APHRODITE in 3.2.2.1 in the presented simulation.

RESPONSE: Thanks for your comments. We apologies for the confusion with this paragraph. We have removed it from Section 3.2.2.1. It was mistakenly left from the original draft (which mentioned about the other application of the BTOPMC model).

Point #7

COMMENT: P12 4.2.1 Please explain the motivation of this discussion at the beginning of this subsection or in the introduction, otherwise this part sounds a bit too sudden and not well connected to the other part.

RESPONSE: Thanks for your comments. Yes, it is. This introduction will be moved to theory section 2 in relation to definition of necessary storages.

---

## Author Response (AR1)

Dear the Editor:

We are grateful for all the reviewers and the editor for their constructive comments and suggestions.

We understand that the main critique was its clarity to avoid misunderstanding and the main message clear. Responding to these comments, we nearly totally rewrote the paper. Thus, instead of showing track-changes, we indicate all modified sections by coloring their titles in yellow. The sentences or words colored in yellow are just highlights that respond some specific comments.

The main revisions and restructuring of the manuscripts are as follows:
1. The manuscript was totally rewritten (except the main part of introduction of methodology chapter 2) with reorganizing the structure and elaborating explanations especially careful elaboration of English.
2. The proposed indictor was clearly explained as a means of characterizing discharge variability and not for reservoir design.
3. In order to make the foci of the paper clearer, in Introduction 1, three main objectives were distinctly stated:
   1) Proposal of necessary storage as an indicator of hydrological variability,
   2) Geographical distribution of necessary storage and
   3) Use of FDC-DDC to calculate necessary storage.
4. Climate change analyses and related discussions were entirely removed.
5. Major improvement was made to chapter 4. Result and discussion on necessary storage of GBM basins in relation to hydrological heterogeneity of the basin, by adding the following:
   1) Analyses of geographical distribution of necessary storages.
   2) Comparison with other statistical indicators, SD and CV.
   3) Their zoom up in the selected 12 sub-basins to highlight the effects of hydrological heterogeneity of sub-basins.
   4) Some potential area of future investigations using necessary storages for water resources management and analyses of scale effects.
   5) In order to support such deeper discussions, the eight figures Figs. 2, 3a, 12, 13, 14, 15, 16 and 17 were added.

We hope those modification cover all the points raised by the kind reviewers and the editor.

Sincerely yours,

Kuniyosi Takeuchi and Muhammad Masood

**Reference Number**: hess-2016-525

**RESPONSE TO THE REVIEWER #1'S COMMENTS**

*We are grateful to Reviewer #1 for the helpful and insightful comments. The provided comments have contributed substantially to improving the manuscript. Accordingly, we have made significant revision as follows.*

**General comments**
**Point #1**

***COMMENT:*** *Impact of climate change on necessary storages is assessed based on assumed target QT = Qmean, 3Qmean, 0.5Qmean. How can be applied for realistic conditions of the basin? If storage from 75 artificial dams in the Ganges are included as flood detention capacity or flood channel capacity, what will be necessary storage during flood?*

***RESPONSE:*** The focus is limited to variability of discharge due to hydrological heterogeneity and not socio-economic activities (stated clearly at the beginning of 1. Introduction). In order to make the focus clear and avoid confusion, we removed the part of climate change impact assessment.

**Specific comments**
**Point #1**

***COMMENT:*** *P5 L2:": : : = AOEB" should be ": : := ADEB".*

***RESPONSE:*** Corrected (p6, L2).

**Point #2**

***COMMENT:*** *P5L9: not found ": : :discussed in 5.1"*

***RESPONSE:*** Corrected (p6, L9) to 2.3.2.

**Point #3**

**COMMENT:** *P5L19: if m' = 150 days, it is not consistent with the location of it on horizontal axis in Figure 3, where m' should be less than 50 days.*

**RESPONSE:** The schematic DDC curve has been changed to a more realistic one (Fig.5 and 6).

**Point #4**

**COMMENT:** *P5L30: ": : := A'OEB' " should be ": : := A'D'EB' "..*

**RESPONSE:** Corrected (p6, L30).

**Point #5**

**COMMENT:** *P9L29: what is the duration (m) for the results in Figure 7.*

**RESPONSE:** "m" is different at each point and it was not analyzed here.

**Point #6**

**COMMENT:** *P10L5: ": : :in Fig. 5." should be ": : :in Fig. 6."*

**RESPONSE:** It has been corrected to Fig 8c.

**Point #7**

**COMMENT:** *P11L9-10: ": : :in Fig.9a" should be ": : :in Fig. 10a", ": : :in Fig.9b" should be ": : :in Fig. 10b". Define (a) and (b) in Figure 10.*

**RESPONSE:** All figure numbers were changed and a), b), c) and d) were indicated in each figure.

**Point #8**

**COMMENT:** *P11L20: what is the Hurst coefficient for GBM?*

**RESPONSE:** It is not calculated and out of scope of this paper.

**Point #9**

*COMMENT: Figure 1: include legend for 5 10 20 50 years and long-term mean discharge.*

*RESPONSE:* The legend is added in Figure 4.

**Point #10**

*COMMENT: Figure 3: if m' is on the left of m, the value of m' < 50 days.*

*RESPONSE:* The schematic DDC curve was changed in Fig. 5 and 6 to a more realistic one.

**Point #11**

*COMMENT: Figure 6-8, 10-11, what is the unit of both axis? Basin boundary presented by green and/or red line make confusion with color legend of necessary storages.*

*RESPONSE:* In the new geographical maps Fig. 3, 8-11, basin boundaries are indicated in black lines (GBM boundaries are bold and selected inner sub-basin boundaries are thin) and country borders are in white. Also E (degree East) and N (degree North) are indicated.

**Point #12**

*COMMENT: Table 2: there is no comment and discussion for the result in this Table.*

*RESPONSE:* Table 2 is removed as climate change analysis was taken out of the paper.

**RESPONSE TO THE REVIEWER #2'S COMMENTS**

*We are grateful to Reviewer #2 for the helpful and insightful comments. The provided comments have contributed substantially to improving the manuscript. We have made the following specific revisions to the points you commented:*

**Point #1**

**COMMENT:** *This is a very stimulating study on the geographical variations of "necessary storages". The sample river basin is the Ganges-Brahmaputra-Meghna basin in South Asia. Distributed river discharge data was calculated by a numerical hydrological model, BTOPMC.*

*What is "necessary storage" is not easy to describe very simply here in a limited space. But, I try to explain shortly what "necessary storage" is in this study as far as I can understand. Let's take a point or a place or a location in a river basin (and in the river channel network of the basin). In terms of "necessary storage" for flood, at the specified location, let's try to imagine how much volume of storage is necessary to keep flood water in the storage for releasing only reasonable water-flow to the downstream. In terms of "necessary storage" for drought, let's imagine how much volume of storage is necessary at the specified location to maintain required water flow during a drought period in the downstream. Here, they do not mean an actual dam-reservoir would be constructed. "Necessary storage" is a virtual or hypothetical variable for representing a hydrological characteristic at the specified point in a river basin or in a river channel network. Please refer to the manuscript for more detailed and exact definitions. The authors expanded relatively old methodologies published in 1970's and 1980's or sometimes in 1950's and 1960's, and tried to apply those methodologies to a continental-scale or outputs of a continental-scale numerical hydrological simulation.*

**RESPONSE:** Thank you very much for your positive comments and your clear interpretation of "necessary storage". It is the necessary storage space to be made available to keep flood water in and necessary storage prepared to augment drought flow to keep releasing the certain specific amount of water. It is used not for reservoir design but as an indicator of discharge variability in time to study its relation to geophysical and geographical characteristics of a basin. The foci of this paper are the importance of the study of hydrological heterogeneity in storage domain and for that purpose the use of FDC-DDC based calculation is useful. It is clearly stated in 1. Introduction.

**Point #2**

*COMMENT: In the above, I mentioned "stimulating". It is true. I basically enjoy reading this manuscript. Their trial is somehow a novel one which I have never seen. In addition, this is a paper for the special issue in Honor of Eric F. Wood. Prof. Wood carried out studies on probabilistic aspects of hydrology in his 70's and 80's, and he carried out continental or global-scale modeling and remote-sensing studies in 90's and in 21$^{st}$ century. Thus, this manuscript by Takeuchi and Masood is very good for the special issue because they try to connect studies in 50-80's to contemporary continental-scale research. We sometimes neglect studies done in 50's, 60's and 70's. In this occasion, this can be a valuable paper.*

*However, I should mention that the structure and presentation of this manuscript is far from satisfactory. I do not know when is the deadline of this special issue, but I should demand thorough revision to the manuscript in terms of structure and presentation. I do not refer to problematic sentences and figures point-by-point, but I believe the authors can completely re-write this paper because they have experiences of writing papers in major journals.*

*RESPONSE:* Thank you for the encouragement. According to your suggestion, we completely rewrote the manuscript in structure as well as presentation, especially clarifying the focus and intention. The concrete list of revision is in the next Point #3.

**Point #3**

*COMMENT: Apart from the structure and presentation of the manuscript, for the contents of this paper, I also have several comments and I would recommend additional analyses in the following.*

*At first, the impact of climate change is not much important (and not much interesting, at this moment) in the context of this research. Thus, I recommend the authors to remove the aspects of climate change impact from this paper. Then, the paper will be a bit much more organized.*

*RESPONSE:* Thank you for your comment. We agree that a sharper focus of the paper is necessary to make the objective of the paper clearer. Accordingly, we remove the climate component of the presentation. This accords with quite many comments and suggestions of other reviewers, too.

The main revision and restructuring of the manuscripts are as follows:

6. Rewriting the whole manuscript (except introduction of methodology chapter 2) with clearer structuring and explanations.
7. Introduction 1 states three main foci clear:
   1) Proposal of necessary storage as an indicator of hydrological variability,
   2) Geographical distribution of necessary storage and

3) Use of FDC-DDC to calculate necessary storage.

8. Climate change analyses and related discussions (originally 3.2.1, 4.1.2 etc.) were entirely removed.

9. Major improvement was made to chapter 4. Result and discussion on necessary storage in relation to hydrological heterogeneity of the basin, namely, the following are added:

    6) Geographical distribution analyses,

    7) Relation with statistical indicators of GBM basins

    8) Their zoom up in the selected 12 sub-basins.

    9) In order to support deeper discussions, the following figures were added: Fig. 2 land cover, 3a elevation, Fig. 12 $V_{km3}$-A relation, Fig. 13 $V_{months}$-A relation, Fig. 14 $V_{km3}$-SD relation, Fig. 15 $V_{months}$-CV relation, 16 zoom up of Fig. 13 and 17 zoom up of Fig. 15 in 12 sub-basins

10. Case study area (originally 3.1) was introduced in an early stage in Chapter 1.

11. Introduction of range theory (originally 4.2.1) was merged to 2.3 with other theories.

12. Captions of tables and figures are improved such as adding a legend of Fig. 4 and sub-figure numbers such as a)-d) etc.

**Point #4**

**COMMENT:** *Next, although the authors can show the geographical distributions of outputs and variables as basic information like as Figure 6, 7, 8, I do not think geographical distribution is enough for what the authors want to discuss. As clearly seen in those figures in this paper, values of "necessary storages" are very much different between the main stream of a large river and a tributary. Whether the pixel is in the main stream (where the number of upstream pixels is several hundreds) or the pixel is in a tributary (where the number of upstream pixels is only several) affects "necessary storage" a lot. In accordance with what I wrote just above, the fact that two lines for Ganges and Brahmaputra in Figure 12 are separate is not much surprising. Those two lines just correspond to the main streams of two large rivers. Actually, dots in the same line (for the same major river) should not be treated independently.*

**RESPONSE:** To improve exactly such insufficiency, the major improvement was made to chapter 4 by adding 4.2 the analyses of relations with statistical indicators SD and CV and 4.3 zoom up of those in the selected 12 sub-basins.

Also, in order to support such discussions, the many figures were added, namely, Fig. 2 land cover, 3a elevation, Fig. 12 $V_{km3}$-A relation, Fig. 13 $V_{months}$-A relation, Fig. 14 $V_{km3}$-SD relation, Fig. 15 $V_{months}$-CV relation, 16 zoom up of Fig. 13 and 17 zoom up of Fig. 15 in 12 sub-basins. The related work of Gao et al. (2014) was also introduced in 4.4.1.

**Point #5**

**COMMENT:** *What the authors want to show should not be the difference in "necessary storage" due to the difference in the number of upstream pixels (= upstream area). The difference in "necessary storage" due to the difference in the number of upstream pixels is not hydrological heterogeneity. In Figures 7 and 8, what can be seen most clearly is main streams of major rivers. That is probably not what the authors firstly want to show.*

*Rather, what the authors may be able to show is the impact of various parameters (like soil types, geology, arid or wet, snow-affected or not, hilly or not) on "necessary storages". For such an analysis, areas of catchments for analysis should be the same or similar. Such an analysis roughly corresponds to dots in the left edge in Figure 12. Why dots in the left edge in Figure 12 are very much scattering?*

**RESPONSE:** Although such analyses are not enough, the authors tried to relate necessary storage with land cover characteristics as much as possible in 4.1 and 4.2. But the major analyses are left for the future study as stated in 4.4.2 and in conclusion 3), 8) and 9).

**Point #6**

**COMMENT:** *Even when the authors want to show the relation of "necessary storage" to catchment area, the authors are required to take different independent catchments; at least, even in the same large river system, two independent tributaries should be taken for analysis. Or, in Figure 12, two different major rivers are converging to two different lines/points. This is somewhat interesting. Thus, another result which may be highlighted is various different converging lines/points of various major rivers of the world. For it, the simulation of the Ganges-Brahmaputra-Meghna only is not enough. Also, as written in the abstract, if the authors want to highlight representative elementary area of necessary storages which may be seen in Figure 12, more detailed analysis and discussion are appreciated. Even when the samples are only two (Ganges and Grahmaputra), detailed analysis and discussion will of help. By the way, what about Meghna?*

**RESPONSE:** As shown in 4.3, the selected 12 sub-basins, 8 from the Ganges, 3 from the Brahmaputra and 1 from the Meghna were analyzed and the scatter from the average $V_{months}$-SD and -CV were attempted to relate with their sub-basin characteristics.

**Point #7**

*COMMENT: I really appreciate the contribution of the authors to this special issue by trying to connect relatively old studies for catchment or local scale in 50's-80's to modern continental-scale research in the 21st century. This would be a very good contribution to the special issue for the Honor of Eric F. Wood. However, as a summary of what I mentioned above, and for making this manuscript to be published in HESS, I would recommend as follows:*

*- thorough revision to the presentation and structure of the manuscript. Here, I at least point out that: a) "case study area" and "data used" should come earlier because even Figure 1 uses the data, b) 4.2.1. may be combined with 2 as a theoretical framework, and how 4.2.1. was used for analysis, discussion and conclusion is unclear, c) captions of figures and tables should be written with enough information to understand figure/table adequately..*

*RESPONSE:* Thanks for your very specific comments. All a)-c) were exactly incorporated as described in Point #3.

**Point #8**

*COMMENT: - I would recommend to stop showing the impact of climate change for making this manuscript simple, concise, clear, and appealing. I do not mean, climate change impact assessment is not of interest. But, because of time limitation and for highlighting what the authors want to say, I would recommend the authors to focus on hydrological characteristics under the current climate condition.*

*RESPONSE:* Yes, all climate change discussions were removed from the text. Accordingly, the discharge data were recalculated by the WFD dataset instead of MRI-AGCM3.2S.

**Point #9**

*COMMENT: - Try to carry out additional analyses focusing on independent tributary catchments (which have similar catchment areas, and belonging to different tributaries). For "necessary storages" of those catchments, the authors may be able to show influential parameters (soil types, geology, arid or wet, snow-affected or not, hilly or not) in a quantitative or qualitative way. I know, dots in the left edge in Figure 12 are outputs after standardization by average monthly flow. That is good. But, just showing a huge vertical scatter is not enough. We want to know what makes this vertical scatter. Only speculation is written in this manuscript. Additional analyses are necessary, even though this paper is a first trial and a first step.*

*RESPONSE:* We hope our 12 sub-basin analyses are barely enough as a first trial and a first step.

**Point #10**

*COMMENT: - Of course, two different converging lines for Ganges and Brahmaputra is interesting, but at least a few more examples would be appreciated. Also, discussion on why those two converging lines are different should be made.*

*RESPONSE:* We hope discussion 4.3 is barely enough.

**Point #11**

*COMMENT: Finally, again, a stimulating paper is very important. Nowadays, well-organized but nonstimulating papers are dominating. That's not fun. But, well-organized or adequately organized is necessary for publication. Depending on the focus of this study, the authors may not take all the points I raised.*

*RESPONSE:* We hope we could cover all the points you kindly suggested although only for the first step.

**Reference Number**: hess-2016-525

**RESPONSE TO THE REVIEWER #3'S COMMENTS**

*We are grateful to Reviewer #3 for the helpful and insightful comments. The provided comments have contributed substantially to improving the manuscript. Accordingly, we have made significant efforts to revise the manuscript with the details being explained as follows.*

**General comments**
**Point #1**

**COMMENT:** *The novelty of the proposed method is not clear. The method appears to be a minor adjustment to previously published versions of this approach. The authors note this in page 3, lines 3-5. Although in my reading of Takeuchi (1986), it appears that the methodology presented using both the FDC and DDC may not have been previously published. Regardless, there are fairly well-established methods to determine the necessary storage in hydrologic design; some of these papers are mentioned briefly in the introduction but there is no attempt to demonstrate the utility of this approach in the context of these other well-established methods - not necessarily show this method is better but - at a minimum - that it performs as well as other methods.*

**RESPONSE:** Although only brief, the difference was mentioned in 2.3.1 including a reference on performance comparison.

**Point #2**

**COMMENT:** *This approach is based on the flow-duration curve, which does not consider the timing or variability of the discharge and, therefore, the accumulation or depletion of storage over time. It is then not clear how this approach can be useful, as it does not consider the storage in the previous time step, particularly for rivers where variability is large and storage is most needed to control this variability.*

**RESPONSE:** We hope explanation in Section 2.2 is enough to explain most of those questions. The storage in the previous time step is important when it is applied to reservoir operation as shown in Takeuchi (1986) but not to this indicator development.

**Point #3**

**COMMENT:** *Following from comment 2, it is repeated throughout the manuscript, "storage is the means to control discharge variation." The relation between variation in discharge and necessary storage is well established in the literature, with more storage needed as the coefficient of variation in the discharge increases; and yet, there is no consideration of this point in the demonstration of these methods. The coefficient of variation (CV) is not reported for the 3 demonstration sites so the reader has no idea of the variation of the discharge that is being "smoothed" by storage under the calibration/validation dataset.*

**RESPONSE:** Relation with CV is discussed in depth in 4.2 and 4.3.

**Point #4a**

**COMMENT:** *The decisions made in Sections 2.2.1 and 2.2.2 seem quite arbitrary with no justification or support to suggest that these would be choices that a water manager or operator would likely choose.*

**RESPONSE:** Again, this is not for reservoir operation but for an indicator for assessing general ease and difficulty of water resources management.

**Point #4b**

**COMMENT:** *Another example is in the assumption that the FDC and DDC curves follow a generalized extreme value distribution. There are no references provided to support this choice of distribution from previously published work and no evidence is presented to demonstrate that this is a reasonable choice.*

**RESPONSE:** Explanation was added (p4, L38-p5, L2).

**Point #4c**

**COMMENT:** *Page 3, line 28 states that the FDC and DDC curve applies precipitation, yet there is no explanation of this further in the manuscript. How is precipitation used in the method?*

**RESPONSE:** The statement in (p5, L8-12) is just an introduction of DDC-FDC in general that includes some information not used in this paper.

**Point #4d**

**COMMENT:** *The methodology described on p. 5, line 6 states that the "interest duration is limited to a year" but I am left to wonder how the analysis is applied to rivers where over-year storage is an important component of controlling variability?*

**RESPONSE:** This is discussed in 2.2.1 (p6, L6-9), in 2.2.2 (p6, L35-39) and in 2.3.2 (p8, L 26-31).

**Editorial issues**
**Point #5a**

**COMMENT:** *There are incomplete sentences: p. 8, l. 13; p. 2, l. 12-14.*

**RESPONSE:** For (original p8), an elaboration was added at (p9, L31-32). About (original p2), climate discussion was removed.

**Point #5b**

**COMMENT:** *Acronyms that are not explained before being used. For example, p. 7, l. 10; Abstract, l. 18-19; p. 5, l. 1-2.*

**RESPONSE:** All acronyms are spelled out at least when it appears for the first time.

**Point #5c**

**COMMENT:** *Although I do not recall that HESS has guidance on the use of "he" and "his," I think is a lack of sensitivity to use a gendered pronoun and there are alternative ways to phrase these sentences. Page 4, line 17 is the first appearance but gendered pronouns appear in quite a few other places, such as throughout Section 2.2.2.*

**RESPONSE:** It was replaced by he/she at all 6 places as seen in 2.2.1 (p.5) and 2.2.2 (p.6).

**Reference Number**: hess-2016-525

**RESPONSE TO THE REVIEWER #4'S COMMENTS**

*We are grateful to Reviewer #4 for the helpful and insightful comments. The provided comments have contributed substantially to improving the manuscript. Accordingly, we have made significant efforts to revise the manuscript with the details being explained as follows.*

**Point #1**

**COMMENT:** *The writing/grammar and arrangement of the paper is poor. Several spelling/grammar issues and "awkward" sentences that have to be read a few times to try and understand what the authors are trying to say. Most of the issues are minor but there is too many to list and I suspect this is detracting from the main points the paper is trying to make.*

**RESPONSE:** We thoroughly revised our manuscript and carefully avoided spelling or grammatical mistakes or the awkward sentences as much as possible.

**Point #2**

**COMMENT:** *The concept being introduced is interesting. The FDC-DDC method proposed has some advantages over the mass curve method. It is a simple approach but is also flawed in that stationary hydroclimatic conditions are assumed. Also, a major strength of the FDC-DDC method is its simplicity and transferability (e.g. the authors say it could be used to create global maps of necessary storage and the state of water resources) but then I wonder why if the method is so simple and transferable is it only demonstrated for one basin?*

**RESPONSE:** This is a first trial and a first step. The global application is the next step. But within the GBM basin, 12 sub-basins were analyzed as shown in 4.2 and 4.3 and figures 16 and 17.

**Point #3**

**COMMENT:** *The other major problem I have with this paper is the way the impact of climate change is simulated. Only 3 lines worth of explanation (sect 3.2.1) are given to explain this and it is not clear at all how the GCM outputs were used as inputs to the hydrological modelling? Which variables were used? At what time step? I assume daily (or maybe monthly) and if so there are*

*known to be significant issues associated with daily GCM data and bias correction is usually required? Precipitation data from GCMs is particularly problematic, especially in the Asian monsoon region where this study is focussed. How were the biases associated with GCM outputs addressed?*

**RESPONSE:** In order to avoid the complexity and keep the focus of this paper clear, the climate change components were all removed from the entire manuscript.

**Point #4**

**COMMENT:** *Sect 3.2.2.1, lines 15-21 is also a bit confusing: : :..here you say CRU data was used for PET: : :.but then in the next sentence you also say that Zhou et al (2006) method was used to compute PET? Why do you need to compute PET if you already have it from CRU. Similarly, you say APHRODITE precip data is used but the previous section and the next section indicate that MRI-AGCM model data is used for the hydro modelling? Maybe you used APHRODITE for the bias correction or maybe APHRODITE was used as the baseline data and the perturbed based on climate change factors from the MRI-AGCM??? Either way some more detailed explanation is required as to what you actually used to run the hydro model (under both the current and future climate simulations).*

**RESPONSE:** The confusing paragraph was simplified and restated about the applicability of BTOPMC as 3.1.1 (p.9, L1-11).

**Point #5**

**COMMENT:** *Other problem is you have just used one GCM and just one emissions scenario (and it is an out of date emission scenario also, IPCC has moved from SRES to RCP several years ago now). I realise at start of Sect 3 you explain you just use one GCM projection as proof of concept. This I guess is ok in a paper like this where you are just demonstrating a method but given one of your main claims is that this FDC-DDC method is easy to apply it should be the case that running multiple GCM/emission scenarios through the method and comparing the differences should be ok. This would make your argument for the acceptance of this method more convincing (as would inclusion of a few other case study locations—as per comment #2). Assuming one GCM is enough to demonstrate your concept I guess is possibly ok: : :: : :but what is definitely not ok is to then make concluding statements that suggest that what the findings/results from your one GCM example are somehow indicative of what will happen (they might be but there is a lot of uncertainty*

*associated with future projections and you need to convey that). For example, concluding point #7 you say "CC impacts on floods increases": : :: : :: : :.based on your single model study maybe this might be true but that is just one plausible scenario: : :: : :there are many other equally plausible*

*scenarios and, as per latest IPCC findings and many other papers focussing on this region and elsewhere around the world, there is no consensus either way on whether floods will increase or decrease: : :: : :: : :..same issue when you say "impact decreases the necessary storage for drought management,: : :.." : :..this is just based on the single GCM you assessed: : :: : :based on just a single GCM run using just one emission scenario you should not be making such a definite conclusion such as this (which could have quite serious and expensive practical implications if decision-makers accepted and acted on this conclusion).*

**RESPONSE:** As mentioned before, the entire climate change analyses was removed from the manuscript.

**Point #6**

**COMMENT:** *Your concluding point #9: : :..this is a good recommendation to use this FDC-DDC method in as many places as possible to create global maps of the necessary storage and water resources situation: : :: : :: : :.but as per previous comment, to cover the climate change impact bit you need to put something in about repeating this using multiple different future scenarios (i.e. different GCMs, different emission scenarios etc): : :: : :: : :.then you might get towards some sort of consensus. Your method could be applied at major basins around the world using, for example, the GCM info available at the CMIP websites and some appropriate downscaling and bias correction methods (also GCM selection methods if required): : :.this would be a useful exercise but it is pointless doing it based on the outputs from just a single GCM as that doesn't really tell us anything much about what is possible in the future. Refer to some of the work done by CSIRO for the Murray-Darling Basin in Australia and also some of the work done by Mekong River Commission for examples of how to comprehensively assess potential impacts of climate change on water resources (i.e. using climate change projections from multiple GCMs and multiple emission scenarios).*

**RESPONSE:** The climate change analyses were removed from the manuscript.

**Reference Number**: hess-2016-525

**RESPONSE TO THE REVIEWER #5'S COMMENTS**

*We are grateful to Reviewer #5 for the helpful and insightful comments. The provided comments have contributed substantially to improving the manuscript. Accordingly, we have made significant efforts to revise the manuscript with the details being explained as follows.*

**Point #1**

*COMMENT: My major review comment on the current manuscript, however, is that the central theme of this manuscript is ambiguous. In the manuscript, I see at least the following five topics are presented in a mixed manner.*

*1) Authors promote the application of various traditional analysis approaches, in particular FDC and DDC here, with a large dataset in modern days to obtain practical implications.*
*2) FDC and DDC curves have been used previously for a dam operation at a single site, while the authors in this manuscript extend the method to spatially distributed data.*
*3) Authors claim the use of FDC and DDC enabled to characterize necessary storages in the Ganges-Brahmaputra-Meghna (GBM) basin.*
*4) Authors claim most of recent climate change impact assessment studies simply evaluate the increase or decrease of hydrologic variables. On the other hand, the presented approach with FDC-DDC can provides different perspective to interpret climate change projections suitable for practical water resource management.*
*5) Finally the authors attempt to present the projected climate change impact in the GBM basin.*

*I believe all the above issues are equally important. Meanwhile with such a many topics, I found difficulty in understanding the main message by the authors. For example, the introduction mainly reviews the original concept of FDC-DDC with some other similar approaches but not necessary arguing the point of 1). The method section solely reviews the FDC and DDC methods with some extensions to the spatially application i.e. point 2). The result sections including the conclusions focus mostly on 3) - 5), whose issues are not well explained in the introduction.*

*RESPONSE:* In order to avoid confusion, three objectives were clearly stated in Chapter 1. The first objective is to propose necessary storage as an indicator of hydrological variability. The second is to show how good it is for analysing hydrological heterogeneity. The third is to introduce

FDC-DDC as a recommended methodology to calculate necessary storage.

About 4) and 5), we omitted the application to climate change from the text.

**Point #2**

*COMMENT: Personally I believe this paper can improve the readability if the authors express their own points on the 1) and 2) in result, discussion and conclusion sections. Just revisiting traditional approach cannot be accepted in a scientific paper, but this is not the case with demonstrating further extensions.*

*RESPONSE:* All three objectives were discussed and summarized in conclusion.

**Point #3**

*COMMENT: Please add some more explanations on the practical use of the quantified necessary storages for river basin managers. Especially for such a large river basins, the meaning of smoothing discharge at a particular river section should be carefully discussed. For example, smoothing river discharge at an upstream point with smaller storage and at a downstream point with large storage have different impacts for both flood (at the downstream of the reservoirs) and drought. Hence I wonder for the effective use of the information, it requires some additional information such as the impact of smoothing to the downstream areas etc. for practical applications. This comment does not request for additional analysis but requesting for how the spatially distributed necessary storage information can be used in practices.*

*RESPONSE:* It is beyond the scope of this paper but indicated three potential areas of future investigation in the new section 4.4.1 Potential use of spatially distributed necessary storage information for water resources management (p14).

**Point #4**

*COMMENT: P2 L24-26 The part of "its scale is different from that of elementary hydrological processes in a small catchment" is unclear. The similar sentences appear also in 4.2.2 describing Representative Elementary Area (REA), but the current manuscript is still unclear how the scale issue dealt in this study is related to REA.*

*RESPONSE:* It is premature to relate the necessary storage behavior to REA, but some thought

was mentioned briefly in 4.4.2. (p14)

**Point #5**

*COMMENT:* *P5 L2 AOEB -> ADEB*

*RESPONSE:* We have corrected it (p6, L2).

**Point #6**

*COMMENT:* *P9 L1 What is the relationship between "WATCH Forcing Data set (WFD)" and previously described datasets including CRU and APHRODITE in 3.2.2.1 in the presented simulation.*

*RESPONSE:* We have removed unnecessary introduction of CRU and APHRODITE originally in 3.2.2.1 from the text.

**Point #7**

*COMMENT:* *P12 4.2.1 Please explain the motivation of this discussion at the beginning of this subsection or in the introduction, otherwise this part sounds a bit too sudden and not well connected to the other part.*

*RESPONSE:* This introduction was moved to theory section 2.3.2 in relation to introduction of the method of calculating necessary storages.

[revised manuscript text omitted]

---

## Editor Decision (ED1)

[revised manuscript text omitted]

[ここに入力]

---

## Author Response (AR2)

Letter to the editor and reviewers

Dear the editor and reviewers:

We, the authors, are extremely thankful to the editor and reviewers. We have considerably revised the manuscript according to your suggestions and comments. Especially, we reconsidered the way of delivering the main messages of the paper and decided to change the title of the paper. Also, we much augmented the discussion and analyses in Section 4 including an addition of geographical and land cover information in Fig. 17 and made thorough English editing thanks to a professional editor at ICHARM. To indicate the records of revision, we attach the track-change version of the manuscript. We appreciate your kind review.

Sincerely yours,

Kuniyoshi Takeuchi and Muhammad Masood

[ここに入力]

Report #1

The authors made substantial revision, I admit. Nevertheless, I would argue that Results and Discussion is not described in an organized way, which means major revision is still necessary. I would say this is not a draft that is fully elaborated before submission. I also do not think figures are very much elaborated before submission.

The authors mention there are three objectives.
1) to propose necessary storages to smooth out discharge variation during flood and drought as a new indicator of discharge variability in time.
2) to analyse the geographical distribution of such the indicator to relate hydrological variability with various hydrological heterogeneity of a basin.
3) to introduce an efficient methodology to calculate necessary storage at all grid points of a basin to make their geographical analyses possible, that is, the flood duration curve and drought duration curve (FDC-DDC) method.

1st and 3rd objectives are met to a certain extent. However, 2nd objective is not well achieved. That is equal to what I wrote "Results and Discussion is not described in an organized way."

I would continue more on 2nd objective.

**Point #1**
It is difficult to get organized results from Figures 8-11. Well, those figures could be just basic information. Then, Figure 12-17 should bring us organized messages to achieve 2nd objective. But, what I can see is that the new index is generally correspondent to SD or CV of discharge. (That is somewhat contradictory to what the authors intended to, because the authors want to show us a new index which can show us something different from CV and/or SD. ) Anyway, any key messages are not brought to me.

**Response:**
Thank you for the comment.
We consider necessary storage is a signature of discharge variability having distinct advantages over SD and CV in two respects: One is its physical meaning to indicate difficulty and ease of smoothing hydrological variability for water resources management and the other to indicate not only magnitude of variability but also persistence characteristics of variability which determines necessary storage. In order to make this point clearer and better understandable, we made the key word of the title of the paper changed from an indicator to a signature. We believe that the new title indicates the message of the paper better than the previous version.
Its relation to basin hydrological heterogeneity was not enough analyzed yet in this paper but some trial examinations were made. Namely, we considered Hurst equation Eq. 8-10 (proportional to SD and CV with unknown coefficients) as a guide and tried to see the different coefficients in different basins. We identified the average relation between necessary storage and SD and CD in the GBM basin and examined obvious departure from and scatter around the average relation in each sub-basin. We tried to relate this scatter and departure with the basin characteristics. Preliminary results indicate some interesting observations for further studies.

[ここに入力]

One of such observations is as mentioned in 4.3 10) and 11) that the scatter is especially large in extended cropland with semi-arid climate influence in the western Ganges and small in the thick forests with much precipitation with snow in the southern hillslopes of the Himalayan mountains.

There are quite a few other observations such as mentioned in 4.2 2) the difference between necessary storage for flood and drought when the smoothing length is limited to over a year. It is considered indicative of the need of inter-annual smoothing for drought.

**Point #2**

What' seen for the main streams and what's seen for tributaries must be clearly separated. Then, both for the main streams, and for tributaries, the authors are recommended to clearly show results and discussion for 2nd objective. Current analyses are descriptive and in a very much scattered way, and it is not sure whether horizontal axis used in Figure 12-17 are adequate enough. The authors are encouraged to try to deeply think about how to best convey their messages to the audience by showing much adequate figures, rather than by just showing quickly made figures.

**Response**

Thank you for the considerate comments.

Separated presentation in main stream and tributaries is attempted in Fig 13 and 15 their zoom-ups in Fig. 16 and 17. Main streams were better analyzed by Fig. 13 and 16 with A for x-axis and small catchments by Fig. 15 and 17 with CV for x-axis. It is true we could not manage to get any clear results on either main stream or tributaries. But some focus points were identified such as on the main stream, the converging months and slopes represent the basin characteristics and on tributaries, scatter and departure from the average relation between $V_{month}$-CV relation do. As stated in conclusion the global maps of necessary storage will reveal clearer nature of the signature with basin characteristics for the main stream and, as stated in 4.2 6), much more elaborate geographical information than Fig. 2 and 3 would reveal clearer relation with basin characteristics for tributaries.

Unfortunately, we could not examine different x-axis coordinates. But added quite much discussion on converging months and slopes of $V_{month}$-CV relation and, scatter and departure of points from the average relation. This is only the first step and further analyses of the detail are left for the next step, which was clearly stated in the first paragraph of 5. Conclusion.

**Point#3**

Something interesting must be embedded in this paper. But, presentation particularly in Results and Discussion is not enough. I am not saying what's written in Results and Discussion is wrong. Rather, text and figures are not presented in an organized way. Those are presented in a way such as a young student presenting in a hurry his/her material to his/her teacher. For example, geographical distributions shown in Figures 16-17 could be interesting, but it is difficult to get organized discussion and conclusions from those figures and its corresponding main text.

**Response:**

Thank you so much for your critique.

[ここに入力]

We admit it and considerably revised 4. Results and Discussion adding many further analyses including adding a numerical summary of Fig. 2 and 3 to each sub-basin diagram of Fig. 17. But we found the analyses with the additional information is not quite enough and need more elaborated research.

**Point#4**

The authors wrote "its relation with geophysical and geographical conditions was examined" (in Abstract) and "By this, the relation between hydrological variability and geographical heterogeneity can be better understood" (in 1.1.2). However, what the authors wrote as such is not much realized/shown in this paper.

**Response:**

Thank you for your comment.

We considerably revised 4. Results and Discussion chapter including relation with hydrological and geographical heterogeneity as much as possible. Nevertheless, we admit this is just an initial trial which was emphasized in abstract, introduction and conclusion to avoid misunderstanding.

**Point#5**

By the way, even Abstract is far from adequate. In general, Abstract is not divided into paragraphs. In addition, too-much specific details and long introductory description should be deleted from Abstract.

**Response:**

Thank you indeed for your comment.

We totally revised abstract making it sharp and concise and avoiding misleading indications.

[ここに入力]

Report #2

I think the authors have done a good job addressing the comments from the reviewers. However, I make the following comments which, if possible, should be addressed or clarified:

**Point#1.**

In my original comment #2 I mentioned that the proposed FDC-DDC method has a flaw in that it assumes stationary hydroclimatic conditions. I still don't see where this is acknowledged or addressed.

10 **Response:**

Thank you for the comment.

It was clearly stated in Introduction 1.1.3 line 21.

**Point#2.**

15 In my original comment #4 I mentioned that I was confused about whether PET was an input or whether it was computed within the hydrological model. In the original version of the paper it was misleading because you seemed to indicate that both were happening (i.e. you said PET came from CRU but then you also said it was computed within the BTOPMC as per the Zhou et al. (2006) method). In response you say you removed the confusing paragraph and replaced it with more simplified text. It is true that there is now no confusion as

20 to whether CRU data is used for PET or whether PET is calculated within BTOPMC as per Zhou et al. (2006) but that is because all mention of PET is now removed……and also the reference to Zhou et al. (2006) is removed……so, in short, I am still confused about what you did to get PET? Sect 3.1 covers precip, temp and discharge data used……but nothing on PET which suggests PET is calculated within the model (since you don't describe the PET input data)….but then Sect 3.2 and Sect 3.3 (the bits on how discharge is calculated

25 and model set up and validation) don't explain how PET was obtained or used either. I suggest you need to clarify this……if PET is not an input and is calculated via BTOPMC as per Zhou et al. (2006) then that reference needs to be brought back in and some summary info on the method provided….if PET is an input and not calculated within the model then where does it come from? And how is the model set up to use it?

30 **Response:**

Thanks for the comment.

The BTOP model requires input of potential evaporation from canopy (PETo) and potential evapotranspiration from soil (PET) to estimate actual evaporation ($ET_0$) and actual evapotranspiration (ET). In the current version of the BTOP model, PETo and PET values are computed using the Shuttleworth-

35 Wallace (S-W) equation. The parameterization of the S-W model was given in detail by Zhou et al. (2006). Please see the User Manual of BTOP model for details about the computation of PET.

**Point#3.**

40 Following from previous comment….APHRODITE data was previously mentioned as something that was used…..now there is no mention of APHRODITE……and you indicate that WFD was used instead for precip and temp….was there any comparison or sensitivity analysis done to assess possible implications of this

[ここに入力]

decision? Do APHRODITE and WFD perform similarly over your study area for precip and temp? How did you decide that WFD info was satisfactory?

**Response:**

Thanks for the comment.

In our previous study, we have compared WFD and APHRODITE dataset. Spatial distribution of annual (1988) precipitation of the WFD and the APHRODITE over entire GBM basin and difference between two data are shown in Fig. 1. We have simulated a hydrological model by using the APHRODITE precipitation and temperature data to compare the model output with that of using WFD dataset. We found the simulation using APHRODITE precipitation and temperature data does not give better simulation results than the simulation using WFD (simulation result presented in Table 1). Time series plot of simulated discharge using both (i) complete dataset from the WFD and (ii) combination of precipitation and temperature data from the APHRODITE dataset and other metrological variables from the WFD is shown in Fig. 2.

[ここに入力]

[Figure]

(a)  APHRODITE (mm year⁻¹)

[Figure]

(b) WFD (mm year⁻¹)

[Figure]

(c) WFD - APHRODITE (mm year⁻¹)

[Figure]

Figure 1 Spatial distribution of annual (1988) precipitation of the (a) WFD and the (b) APHRODITE over entire GBM basin and (c) difference between two data.

[ここに入力]

[Figure]

Figure 2 Time series plot of simulated discharge using both (i) complete dataset from the WFD and (ii) combination of precipitation and temperature data from the APHRODITE dataset and other metrological variables from the WFD

Table 1: Result obtained from two different simulations (1988) using the APHRODITE and the WFD precipitation data (unit: mm year-1)

|  |  | Rainfall | Snowfall | Total runoff | ET |
|---|---|---|---|---|---|
| APHRODITE | Entire GBM | 1 171 | 27 | 664 | 524 |
|  | Brahmaputra | 1 252 | 9 | 852 | 424 |
|  | Ganges | 959 | 27 | 442 | 537 |
| WFD | Entire GBM | 1 555 | 27 | 1 034 | 538 |
|  | Brahmaputra | 1 819 | 16 | 1 430 | 426 |
|  | Ganges | 1 178 | 18 | 627 | 565 |

[ここに入力]

[revised manuscript text omitted]

[ここに入力]

[ここに入力]